# CLIFFORD NEURAL LAYERS FOR PDE MODELING

**Johannes Brandstetter**
Microsoft Research AI4Science
johannesb@microsoft.com

**Rianne van den Berg**
Microsoft Research AI4Science
rvandenberg@microsoft.com

**Max Welling**
Microsoft Research AI4Science
maxwelling@microsoft.com

**Jayesh K. Gupta**
Microsoft Autonomous Systems and Robotics Research
jayesh.gupta@microsoft.com

## ABSTRACT

Partial differential equations (PDEs) see widespread use in sciences and engineering to describe simulation of physical processes as scalar and vector fields interacting and coevolving over time. Due to the computationally expensive nature of their standard solution methods, neural PDE surrogates have become an active research topic to accelerate these simulations. However, current methods do not explicitly take into account the relationship between different fields and their internal components, which are often correlated. Viewing the time evolution of such correlated fields through the lens of multivector fields allows us to overcome these limitations. Multivector fields consist of scalar, vector, as well as higher-order components, such as bivectors and trivectors. Their algebraic properties, such as multiplication, addition and other arithmetic operations can be described by Clifford algebras. To our knowledge, this paper presents the first usage of such multivector representations together with Clifford convolutions and Clifford Fourier transforms in the context of deep learning. The resulting Clifford neural layers are universally applicable and will find direct use in the areas of fluid dynamics, weather forecasting, and the modeling of physical systems in general. We empirically evaluate the benefit of Clifford neural layers by replacing convolution and Fourier operations in common neural PDE surrogates by their Clifford counterparts on 2D Navier-Stokes and weather modeling tasks, as well as 3D Maxwell equations. For similar parameter count, Clifford neural layers consistently improve generalization capabilities of the tested neural PDE surrogates. Source code for our PyTorch implementation is available at https://microsoft.github.io/cliffordlayers/

## 1 INTRODUCTION

Most scientific phenomena are described by the evolution and interaction of physical quantities over space and time. The concept of fields is one widely used construct to continuously parameterize these quantities over chosen coordinates (McMullin, 2002). Prominent examples include (i) fluid mechanics, which has applications in domains ranging from mechanical and civil engineering, to geophysics and meteorology, and (ii) electromagnetism, which provides mathematical models for electric, optical, or radio technologies. The underlying equations of these examples are famously described in various forms of the Navier-Stokes equations and Maxwell's equations. For the majority of these equations, solutions are analytically intractable, and obtaining accurate predictions necessitates falling back on numerical approximation schemes often with prohibitive computation costs. Deep learning's success in various fields has led to a surge of interest in scientific applications, especially at augmenting and replacing numerical solving schemes in fluid dynamics with neural networks (Li et al., 2020; Kochkov et al., 2021; Lu et al., 2021; Rasp & Thuerey, 2021; Keisler, 2022; Weyn et al., 2020; Sønderby et al., 2020; Pathak et al., 2022).

Taking weather simulations as our motivating example to ground our discussion, two different kinds of fields emerge: **scalar** fields such as temperature or humidity, and **vector** fields such as wind velocity or pressure gradients. Current deep learning based approaches treat different vector field

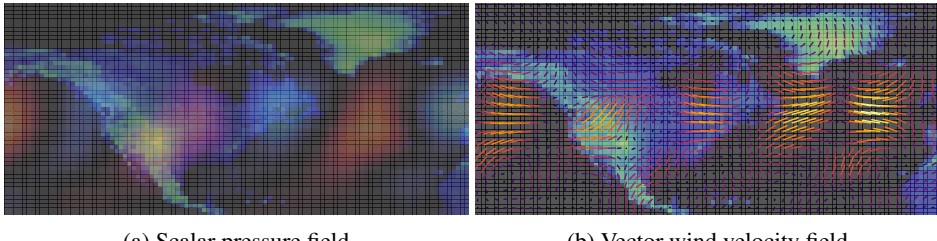

(a) Scalar pressure field  (b) Vector wind velocity field

Figure 1: Fields of the Earth's shallow water model. Vector components of the wind velocities (right) are strongly related, i.e. they form a vector field. Additionally, the wind vector field and the scalar pressure field (left) are related since the gradient of the pressure field causes air movement and subsequently influences the wind components. We therefore aim to describe scalar and vector field as one multivector field, which models the dependencies correctly.

components the same as scalar fields, and stack all scalar fields along the channel dimension, thereby omitting the geometric relations between different components, both within vector fields as well as between individual vector and scalar fields. This practice leaves out important inductive bias information present in the input data. For example, wind velocities in the $x$- and $y$- directions are strongly related, i.e. they form a vector field. Additionally, the wind vector field and the scalar pressure field are related since the gradient of the pressure field causes air movement and subsequently influences the wind components. In this work, we therefore build neural PDE surrogates which model the relation between different fields (e.g. wind and pressure field) and field components (e.g. $x$- and $y$- component of the wind velocities). Figure 1 shows an example of a wind vector field as per the Earth's shallow water model in two dimensions, and the related scalar pressure field.

Clifford algebras (Suter, 2003; Hestenes, 2003; 2012; Dorst et al., 2010; Renaud, 2020) are at the core intersection of geometry and algebra, introduced to simplify spatial and geometrical relations between many mathematical concepts. For example, Clifford algebras naturally unify real numbers, vectors, complex numbers, quaternions, exterior algebras, and many more. Most notably, in contrast to standard vector analysis where primitives are scalars and vectors, Clifford algebras have additional spatial primitives for representing plane and volume segments. An expository example is the cross-product of two vectors in 3 dimensions, which naturally translates to a plane segment spanned by these two vectors. The cross product is often represented as a vector due to its 3 independent components, but the cross product has a sign flip under reflection that a true vector does not. In Clifford algebras, different spatial primitives can be summarized into objects called **multivectors**, as illustrated in Figure 2. In this work, we replace operations over feature fields in deep learning architectures by their Clifford algebra counterparts, which operate on multivector feature fields. Operations on, and mappings between multivectors are defined by Clifford algebras. For example, we will endow a convolutional kernel with multivector components, such that it can convolve over multivector feature maps.

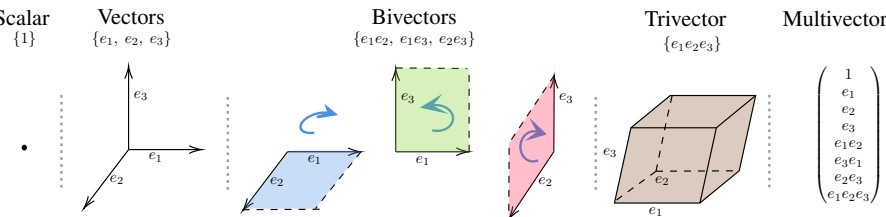

Figure 2: Multivector components of Clifford algebras.

## 2 BACKGROUND: CLIFFORD ALGEBRAS

We introduce important mathematical concepts and discuss three Clifford algebras, $Cl_{2,0}(\mathbb{R})$, $Cl_{0,2}(\mathbb{R})$, $Cl_{3,0}(\mathbb{R})$, which we later use for the layers introduced in Section 3. A more detailed introduction as well as connections to complex numbers and quaternions is given in Appendix A.

**Clifford algebras.** Consider the vector space $\mathbb{R}^n$ with standard Euclidean product $\langle .,. \rangle$, where $n = p + q$, and $p$ and $q$ are non-negative integers. A real Clifford algebra $Cl_{p,q}(\mathbb{R})$ is an associative algebra[1] generated by $p + q$ orthonormal basis elements $e_1, \ldots, e_{p+q}$ of the *generating* vector space $\mathbb{R}^n$, such that the following quadratic relations hold:

$$e_i^2 = +1 \text{ for } 1 \leq i \leq p; \quad e_j^2 = -1 \text{ for } p < j \leq p + q; \quad e_i e_j = -e_j e_i \text{ for } i \neq j . \quad (1)$$

The pair $(p, q)$ is called the *signature* and defines a Clifford algebra $Cl_{p,q}(\mathbb{R})$, together with the basis elements that span the vector space $G^{p+q}$ of $Cl_{p,q}(\mathbb{R})$. Vector spaces of Clifford algebras have scalar elements and vector elements, but can also have elements consisting of multiple basis elements of the generating vector space $\mathbb{R}^n$, which can be interpreted as plane and volume segments. Exemplary low-dimensional Clifford algebras are: (i) $Cl_{0,0}(\mathbb{R})$ which is a one-dimensional algebra that is spanned by the basis element $\{1\}$ and is therefore isomorphic to $\mathbb{R}$, the field of real numbers; (ii) $Cl_{0,1}(\mathbb{R})$ which is a two-dimensional algebra with vector space $G^1$ spanned by $\{1, e_1\}$ where the basis vector $e_1$ squares to $-1$, and is therefore isomorphic to $\mathbb{C}$, the field of complex numbers; (iii) $Cl_{0,2}(\mathbb{R})$ which is a 4-dimensional algebra with vector space $G^2$ spanned by $\{1, e_1, e_2, e_1e_2\}$, where $e_1, e_2, e_1e_2$ all square to $-1$ and anti-commute. Thus, $Cl_{0,2}(\mathbb{R})$ is isomorphic to the quaternions $\mathbb{H}$.

**Grade, dual, geometric product.** The *grade* of a Clifford algebra basis element is the dimension of the subspace it represents. For example, the basis elements $\{1, e_1, e_2, e_1e_2\}$ of the vector space $G^2$ of the Clifford algebra $Cl_{2,0}(\mathbb{R})$ have the grades $\{0, 1, 1, 2\}$. Using the concept of grades, we can divide Clifford algebras into linear subspaces made up of elements of each grade. The grade subspace of smallest dimension is $M_0$, the subspace of all scalars (elements with 0 basis vectors of the generating vector space). Elements of $M_1$ are called vectors, elements of $M_2$ are bivectors, and so on. In general, a vector space $G^{p+q}$ of a Clifford algebra $Cl_{p,q}(\mathbb{R})$ can be written as the direct sum of all of these subspaces: $G^{p+q} = M_0 \oplus M_1 \oplus \ldots \oplus M_{p+q}$. The elements of a Clifford algebra are called *multivectors*, containing elements of subspaces, i.e. scalars, vectors, bivectors, $\ldots$, $k$-vectors. The basis element with the highest grade is called the *pseudoscalar*[2], which in $\mathbb{R}^2$ corresponds to the bivector $e_1e_2$, and in $\mathbb{R}^3$ to the trivector $e_1e_2e_3$.

The *dual* $\boldsymbol{a}^*$ of a multivector $\boldsymbol{a}$ is defined as $\boldsymbol{a}^* = \boldsymbol{a}i_{p+q}$, where $i_{p+q}$ represents the respective pseudoscalar of the Clifford algebra. This definition allows us to relate different multivectors to each other, which is a useful property when defining Clifford Fourier transforms. For example, for Clifford algebras in $\mathbb{R}^2$ the dual of the scalar is the bivector, and in $\mathbb{R}^3$, the dual of the scalar is the trivector. Finally, the *geometric product* is a bilinear operation on multivectors. For arbitrary multivectors $\boldsymbol{a}, \boldsymbol{b}, \boldsymbol{c} \in G^{p+q}$, and scalar $\lambda$, the geometric product has the following properties: (i) closure, i.e. $\boldsymbol{ab} \in G^{p+q}$, (ii) associativity, i.e. $(\boldsymbol{ab})\boldsymbol{c} = \boldsymbol{a}(\boldsymbol{bc})$; (iii) commutative scalar multiplication, i.e. $\lambda\boldsymbol{a} = \boldsymbol{a}\lambda$; (iv) distributive over addition, i.e. $\boldsymbol{a}(\boldsymbol{b} + \boldsymbol{c}) = \boldsymbol{ab} + \boldsymbol{ac}$. The geometric product is in general non-commutative, i.e. $\boldsymbol{ab} \neq \boldsymbol{ba}$. Note that Equation 1 describe the geometric product specifically between basis elements of the generating vector space.

**Clifford algebras $Cl_{2,0}(\mathbb{R})$ and $Cl_{0,2}(\mathbb{R})$.** The 4-dimensional vector spaces of these Clifford algebras have the basis vectors $\{1, e_1, e_2, e_1e_2\}$ where $e_1, e_2$ square to $+1$ for $Cl_{2,0}(\mathbb{R})$ and to $-1$ for $Cl_{0,2}(\mathbb{R})$. For $Cl_{2,0}(\mathbb{R})$, the geometric product of two multivectors $\boldsymbol{a} = a_0 + a_1e_1 + a_2e_2 + a_{12}e_1e_2$ and $\boldsymbol{b} = b_0 + b_1e_1 + b_2e_2 + b_{12}e_1e_2$ is given by:

$$\begin{aligned} \boldsymbol{ab} = {} & (a_0b_0 + a_1b_1 + a_2b_2 - a_{12}b_{12})1 + (a_0b_1 + a_1b_0 - a_2b_{12} + a_{12}b_2)e_1 \\ & + (a_0b_2 + a_1b_{12} + a_2b_0 - a_{12}b_1)e_2 + (a_0b_{12} + a_1b_2 - a_2b_1 + a_{12}b_0)e_1e_2 , \end{aligned} \quad (2)$$

which can be derived by collecting terms that multiply the same basis elements, see Appendix A.

A vector $x = (x_1, x_2) \in \mathbb{R}^2$ with standard Euclidean product $\langle .,. \rangle$ can be related to $x_1e_1 + x_2e_2 \in \mathbb{R}^2 \subset G^2$. Clifford multiplication of two vectors $x, y \in \mathbb{R}^2 \subset G^2$ yields the geometric product $xy$:

$$\begin{aligned} xy &= (x_1e_1 + x_2e_2)(y_1e_1 + y_2e_2) \\ &= x_1y_1e_1^2 + x_2y_2e_2^2 + x_1y_2e_1e_2 + x_2y_1e_2e_1 \\ &= \langle x, y \rangle + x \wedge y , \end{aligned} \quad (3)$$

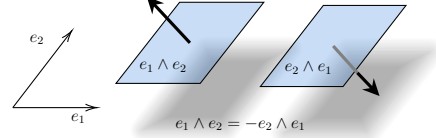

Figure 3: Antisymmetry of bivector exterior (wedge) product.

---

[1] Operations of addition and multiplication are associative.

[2] In contrast to scalars, pseudoscalars change sign under reflections.

where $\wedge$ is the exterior or wedge product. The asymmetric quantity $x \wedge y = -y \wedge x$ is associated with the *bivector*, which can be interpreted as an oriented plane segment as shown in Figure 3. A unit bivector $i_2$, spanned by the (orthonormal) basis vectors $e_1$ and $e_2$ is determined by the product:

$$i_2 = e_1 e_2 = \underbrace{\langle e_1, e_2 \rangle}_{=0} + e_1 \wedge e_2 = -e_2 \wedge e_1 = -e_2 e_1 \,, \tag{4}$$

which if squared yields $i_2^2 = -1$. Thus, $i_2$ represents a geometric $\sqrt{-1}$. From Equation 4, it follows that $e_2 = e_1 i_2 = -i_2 e_1$ and $e_1 = i_2 e_2 = -e_2 i_2$ . Using the pseudoscalar $i_2$, the dual of a scalar is a bivector and the dual of a vector is again a vector. The dual pairs of the base vectors are $1 \leftrightarrow e_1 e_2$ and $e_1 \leftrightarrow e_2$. For $Cl_{2,0}(\mathbb{R})$, these dual pairs allow us to write an arbitrary multivector $\boldsymbol{a}$ as

$$\boldsymbol{a} = a_0 + a_1 e_1 + a_2 e_2 + a_{12} e_1 e_2 = 1 \underbrace{(a_0 + a_{12} i_2)}_{\text{spinor part}} + e_1 \underbrace{(a_1 + a_2 i_2)}_{\text{vector part}} \,, \tag{5}$$

which can be regarded as two complex-valued parts: the spinor[3] part, which commutes with the base element 1, i.e. $1 i_2 = i_2 1$, and the vector part, which anti-commutes with the respective base element $\boldsymbol{e}_1$, i.e. $e_1 i_2 = e_1 e_1 e_2 = -e_1 e_2 e_1 = -i_2 e_1$. For $Cl(0, 2)(\mathbb{R})$, the vector part changes to $e_1 (a_1 - a_2 i_2)$. This decomposition will be the basis for Clifford Fourier transforms.

The Clifford algebra $Cl_{0,2}(\mathbb{R})$ is isomorphic to the quaternions $\mathbb{H}$, which are an extension of complex numbers and are commonly written in the literature as $a + b\hat{\imath} + c\hat{\jmath} + d\hat{k}$. Quaternions also form a 4-dimensional algebra spanned by $\{1, \hat{\imath}, \hat{\jmath}, \hat{k}\}$, where $\hat{\imath}, \hat{\jmath}, \hat{k}$ all square to $-1$. The algebra isomorphism to $Cl_{0,2}(\mathbb{R})$ is easy to verify since $e_1, e_2, e_1 e_2$ all square to $-1$ and anti-commute. The basis element 1 is called the scalar and the basis elements $\hat{\imath}, \hat{\jmath}, \hat{k}$ are called the vector part of a quaternion. Quaternions have practical uses in applied mathematics, particularly for expressing rotations, which we will use to define the rotational Clifford convolution layer in Section 3.

**Clifford algebra $Cl_{3,0}(\mathbb{R})$.** The 8-dimensional vector space $G^3$ of the Clifford algebra $Cl_{3,0}(\mathbb{R})$ has the basis vectors $\{1, e_1, e_2, e_3, e_1 e_2, e_3 e_1, e_2 e_3, e_1 e_2 e_3\}$, i.e. it consists of one scalar, three vectors $\{e_1, e_2, e_3\}$, three bivectors $\{e_1 e_2, e_3 e_1, e_2 e_3\}$[4], and one trivector $e_1 e_2 e_3$. The trivector is the pseudoscalar $i_3$ of the algebra. The geometric product of two multivectors is defined analogously to the geometric product of $Cl_{2,0}(\mathbb{R})$, see Appendix A. The dual pairs of $Cl_{3,0}(\mathbb{R})$ are: $1 \leftrightarrow e_1 e_2 e_3 = i_3$, $e_1 \leftrightarrow e_2 e_3$, $e_2 \leftrightarrow e_3 e_1$, and $e_3 \leftrightarrow e_1 e_2$. An intriguing example of the duality of the multivectors of $Cl_{3,0}(\mathbb{R})$ emerges when writing the expression of the electromagnetic field $\boldsymbol{F}$ in terms of an electric vector field $E$ and a magnetic vector field $B$, such that $\boldsymbol{F} = E + B i_3$ , where $E = E_x e_1 + E_y e_2 + E_z e_3$ and $B = B_x e_1 + B_y e_2 + B_z e_3$. In this way the electromagnetic field $\boldsymbol{F}$ decomposes into electric vector and magnetic bivector parts via the pseudoscalar $i_3$ (Hestenes, 2003). For example, for the base component $B_x e_1$ of $B$ it holds that $B_x e_1 i_3 = B_x e_1 e_1 e_2 e_3 = B_x e_2 e_3$ which is a bivector and the dual to the base component $E_x e_1$ of $E$. Consequently, the multivector representing $\boldsymbol{F}$ consists of three vectors (the electric field components) and three bivectors (the magnetic field components multiplied by $i_3$). This viewpoint gives Clifford neural layers a natural advantage over their default counterparts as we will see in Section 4.

## 3 CLIFFORD NEURAL LAYERS

Here, we introduce 2D Clifford convolution and 2D Clifford Fourier transform layers. Appendix B contains extensions to 3 dimensions. In Appendices B, D, related literature is discussed, most notably complex (Bassey et al., 2021) and quaternion neural networks (Parcollet et al., 2020).

**Clifford CNN layers.** Regular convolutional neural network (CNN) layers take as input feature maps $f : \mathbb{Z}^2 \to \mathbb{R}^{c_{\text{in}}}$ and convolve[5] them with a set of $c_{\text{out}}$ filters $\{w^i\}_{i=1}^{c_{\text{out}}}$ with $w^i : \mathbb{Z}^2 \to \mathbb{R}^{c_{\text{in}}}$:

$$[f \star w^i](x) = \sum_{y \in \mathbb{Z}^2} \langle f(y), w^i(y - x) \rangle = \sum_{y \in \mathbb{Z}^2} \sum_{j=1}^{c_{\text{in}}} f^j(y) w^{i,j}(y - x) \,, \tag{6}$$

---

[3]Spinors are elements of a complex vector space that can be associated with Euclidean space. Unlike vectors, spinors transform to their negative when rotated $360°$.

[4]The bivector $e_1 e_3$ has negative orientation.

[5]In deep learning, a convolution operation in the forward pass is implemented as cross-correlation.

which can be interpreted as an inner product of input feature maps with the corresponding filters at every point $y \in \mathbb{Z}^2$. By applying $c_{\text{out}}$ filters, the output feature maps can be interpreted as $c_{\text{out}}$ dimensional feature vectors at every point $y \in \mathbb{Z}^2$. We now extend CNN layers such that the element-wise product of scalars $f^j(y)w^{i,j}(y-x)$ is replaced by the geometric product of multivector inputs and multivector filters $\boldsymbol{f}^j(y)\boldsymbol{w}^{i,j}(y-x)$, where the chosen signature of $Cl$ is reflected in the geometric product. We replace the feature maps $f : \mathbb{Z}^2 \to \mathbb{R}^{c_{\text{in}}}$ by multivector feature maps $\boldsymbol{f} : \mathbb{Z}^2 \to (G^2)^{c_{\text{in}}}$ and convolve them with a set of $c_{\text{out}}$ multivector filters $\{\boldsymbol{w}^i\}_{i=1}^{c_{\text{out}}} : \mathbb{Z}^2 \to (G^2)^{c_{\text{in}}}$:

$$\left[\boldsymbol{f} \star \boldsymbol{w}^i\right](x) = \sum_{y \in \mathbb{Z}^2} \sum_{j=1}^{c_{\text{in}}} \underbrace{\boldsymbol{f}^j(y)\boldsymbol{w}^{i,j}(y-x)}_{\boldsymbol{f}^j \boldsymbol{w}^{i,j} \, : \, G^2 \times G^2 \to G^2} . \tag{7}$$

Note that each geometric product, indexed by $i \in \{1, ..., c_{\text{out}}\}$ and $j \in \{1, ..., c_{\text{in}}\}$, now results in a new multivector rather than a scalar. Hence, the output of a layer is a grid of $c_{\text{out}}$ multivectors. We can e.g. implement a $Cl(2,0)(\mathbb{R})$ Clifford CNN layer using Equation 2 where $\{b_0, b_1, b_2, b_{12}\} \to \{w_0^{i,j}, w_1^{i,j}, w_2^{i,j}, w_{12}^{i,j}\}$ correspond to 4 different kernels representing one 2D multivector kernel, i.e. 4 different convolution layers, and $\{a_0, a_1, a_2, a_{12}\} \to \{f_0^j, f_1^j, f_2^j, f_{12}^j\}$ correspond to the scalar, vector and bivector parts of the

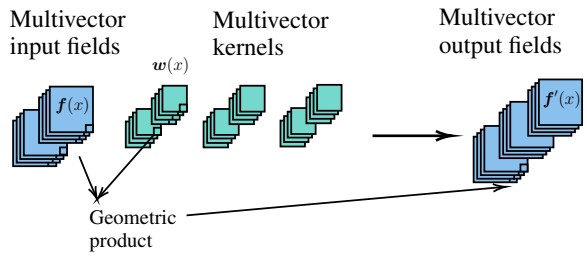

Figure 4: Sketch of Clifford convolution. Multivector input fields are convolved with multivector kernels.

input multivector field. The channels of the different layers represent different stacks of scalars, vectors, and bivectors. Analogously, we can implement a $Cl(3,0)(\mathbb{R})$ CNN layer using Equation 42 in Appendix B. A schematic sketch of a Clifford convolution layer is shown in Figure 4.

**Rotational Clifford CNN layers.** Here we introduce an alternative parameterization to the Clifford CNN layer introduced in Equation 7 by using the isomorphism of the Clifford algebra $Cl_{0,2}(\mathbb{R})$ to quaternions. We take advantage of the fact that a quaternion rotation can be realized by a matrix multiplication (Jia, 2008; Kuipers, 1999; Schwichtenberg, 2015). Using the isomorphism, we can represent the feature maps $\boldsymbol{f}^j$ and filters $\boldsymbol{w}^{i,j}$ as quaternions: $\boldsymbol{f}^j = f_0^j + f_1^j \hat{\imath} + f_2^j \hat{\jmath} + f_3^j \hat{k}$ and $\boldsymbol{w}^{i,j} = w_0^{i,j} + w_1^{i,j}\hat{\imath} + w_2^{i,j}\hat{\jmath} + w_3^{i,j}\hat{k}$[6]. We can now devise an alternative parameterization of the product between the feature map $\boldsymbol{f}^j$ and $\boldsymbol{w}^{i,j}$. To be more precise, we introduce a composite operation that results in a scalar quantity and a quaternion rotation, where the latter acts on the vector part of the quaternion $\boldsymbol{f}^j$ and only produces nonzero expansion coefficients for the vector part of the quaternion output. A quaternion rotation $\boldsymbol{w}^{i,j}\boldsymbol{f}^j(\boldsymbol{w}^{i,j})^{-1}$ acts on the vector part $(\hat{\imath}, \hat{\jmath}, \hat{k})$ of $\boldsymbol{f}^j$, and can be algebraically manipulated into a vector-matrix operation $\boldsymbol{R}^{i,j}\boldsymbol{f}^j$, where $\boldsymbol{R}^{i,j} : \mathbb{H} \to \mathbb{H}$ is built up from the elements of $\boldsymbol{w}^{i,j}$ (Kuipers, 1999). In other words, one can transform the vector part $(\hat{\imath}, \hat{\jmath}, \hat{k})$ of $\boldsymbol{f}^j \in \mathbb{H}$ via a rotation matrix $\boldsymbol{R}^{i,j}$ that is built from the scalar and vector part $(1, \hat{\imath}, \hat{\jmath}, \hat{k})$ of $\boldsymbol{w}^{i,j} \in \mathbb{H}$. Altogether, a rotational multivector filter $\{\boldsymbol{w}_{\text{rot}}^i\}_{i=1}^{c_{\text{out}}} : \mathbb{Z}^2 \to (G^2)^{c_{\text{in}}}$ acts on the feature map $\boldsymbol{f}^j$ through a rotational transformation $\boldsymbol{R}^{i,j}(w_{\text{rot},0}^{i,j}, w_{\text{rot},1}^{i,j}, w_{\text{rot},2}^{i,j}, w_{\text{rot},12}^{i,j})$ acting on vector and bivector parts of the multivector feature map $\boldsymbol{f} : \mathbb{Z}^2 \to (G^2)^{c_{\text{in}}}$, and an additional scalar response of the multivector filters:

$$\left[\boldsymbol{f} \star \boldsymbol{w}_{\text{rot}}^i\right](x) = \sum_{y \in \mathbb{Z}^2} \sum_{j=1}^{c_{\text{in}}} \underbrace{\left[\boldsymbol{f}^j(y)\boldsymbol{w}_{\text{rot}}^{i,j}(y-x))\right]_0}_{\text{scalar output}} + \boldsymbol{R}^{i,j}(y-x) \cdot \begin{pmatrix} f_1^j(y) \\ f_2^j(y) \\ f_{12}^j(y) \end{pmatrix} , \tag{8}$$

where $\left[\boldsymbol{f}^j(y)\boldsymbol{w}_{\text{rot}}^{i,j}(y-x))\right]_0 = f_0^j w_{\text{rot},0}^{i,j} - f_1^j w_{\text{rot},1}^{i,j} - f_2^j w_{\text{rot},2}^{i,j} - f_{12}^j w_{\text{rot},12}^{i,j}$, i.e., the scalar output of the geometric product of $Cl_{0,2}(\mathbb{R})$ as in Equation 34. A detailed description of the rotational multivector filters $\boldsymbol{R}^{i,j}(y-x)$ is outlined in Appendix B. While in principle the Clifford CNN layer in Equation 7 and the rotational Clifford CNN layer in Equation 8 are equally flexible, our experiments in Section 4 show that rotational Clifford CNN layers lead to better performance.

---

[6]Note that the expansion coefficients for the feature map $\boldsymbol{f}^j$ and filters $\boldsymbol{w}^{i,j}$ in terms of the basis elements of $G^2$ and in terms of quaternion elements $\hat{\imath}, \hat{\jmath}$ and $\hat{k}$ are the same.

Clifford convolutions satisfy the property of equivariance under translation of the multivector inputs, as shown in theorem 1 in Appendix B. Analogous to Theorem 1, translation equivariance can be derived for rotational Clifford CNN layers.

**Clifford Fourier layers.** The discrete Fourier transform of an $n$-dimensional complex signal $f(x) = f(x_1, \ldots, x_n) : \mathbb{R}^n \to \mathbb{C}$ at $M_1 \times \ldots \times M_n$ grid points is defined as:

$$\mathcal{F}\{f\}(\xi_1, \ldots, \xi_n) = \sum_{m_1=0}^{M_1} \ldots \sum_{m_n=0}^{M_n} f(m_1, \ldots, m_n) \cdot e^{-2\pi i \cdot \left(\frac{m_1 \xi_1}{M_1} + \ldots + \frac{m_n \xi_n}{M_n}\right)} , \qquad (9)$$

where $(\xi_1, \ldots, \xi_n) \in \mathbb{Z}_{M_1} \ldots \times \ldots \mathbb{Z}_{M_n}$. In Fourier Neural Operators (FNO) (Li et al., 2020), discrete Fourier transforms on real-valued input fields and respective back-transforms – implemented as Fast Fourier Transforms on real-valued inputs (RFFTs)[7] – are interleaved with a weight multiplication by a complex weight matrix of shape $c_{\text{in}} \times c_{\text{out}}$ for each mode, which results in a complex-valued weight tensor of the form $W \in \mathbb{C}^{c_{\text{in}} \times c_{\text{out}} \times (\xi_1^{\max} \times \ldots \times \xi_n^{\max})}$, where Fourier modes above cut-off frequencies $(\xi_1^{\max}, \ldots, \xi_n^{\max})$ are set to zero. Additionally, a residual connection is usually implemented as convolution layer with kernel size 1. In Figure 5a, a sketch of an FNO layer is shown. For $Cl(2,0)(\mathbb{R})$, the Clifford Fourier transform (Ebling & Scheuermann, 2005; Ebling, 2006; Hitzer, 2012) for multivector valued functions $\boldsymbol{f}(x) : \mathbb{R}^2 \to G^2$ and vectors $x, \xi \in \mathbb{R}^2$ is defined as:

$$\hat{\boldsymbol{f}}(\xi) = \mathcal{F}\{\boldsymbol{f}\}(\xi) = \frac{1}{2\pi} \int_{\mathbb{R}_2} \boldsymbol{f}(x) e^{-2\pi i_2 \langle x, \xi \rangle} \, dx , \; \forall \xi \in \mathbb{R}^2 , \qquad (10)$$

provided that the integral exists. In contrast to standard Fourier transforms, $\boldsymbol{f}(x)$ and $\hat{\boldsymbol{f}}(\xi)$ represent multivector fields in the spatial and the frequency domain, respectively. Furthermore, $i_2 = e_1 e_2$ is used in the exponent. Inserting the definition of multivector fields, we can rewrite Equation 10 as:

$$\mathcal{F}\{\boldsymbol{f}\}(\xi) = \frac{1}{2\pi} \int_{\mathbb{R}_2} \left[ 1\left( \underbrace{f_0(x) + f_{12}(x)i_2}_{\text{spinor part}} \right) + e_1\left( \underbrace{f_1(x) + f_2(x)i_2}_{\text{vector part}} \right) \right] e^{-2\pi i_2 \langle x, \xi \rangle} \, dx$$

$$= 1\left[ \mathcal{F}\left( f_0(x) + f_{12}(x)i_2 \right)(\xi) \right] + e_1\left[ \mathcal{F}\left( f_1(x) + f_2(x)i_2 \right)(\xi) \right]. \qquad (11)$$

We obtain a Clifford Fourier transform by applying two standard Fourier transforms to the dual pairs $\boldsymbol{f}_0 = f_0(x) + f_{12}(x)i_2$ and $\boldsymbol{f}_1 = f_1(x) + f_2(x)i_2$, which both can be treated as a complex-valued signals $\boldsymbol{f}_0, \boldsymbol{f}_1 : \mathbb{R}^2 \to \mathbb{C}$. Consequently, $\boldsymbol{f}(x)$ can be understood as an element of $\mathbb{C}^2$. The 2D Clifford Fourier transform is the linear combination of two classical Fourier transforms. Discrete versions of Equation 11 are obtained analogously to Equation 9, see Appendix B. Similar to FNO, multivector weight tensors $\boldsymbol{W} \in (G^2)^{c_{\text{in}} \times c_{\text{out}} \times (\xi_1^{\max} \times \xi_2^{\max})}$ are applied, where again Fourier modes above cut-off frequencies $(\xi_1^{\max}, \xi_2^{\max})$ are set to zero. In doing so, we point-wise modify the Clifford Fourier modes $\hat{\boldsymbol{f}}(\xi) = \mathcal{F}\{\boldsymbol{f}\}(\xi) = \hat{f}_0(\xi) + \hat{f}_1(\xi)e_1 + \hat{f}_2(\xi)e_2 + \hat{f}_{12}(\xi)e_{12}$ via the geometric product. The Clifford Fourier modes follow naturally when combining spinor and vector parts of Equation 11. Finally, the residual connection is replaced by a Clifford convolution with multivector kernel $\boldsymbol{k}$. A schematic sketch is shown in Figure 5b. For $Cl(3,0)(\mathbb{R})$, Clifford Fourier transforms follow a similar elegant construction, where we apply four separate Fourier transforms to

$$\begin{aligned} \boldsymbol{f}_0(x) &= f_0(x) + f_{123}(x)i_3 & \boldsymbol{f}_1(x) &= f_1(x) + f_{23}(x)i_3 \\ \boldsymbol{f}_2(x) &= f_2(x) + f_{31}(x)i_3 & \boldsymbol{f}_3(x) &= f_3(x) + f_{12}(x)i_3 , \end{aligned} \qquad (12)$$

i.e. scalar/trivector and vector/bivector components are combined into complex fields and then subjected to a Fourier transform.

## 4 EXPERIMENTS

We assess Clifford neural layers for different architectures in three experimental settings: the incompressible Navier-Stokes equations, shallow water equations for weather modeling, and 3-dimensional Maxwell's equations. We replace carefully designed baseline architectures by their

---

[7]The FFT of a real-valued signal is Hermitian-symmetric, so the output contains only the positive frequencies below the Nyquist frequency for the last spatial dimension.

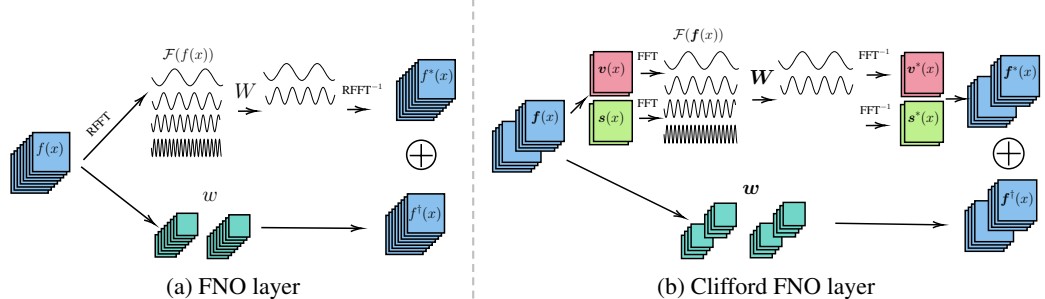

(a) FNO layer              (b) Clifford FNO layer

Figure 5: Sketch of Fourier Neural Operator (FNO) and Clifford Fourier Operator (CFNO) layers. The real valued Fast Fourier transform (RFFT) over real valued scalar input fields $f(x)$ is replaced by the complex Fast Fourier transform (FFT) over the complex valued dual parts $\boldsymbol{v}(x)$ and $\boldsymbol{s}(x)$ of multivector fields $\boldsymbol{f}(x)$. Pointwise multiplication in the Fourier space via complex weight tensor $W$ is replaced by the geometric product in the Clifford Fourier space via multivector weight tensor $\boldsymbol{W}$. Additionally, the convolution path is replaced by Clifford convolutions with multivector kernels $\boldsymbol{w}$.

Clifford counterparts. Baseline ResNet architectures comprise 8 residual blocks, each consisting of two convolution layers with $3 \times 3$ kernels, shortcut connections, group normalization (Wu & He, 2018), and GeLU activation functions (Hendrycks & Gimpel, 2016). Baseline 2-dimensional Fourier Neural Operators (FNOs) consist of 8 (4) FNO blocks, GeLU activations and no normalization scheme, using 16 (8) Fourier modes for the 2- and 3-dimensional equations, respectively. For Clifford networks, we change convolutions and Fourier transforms to their respective Clifford operation, and substitute normalization techniques and activation functions with Clifford counterparts, keeping the number of parameters similar. We evaluate different training set sizes, and report losses for scalar and vector fields. All datasets share the common trait of containing multiple input and output fields. More precisely, one scalar and one 2-dimensional vector field in case of the Navier-Stokes and the shallow water equations, and a 3-dimensional (electric) vector field and its dual (magnetic) bivector field in case of the Maxwell's equations.

Example inputs and targets of the neural PDE surrogates are shown in Figure 6. The number of input timesteps $t$ vary for different experiments. The *one-step loss* is the mean-squared error at the next timestep summed over fields. The *rollout loss* is the mean-squared error after applying the neural PDE surrogate 5 times, summing over fields and time dimension. More information on the implementation details of the tested architectures, loss functions, and more detailed results can be found in Appendix C.

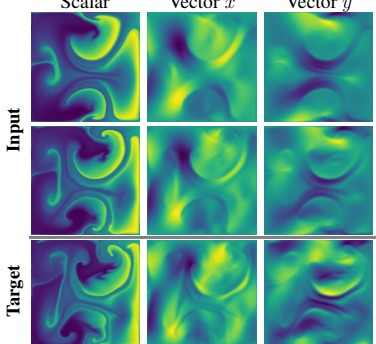

Figure 6: Example input and target fields for the Navier-Stokes experiments. Input fields comprise a $t = 2$ timestep history.

**Navier-Stokes in 2D.** The incompressible Navier-Stokes equations (Temam, 2001) conserve the velocity flow fields $v : \mathcal{X} \to \mathbb{R}^2$ where $\mathcal{X} \in \mathbb{R}^2$ via:

$$\frac{\partial v}{\partial t} = -v \cdot \nabla v + \mu \nabla^2 v - \nabla p + f , \quad \nabla \cdot v = 0 , \quad (13)$$

where $v \cdot \nabla v$ is the convection, i.e. the rate of change of $v$ along $v$, $\mu \nabla^2 v$ the viscosity, i.e. the diffusion or net movement of $v$, $\nabla p$ the internal pressure and $f$ an external force, which in our case is a buoyancy force. An additional incompressibility constraint $\nabla \cdot v = 0$ yields mass conservation of the Navier-Stokes equations. In addition to the velocity field, we introduce a scalar field representing a scalar quantity, i.e. smoke, that is being transported via the velocity field. The scalar field is *advected* by the vector field, i.e. as the vector field changes, the scalar field is transported along with it, whereas the scalar field influences the vector field only via an external force term. We call this **weak coupling** between vector and scalar fields. We implement the 2D Navier-Stokes equation using $\Phi\texttt{Flow}$[8](Holl et al., 2020), obtaining data on a grid with spatial resolution of $128 \times 128$ ($\Delta x = 0.25$, $\Delta y = 0.25$),

---

[8] https://github.com/tum-pbs/PhiFlow

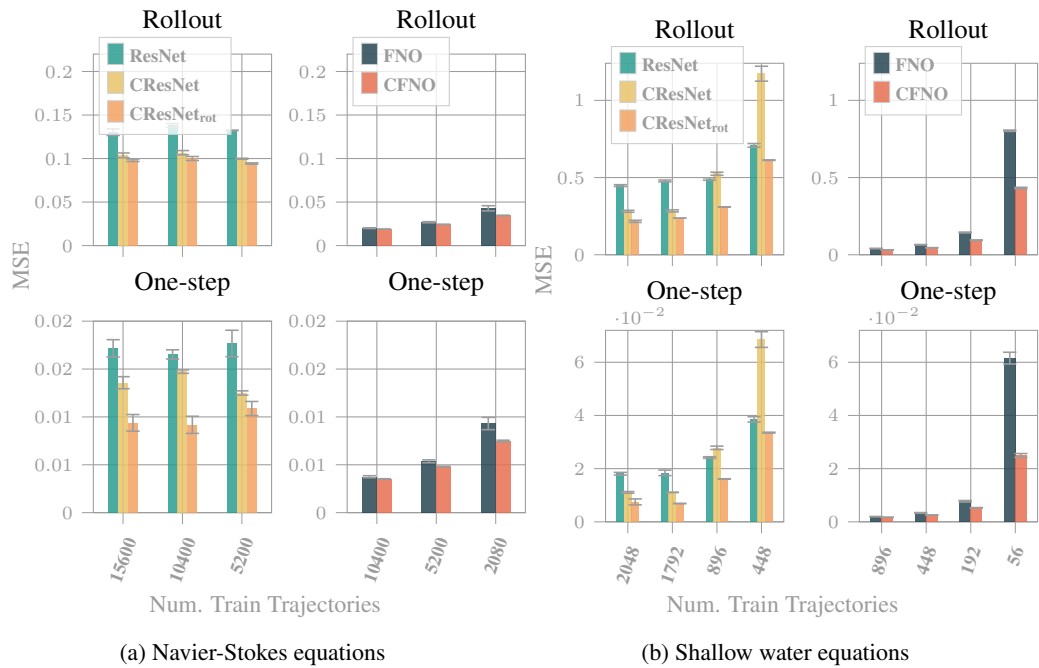

(a) Navier-Stokes equations

(b) Shallow water equations

Figure 7: Results for ResNet based (left) and Fourier based (right) architectures on the 2-dimensional Navier-Stokes and Shallow water experiments. One-step and rollout loss are shown.

and temporal resolution of $\Delta t = 1.5$ s. Results for one-step loss and rollout loss on the test set are shown in Figure 7a. For ResNet-like architectures, we observe that both CResNet and CResNet$_{\text{rot}}$ improve upon the ResNet baseline. Additionally, we observe that rollout losses are also lower for the two Clifford based architectures, which we attribute to better and more stable models that do not overfit to one-step predictions so easily. Lastly, while in principle CResNet and CResNet$_{\text{rot}}$ based architectures are equally flexible, CResNet$_{\text{rot}}$ ones in general perform better than CResNet ones. For FNO and respective Clifford Fourier based (CFNO) architectures, the loss is in general much lower than for ResNet based architectures. CFNO architectures improve upon FNO architectures for all dataset sizes, and for one-step as well as rollout losses.

**Shallow water equations.** This set of coupled equations (Vreugdenhil, 1994) can be derived from integrating the incompressible Navier–Stokes equations, in cases where the horizontal length scale is much larger than the vertical length scale. As such, the equations model a thin layer of fluid of constant density in hydrostatic balance, bounded from below by the bottom topography and from above by a free surface via 3 coupled PDEs, describing the velocity in $x$- direction, the velocity in the $y$- direction, and the scalar pressure field. The shallow water equations can be therefore be used as simplified weather model, as done in this work and exemplified in Figure 1. The relation between vector and scalar components is relatively strong (**strong coupling** due to the 3-coupled PDEs). We obtain data for the 2D shallow water equations on a grid with spatial resolution of $192 \times 96$ ($\Delta x = 1.875°$, $\Delta y = 3.75°$), and temporal resolution of $\Delta t = 6$ h. We observe similar results than for the Navier-Stokes experiments. For low number of trajectories, ResNet architectures seem to lack expressiveness, where arguably some data smoothing is learned first. Thus, ResNets need significantly more trajectories compared to (C)FNO architectures to obtain reasonable loss values, which seems to go hand in hand with Clifford layers gaining advantage. In general, performance differences between baseline and Clifford architectures are even more pronounced, which we attribute to the stronger coupling of the scalar and the vector fields.

**Maxwell's equations in matter in 3D.** In isotropic media, Maxwell's equations (Griffiths, 2005) propagate solutions of the displacement field $D$, which is related to the electrical field via $D = \epsilon_0 \epsilon_r E$, where $\epsilon_0$ is the permittivity of free space and $\epsilon_r$ is the permittivity of the medium, and the magnetization field $H$, which is related to the magnetic field $B$ via $H = \mu_0 \mu_r B$, where $\mu_0$ is the permeability of free space and $\mu_r$ is the permeability of the medium. The electromagnetic field $\boldsymbol{F}$ has the intriguing property that the electric field $E$ and the magnetic field $B$ are dual pairs, thus

$\boldsymbol{F} = E + Bi_3$, i.e. **strong coupling** between the electric field and its dual (bivector) magnetic field. This duality also holds for $D$ and $H$. Concretely, the fields of interest are the vector-valued $D$-field $(D_x, D_y, D_z)$ and the vector-valued $H$-field $(H_x, H_y, H_z)$. We obtain data for the 3D Maxwell's equations on a grid with spatial resolution of $32 \times 32 \times 32$ ($\Delta x = \Delta y = \Delta z = 5 \cdot 10^{-7} m$), and temporal resolution of $\Delta t = 50\,\text{s}$. We randomly place 18 different light sources outside a cube which emit light with different amplitude and different phase shifts, causing the resulting $D$ and $H$ fields to interfere. The wavelength of the emitted light is $10^{-5} m$.

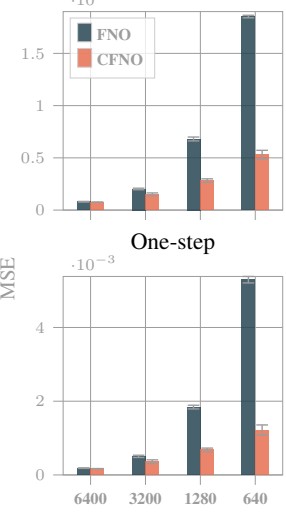

Figure 8: Results for Fourier based architectures on Maxwell equation's.

We test FNO based architectures and respective Clifford counterparts (CFNO). Due to the vector-bivector character of electric and magnetic field components, Maxwell's equations are an ideal playground to stress-test the inductive bias advantages of Clifford base architectures. Results for one-step loss and rollout loss on the test set are shown in Figure 8. CFNO architectures improve upon FNO architectures, especially for low numbers of trajectories. Results demonstrate the much stronger inductive bias of Clifford based 3-dimensional Fourier layers, and their general applicability to 3-dimensional problems, which are structurally even more interesting than 2-dimensional ones.

## 5 CONCLUSION

We introduced Clifford neural layers that handle the various scalar (e.g. charge density), vector (e.g. electric field), bivector (magnetic field) and higher order fields as proper geometric objects organized as multivectors. This geometric algebra perspective allowed us to naturally generalize convolution and Fourier transformations to their Clifford counterparts, providing an elegant rule to design new neural network layers. The multivector viewpoint denotes an inductive bias advantage, leading to a better representation of the relationship between fields and their individual components, which is prominently demonstrated by the fact that our Clifford layers significantly outperformed equivalent standard neural PDE surrogates.

**Limitations.** One limitation is the current speed of Fast Fourier Transform (FFT) operations on machine learning accelerators like GPUs. While an active area of research, current available versions of cuFFT[9] kernels wrapped in PyTorch (Paszke et al., 2019) are not yet as heavily optimized[10], especially for the gradient pass. In contrast to FNO layers, which operate on real-valued signals, Clifford Fourier layers use complex-valued FFT operations where the backward pass is approximately twice as slow. For similar parameter counts, inference times of FNO and CFNO networks are similar. Similar to Grassucci et al. (2021) who investigated the speed of geometric convolution layers, we found that Clifford convolutions are more parameter efficient since they share parameters among filters, with the downside that the net number of operations is larger, resulting in increased training times by a factor of about 2. Finally, from a PDE point of view, the presented approaches to obtain PDE surrogates are limited since the neural networks have to be retrained for different equation parameters or e.g. different $\Delta t$.

**Future work.** Besides modeling of PDEs, weather, and fluid dynamics, we see potential applications of Clifford layers for e.g. MRI or radar data, and for neural implicit representations (Xie et al., 2022; Rella et al., 2022). Extensions towards graph networks and attention based models will be useful to explore. Furthermore, custom multivector GPU kernels can overcome many of the speed issues as the compute density of Clifford operations is much higher which is better for hardware accelerators (Hoffmann et al., 2020). The use of a just-in-time compiled language with better array abstractions like Julia (Bezanson et al., 2017) could significantly simplify the interface. Finally, combining the ideas of multivector modeling together with various physics-informed neural network approaches (Raissi et al., 2019; Lutter et al., 2018; Gupta et al., 2019; Cranmer et al., 2020; Zubov et al., 2021) is an attractive next step.

---

[9] https://developer.nvidia.com/cufft

[10] For alternative efficient GPU-accelerated multidimensional FFT libraries see e.g. https://github.com/DTolm/VkFFT

## REPRODUCIBILITY AND ETHICAL STATEMENT

**Reproducibility statement.** We have included error bars, and ablation studies wherever we found it necessary and appropriate. We have described our architectures in Section 4 and provided further implementation details in Appendix Section C. We have further include pseudocode for the newly proposed layers in Appendix Section B.6. We open-sourced our PyTorch implementation at https://microsoft.github.io/cliffordlayers/ for others to use. We aim to develop this codebase further in the future.

**Ethical statement.** Neural PDE surrogates will play an important role in modeling many natural phenomena, and thus developing them further might enable us to achieve shortcuts or alternatives for computationally expensive simulations. For example, if used as such, PDE surrogates will potentially help to advance different fields of research, especially in the natural sciences. Examples related to this paper are fluid dynamics or weather modeling. Therefore, PDE surrogates might potentially be directly or indirectly related to reducing the carbon footprint. On the downside, relying on simulations always requires rigorous cross-checks and monitoring, especially when we "learn to simulate".

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

# Appendices

CONTENTS

# A  MATHEMATICAL BACKGROUND

This appendix supports Section 2 of the main paper. We give a more detailed explanation of real Clifford algebras and have a closer look at $Cl_{2,0}(\mathbb{R})$, $Cl_{0,2}(\mathbb{R})$, and $Cl_{3,0}(\mathbb{R})$. For a detailed introduction into Clifford algebras we recommend Suter (2003); Hestenes (2003; 2012); Dorst et al. (2010); Renaud (2020)

## A.1  CLIFFORD ALGEBRAS

**Vector spaces and algebras over a field.**  A *vector space* over a field $F$ is a set $V$ together with two binary operations that satisfy the axioms for vector addition and scalar multiplication. The axioms of addition ensure that if two elements of $V$ get added together, we end up with another element of $V$. The elements of $F$ are called scalars. Examples of a field $F$ are the real numbers $\mathbb{R}$ and the complex numbers $\mathbb{C}$. Although it is common practice to refer to the elements of a general vector space $V$ as vectors, to avoid confusion we will reserve the usage of this term to the more specific case of elements of $\mathbb{R}^n$. As we will see below, general vector spaces can consist of more complicated, higher-order objects than scalars, vectors or matrices.

An *algebra over a field* consists of a vector space $V$ over a field $F$ together with an additional *bilinear* law of composition of elements of the vector space, $V \times V \to V$, that is, if $a$ and $b$ are any two elements of $V$, then $ab : V \times V \to V$ is an element of $V$, satisfying a pair of distribution laws: $a(\lambda_1 b + \lambda_2 c) = \lambda_1 ab + \lambda_2 ac$ and $(\lambda_1 a + \lambda_2 b)c = \lambda_1 ac + \lambda_2 bc$ for $\lambda_1, \lambda_2 \in F$ and $a, b, c \in V$. Note that general vector spaces don't have bilinear operations defined on their elements.

**Clifford algebras over $\mathbb{R}$.**  In this manuscript we will focus on Clifford algebras over $\mathbb{R}$. For a more general exposition on Clifford algebras over different fields the reader is referred to Lounesto (1986).

A real Clifford algebra is generated by the $n$-dimensional vector space $\mathbb{R}^n$ through a set of relations that hold for the basis elements of the vector space $\mathbb{R}^n$. Let us denote the basis elements of $\mathbb{R}^n$ with $e_1, ..., e_n$, and without loss of generality choose these basis elements to be mutually orthonormal. Taking two nonnegative integers $p$ and $q$, such that $p + q = n$, then a real Clifford algebra $Cl_{p,q}(\mathbb{R})$ with the "signature" $(p, q)$, is generated through the following relations that define how the bilinear product of the algebra operates on the basis elements of $\mathbb{R}^n$:

$$e_i^2 = +1 \qquad\qquad \text{for } 1 \leq i \leq p \,, \qquad\qquad (14)$$

$$e_j^2 = -1 \qquad\qquad \text{for } p < j \leq p + q \,, \qquad\qquad (15)$$

$$e_i e_j = -e_j e_i \qquad\qquad \text{for } i \neq j \,. \qquad\qquad (16)$$

Through these relations we can generate a basis for the vector space of the Clifford algebra, which we will denote with $G$. Equations 14 and 15 show that the product between two vectors yields a scalar. According to the aforementioned definition of an algebra over a field, a Clifford algebra with a vector space $G$ is equipped with a bilinear product $G \times G \mapsto G$, that combines two elements from the vector space $G$ and yields another element of the same space $G$. Therefore, both scalars and vectors must be elements of the vector space $G$. Equation 16 shows that besides scalar and vector elements, higher order elements consisting of a combination of two basis elements, such as $e_i e_j$ and $e_j e_i$, are also part of the vector space $G$. Finally, by combining Equations 14, 15, 16 we can create even higher order elements such as $e_i e_j e_k$ for $i \neq j \neq k$, or $e_1 e_2 ... e_{p+q}$, which all must be part of the vector space $G$.

In order to determine what the basis elements are that span the vector space $G$ of $Cl_{p,q}(\mathbb{R})$, we note that elements $e_{\sigma(1)} e_{\sigma(2)} ... e_{\sigma(k)}$ and $e_1 e_2 ... e_k$ are related through a simple scalar multiplicative factor of plus or minus one, depending on the sign of the permutation $\sigma$. Therefore, it suffices to consider the unordered combinations of basis elements of $\mathbb{R}^n$: the basis of the vector space $G$ is given by $\{1, e_1, e_2, ..., e_{p+q}, e_1 e_2, ..., e_{p+q-1} e_{p+q}, ..., e_1 e_2 ... e_{p+q}\}$.

In summary, we have introduced two different vector spaces. First, the vector space $\mathbb{R}^n$ which *generates* the Clifford algebra, and second the vector space $G$, which is the vector space spanned by the basis elements of the Clifford algebra $Cl_{p,q}(\mathbb{R})$. Convention is to denote the vector space of a real Clifford algebra with a superscript $n$ of the dimension of the generating vector space, yielding $G^n$ for a generating vector space $\mathbb{R}^n$. Note that the dimension of the vector space $G^n$ is $2^n = 2^{p+q}$.

Exemplary low-dimensional Clifford algebras are: (i) $Cl_{0,0}(\mathbb{R})$ which is a one-dimensional algebra that is spanned by the vector $\{1\}$ and is therefore isomorphic to $\mathbb{R}$, the field of real numbers; (ii) $Cl_{0,1}(\mathbb{R})$ which is a two-dimensional algebra with vector space $G^1$ spanned by $\{1, e_1\}$ where the basis vector $e_1$ squares to $-1$, and is therefore isomorphic to $\mathbb{C}$, the field of complex numbers; (iii) $Cl_{0,2}(\mathbb{R})$ which is a 4-dimensional algebra with vector space $G^2$ spanned by $\{1, e_1, e_2, e_1e_2\}$, where $e_1, e_2$ square to $-1$ and anti-commute. Thus, $Cl_{0,2}(\mathbb{R})$ is isomorphic to the quaternions $\mathbb{H}$.

---

**Definition 1: Grade of Clifford algebra element**

The grade of a Clifford algebra basis element is the dimension of the subspace it represents.

---

For example, the basis elements $\{1, e_1, e_2, e_1e_2\}$ of the Clifford algebras $Cl_{0,2}(\mathbb{R})$ and $Cl_{2,0}(\mathbb{R})$ have the grades $\{0, 1, 1, 2\}$. Using the concept of grades, we can divide the vector spaces of Clifford algebras into linear subspaces made up of elements of each grade. The grade subspace of smallest dimension is $M_0$, the subspace of all scalars (elements with 0 basis vectors). Elements of $M_1$ are called vectors, elements of $M_2$ are bivectors, and so on. In general, the vector space $G^{p+q}$ of a Clifford algebra $Cl_{p,q}$ can be written as the direct sum of all of these subspaces:

$$G^{p+q} = M_0 \oplus M_1 \oplus \ldots \oplus M_{p+q} . \tag{17}$$

The elements of a Clifford algebra are called *multivectors*, containing elements of subspaces, i.e. scalars, vectors, bivectors, trivectors etc. The basis element with the highest grade is called the *pseudoscalar*[11], which in $\mathbb{R}^2$ corresponds to the bivector $e_1e_2$, and in $\mathbb{R}^3$ to the trivector $e_1e_2e_3$. The pseudoscalar is often denoted with the symbol $i_{p+q}$. From hereon, only multivectors will be denoted with boldface symbols.

**Geometric product.** Using Equations 14, 15, 16, we have seen how basis elements of the vector space $G^{p+q}$ of the Clifford algebra are formed using basis elements of the generating vector space $V$. We now, look at how elements of $G^{p+q}$ are combined, i.e. how multivectors are bilinearly operated on. The *geometric product* is the bilinear operation on multivectors in Clifford algebras. For arbitrary multivectors $\boldsymbol{a}, \boldsymbol{b}, \boldsymbol{c} \in G^{p+q}$, and scalar $\lambda$ the geometric product has the following properties:

$$\boldsymbol{ab} \in G^{p+q} \qquad \text{closure ,} \tag{18}$$
$$(\boldsymbol{ab})\boldsymbol{c} = \boldsymbol{a}(\boldsymbol{bc}) \qquad \text{associativity ,} \tag{19}$$
$$\lambda\boldsymbol{a} = \boldsymbol{a}\lambda \qquad \text{commutative scalar multiplication ,} \tag{20}$$
$$\boldsymbol{a}(\boldsymbol{b} + \boldsymbol{c}) = \boldsymbol{ab} + \boldsymbol{ac} \qquad \text{distributive over addition .} \tag{21}$$

The geometric product is in general non-commutative, i.e. $\boldsymbol{ab} \neq \boldsymbol{ba}$. As we describe later, the geometric product is made up of two things: an inner product (that captures similarity) and exterior (wedge) product that captures difference.

---

**Definition 2: Dual of a multivector**

The dual $\boldsymbol{a}^*$ of a multivector $\boldsymbol{a}$ is defined as:

$$\boldsymbol{a}^* = \boldsymbol{a}i_{p+q} , \tag{22}$$

where $i_{p+q}$ represents the respective pseudoscalar of the Clifford algebra.

---

This definition allows us to relate different multivectors to each other, which is a useful property when defining Clifford Fourier transforms. For example, for Clifford algebras in $\mathbb{R}^2$ the dual of a scalar is a bivector, and for the Clifford algebra $\mathbb{R}^3$ the dual of a scalar is a trivector.

---

[11]In contrast to scalars, pseudoscalars change sign under reflection.

## A.2 EXAMPLES OF LOW-DIMENSIONAL CLIFFORD ALGEBRAS

### A.2.1 CLIFFORD ALGEBRA $Cl_{0,1}(\mathbb{R})$

The Clifford algebra $Cl_{0,1}(\mathbb{R})$ is a two-dimensional algebra with vector space $G^1$ spanned by $\{1, e_1\}$, and where the basis vector $e_1$ squares to $-1$. $Cl_{0,1}(\mathbb{R})$ is thus algebra-isomorphic to $\mathbb{C}$, the field of complex numbers. This becomes more obvious if we identify the basis element with the highest grade, i.e. $e_1$, as the pseudoscalar $i_1$ which is the imaginary part of the complex numbers. The geometric product between two multivectors $\boldsymbol{a} = a_0 + a_1 e_1$ and $\boldsymbol{b} = b_0 + b_1 e_1$ is therefore also isomorphic to the product of two complex numbers:

$$\begin{aligned} \boldsymbol{ab} &= a_0 b_0 + a_0 b_1 \; e_1 \; + a_1 b_0 \; e_1 \; + a_1 b_1 e_1 e_1 \\ &= (a_0 b_0 - a_1 b_1) + (a_0 b_1 + a_1 b_0) \; e_1 \; . \end{aligned} \tag{23}$$

### A.2.2 CLIFFORD ALGEBRA $Cl_{2,0}(\mathbb{R})$

The Clifford algebra $Cl_{2,0}(\mathbb{R})$ is a 4-dimensional algebra with vector space $G^2$ spanned by the basis vectors $\{1, e_1, e_2, e_1 e_2\}$ where $e_1, e_2$ square to $+1$. The geometric product of two multivectors $\boldsymbol{a} = a_0 + a_1 e_1 + a_2 e_2 + a_{12} e_1 e_2$ and $\boldsymbol{b} = b_0 + b_1 e_1 + b_2 e_2 + b_{12} e_1 e_2$ is defined via:

$$\begin{aligned} \boldsymbol{ab} &= a_0 b_0 + a_0 b_1 \; e_1 \; + a_0 b_2 \; e_2 \; + a_0 b_{12} \; e_1 e_2 \\ &+ a_1 b_0 \; e_1 \; + a_1 b_1 e_1 e_1 + a_1 b_2 \; e_1 e_2 \; + a_1 b_{12} e_1 e_1 \; e_2 \\ &+ a_2 b_0 \; e_2 \; + a_2 b_1 \; e_2 e_1 \; + a_2 b_2 e_2 e_2 + a_2 b_{12} e_2 \; e_1 \; e_2 \\ &+ a_{12} b_0 \; e_1 e_2 \; + a_{12} b_1 e_1 \; e_2 \; e_1 + a_{12} b_2 \; e_1 \; e_2 e_2 + a_{12} b_{12} e_1 e_2 e_1 e_2 \; . \end{aligned} \tag{24}$$

Using the relations $e_1 e_1 = 1$, $e_2 e_2 = 1$, and $e_i e_j = -e_j e_i$ for $i \neq j \in \{e_1, e_2\}$, from which it follows that $e_1 e_2 e_1 e_2 = -1$, we obtain:

$$\begin{aligned} \boldsymbol{ab} &= a_0 b_0 + a_1 b_1 + a_2 b_2 - a_{12} b_{12} \\ &+ (a_0 b_1 + a_1 b_0 - a_2 b_{12} + a_{12} b_2) \; e_1 \\ &+ (a_0 b_2 + a_1 b_{12} + a_2 b_0 - a_{12} b_1) \; e_2 \\ &+ (a_0 b_{12} + a_1 b_2 - a_2 b_1 + a_{12} b_0) \; e_1 e_2 \; . \end{aligned} \tag{25}$$

A vector $x \in \mathbb{R}^2 \subset G^2$ is identified with $x_1 e_1 + x_2 e_2 \in \mathbb{R}^2 \subset G^2$. Clifford multiplication of two vectors $x, y \in \mathbb{R}^2 \subset G^2$ yields the geometric product $xy$:

$$\begin{aligned} xy &= (x_1 \; e_1 \; + x_2 \; e_2 \; )(y_1 \; e_1 \; + y_2 \; e_2 \; ) \\ &= \; x_1 y_1 e_1^2 + x_2 y_2 e_2^2 \; + \; x_1 y_2 e_1 e_2 + x_2 y_1 e_2 e_1 \\ &= \; \langle x, y \rangle \; + \; x \wedge y \; , \end{aligned} \tag{26}$$

<p style="text-align:center;">Inner product ↑      ↑ Outer/Wedge product</p>

The asymmetric quantity $x \wedge y = -y \wedge x$ is associated with the now often mentioned *bivector*, which can be interpreted as an oriented plane segment.

Equation 26 can be rewritten to express the (symmetric) inner product and the (anti-symmetric) outer product in terms of the geometric product:

$$x \wedge y = \frac{1}{2}(xy - yx) \tag{27}$$

$$\langle x, y \rangle = \frac{1}{2}(xy + yx) \; . \tag{28}$$

From the basis vectors of the vector space $G^2$ of the Clifford algebra $Cl_{2,0}(\mathbb{R})$, i.e. $\{1, e_1, e_2, e_1e_2\}$, probably the most interesting is $e_1e_2$. We therefore have a closer look the unit bivector $i_2 = e_1e_2$ which is the plane spanned by $e_1$ and $e_2$ and determined by the geometric product:

$$i_2 = e_1e_2 = \underbrace{\langle e_1, e_2 \rangle}_{=0} + e_1 \wedge e_2 = - e_2 \wedge e_1 = - e_2e_1 \,, \tag{29}$$

where the inner product $\langle e_1, e_2 \rangle$ is zero due to the orthogonality of the base vectors. The bivector $i_2$ if squared yields $i_2^2 = -1$, and thus $i_2$ represents a true geometric $\sqrt{-1}$. From Equation 29, it follows that

$$e_2 = e_1 i_2 = - i_2 e_1$$
$$e_1 = i_2 e_2 = - e_2 i_2 \,. \tag{30}$$

Using definition 2, the dual of a multivector $\boldsymbol{a} \in G^2$ is defined via the bivector as $i_2 \boldsymbol{a}$. Thus, the dual of a scalar is a bivector and the dual of a vector is again a vector. The dual pairs of the base vectors are $1 \leftrightarrow e_1e_2$ and $e_1 \leftrightarrow e_2$. These dual pairs allow us to write an arbitrary multivector $\boldsymbol{a}$ as

$$\boldsymbol{a} = a_0 + a_1e_1 + a_2e_2 + a_{12}e_{12} \,,$$
$$\boldsymbol{a} = 1 \underbrace{\left(a_0 + a_{12}i_2\right)}_{\text{spinor part}} + e_1 \underbrace{\left(a_1 + a_2i_2\right)}_{\text{vector part}} \,, \tag{31}$$

which can be regarded as two complex-valued parts: the spinor part, which commutes with $i_2$ and the vector part, which anti-commutes with $i_2$.

### A.2.3 CLIFFORD ALGEBRA $Cl_{0,2}(\mathbb{R})$

The Clifford algebra $Cl_{0,2}(\mathbb{R})$ is a 4-dimensional algebra with vector space $G^2$ spanned by the basis vectors $\{1, e_1, e_2, e_1e_2\}$ where $e_1, e_2$ square to $-1$. The Clifford algebra $Cl_{0,2}(\mathbb{R})$ is algebra-isomorphic to the quaternions $\mathbb{H}$, which are commonly written in literature (Schwichtenberg, 2015) as $a + b\hat{\imath} + c\hat{\jmath} + d\hat{k}$, where the (imaginary) base elements $\hat{\imath}$, $\hat{\jmath}$, and $\hat{k}$ fulfill the relations:

$$\hat{\imath}^2 = \hat{\jmath}^2 = -1$$
$$\hat{\imath}\hat{\jmath} = \hat{k}$$
$$\hat{\jmath}\hat{\imath} = -\hat{k}$$
$$\hat{k}^2 = \hat{\imath}\hat{\jmath}\hat{\imath}\hat{\jmath} = - \hat{\imath}\hat{\jmath}\hat{\jmath}\hat{\imath} = \hat{\imath}\hat{\imath} = -1 \,. \tag{32}$$

Quaternions also form a 4-dimensional algebra spanned by $\{1, \hat{\imath}, \hat{\jmath}, \hat{k}\}$, where $\hat{\imath}$, $\hat{\jmath}$, $\hat{k}$ all square to $-1$. The basis element 1 is often called the scalar part, and the basis elements $\hat{\imath}$, $\hat{\jmath}$, $\hat{k}$ are called the vector part of a quaternion.

The geometric product of two multivectors $\boldsymbol{a} = a_0 + a_1e_1 + a_2e_2 + a_{12}e_1e_2$ and $\boldsymbol{b} = b_0 + b_1e_1 + b_2e_2 + b_{12}e_1e_2$ is defined as:

$$\begin{aligned}
\boldsymbol{ab} = {} & a_0b_0 + a_0b_1\, e_1 + a_0b_2\, e_2 + a_0b_{12}\, e_1e_2 \\
& + a_1b_0\, e_1 + a_1b_1e_1e_1 + a_1b_2\, e_1e_2 + a_1b_{12}e_1e_1\, e_2 \\
& + a_2b_0\, e_2 + a_2b_1\, e_2e_1 + a_2b_2e_2e_2 + a_2b_{12}e_2\, e_1\, e_2 \\
& + a_{12}b_0\, e_1e_2 + a_{12}b_1e_1\, e_2\, e_1 + a_{12}b_2\, e_1\, e_2e_2 + a_{12}b_{12}e_1e_2e_1e_2 \,.
\end{aligned} \tag{33}$$

Using the relations $e_1e_1 = -1$, $e_2e_2 = -1$, and $e_ie_j = - e_je_i$ for $i \neq j \in \{e_1, e_2\}$, from which it follows that $e_1e_2e_1e_2 = -1$, we obtain:

$$\begin{aligned}
\boldsymbol{ab} = {} & a_0b_0 - a_1b_1 - a_2b_2 - a_{12}b_{12} \\
& + (a_0b_1 + a_1b_0 + a_2b_{12} - a_{12}b_2)\, e_1 \\
& + (a_0b_2 - a_1b_{12} + a_2b_0 + a_{12}b_1)\, e_2 \\
& + (a_0b_{12} + a_1b_2 - a_2b_1 + a_{12}b_0)\, e_1e_2 \,.
\end{aligned} \tag{34}$$

### A.2.4 CLIFFORD ALGEBRA $Cl_{3,0}(\mathbb{R})$

The Clifford algebra is a 8-dimensional algebra with vector space $G^3$ spanned by the basis vectors $\{1, e_1, e_2, e_3, e_1e_2, e_1e_3, e_2e_3, e_1e_2e_3\}$, i.e. one scalar, three vectors $\{e_1, e_2, e_3\}$, three bivectors $\{e_1e_2, e_1e_3, e_2e_3\}$, and one trivector $e_1e_2e_3$. The trivector is the pseudoscalar $i_3$ of the algebra. The geometric product of two multivectors is defined analogously to the geometric product of $Cl_{2,0}(\mathbb{R})$, following the associative and bilinear multiplication of multivectors follows:

$$e_i^2 = 1 \qquad \text{for } i = 1, 2, 3 \tag{35}$$

$$e_i e_j = -e_j e_i \qquad \text{for } i, j = 1, 2, 3, i \neq j . \tag{36}$$

Using Definition 2, the dual pairs of $Cl_{3,0}$ are:

$$1 \leftrightarrow e_1 e_2 e_3 = i_3 \tag{37}$$

$$e_1 \leftrightarrow e_2 e_3 \tag{38}$$

$$e_2 \leftrightarrow e_3 e_1 \tag{39}$$

$$e_3 \leftrightarrow e_1 e_2 . \tag{40}$$

The geometric product for $Cl_{3,0}(\mathbb{R})$ is defined analogously to the geometric product of $Cl_{2,0}(\mathbb{R})$ via:

$$
\begin{aligned}
\boldsymbol{ab} = {} & a_0 b_0 + a_0 b_1\, e_1 + a_0 b_2\, e_2 + a_0 b_3\, e_3 \\
& + a_0 b_{12}\, e_1 e_2 + a_0 b_{13}\, e_1 e_3 + a_0 b_{23}\, e_2 e_3 + a_0 b_{123}\, e_1 e_2 e_3 \\
& + a_1 b_0\, e_1 + a_1 b_1 e_1 e_1 + a_1 b_2\, e_1 e_2 + a_1 b_3\, e_1 e_3 \\
& + a_1 b_{12} e_1 e_1\, e_2 + a_1 b_{13} e_1 e_1\, e_3 + a_1 b_{23}\, e_1 e_2 e_3 + a_1 b_{123} e_1 e_1\, e_2 e_3 \\
& + a_2 b_0\, e_2 + a_2 b_1\, e_2 e_1 + a_2 b_2 e_2 e_2 + a_2 b_3\, e_2 e_3 \\
& + a_2 b_{12} e_2\, e_1\, e_2 + a_2 b_{13}\, e_1 e_3 e_2 + a_2 b_{23} e_2 e_2\, e_3 - a_2 b_{123} e_2 e_2\, e_1 e_3 \\
& + a_3 b_0\, e_3 + a_3 b_1\, e_3 e_1 + a_3 b_2\, e_3 e_2 + a_3 b_3 e_3 e_3 \\
& + a_3 b_{12}\, e_1 e_3 e_2 - a_3 b_{13}\, e_1\, e_3 e_3 - a_3 b_{23}\, e_2\, e_3 e_3 + a_3 b_{123}\, e_1 e_2\, e_3 e_3 \\
& + a_{12} b_0\, e_1 e_2 - a_{12} b_1\, e_2\, e_1 e_1 + a_{12} b_2\, e_1\, e_2 e_2 + a_{12} b_3\, e_1 e_2 e_3 \\
& + a_{12} b_{12} e_1 e_2 e_1 e_2 - a_{12} b_{13} e_1 e_1\, e_2 e_3 + a_{12} b_{23} e_2 e_2\, e_1 e_3 \\
& + a_{12} b_{123} e_1 e_2 e_1 e_2\, e_3 \\
& + a_{13} b_0\, e_1 e_3 - a_{13} b_1\, e_3\, e_1 e_1 + a_{13} b_2\, e_1 e_3 e_2 + a_{13} b_3\, e_1\, e_3 e_3 \\
& a_{13} b_{12} e_1 e_1\, e_3 e_2 + a_{13} b_{13} e_1 e_3 e_1 e_3 - a_{13} b_{23}\, e_1 e_2\, e_3 e_3 \\
& + a_{13} b_{123} e_1 e_3 e_1\, e_2\, e_3 \\
& + a_{23} b_0\, e_2 e_3 + a_{23} b_1\, e_1 e_3 e_2 + a_{23} b_2 e_2\, e_3\, e_2 + a_{23} b_3\, e_2\, e_3 e_3 \\
& + a_{23} b_{12} e_2\, e_3 e_1\, e_2 - a_{23} b_{13}\, e_2 e_1\, e_3 e_3 + a_{23} b_{23} e_2 e_3 e_2 e_3 \\
& + a_{23} b_{123} e_2 e_3\, e_1\, e_2 e_3 \\
& + a_{123} b_0\, e_1 e_2 e_3 + a_{123} b_1 e_1\, e_2 e_3\, e_1 - a_{123} b_2\, e_1 e_3\, e_1 e_2 + a_{123} b_3\, e_1 e_2\, e_3 e_3 \\
& + a_{123} b_{12} e_1 e_2\, e_2\, e_1 e_2 + a_{123} b_{13} e_1\, e_2\, e_3 e_1 e_3 + a_{123} b_{23}\, e_1\, e_2 e_3 e_2 e_3 \\
& + a_{123} b_{123} e_1 e_2 e_3 e_1 e_2 e_3 ,
\end{aligned}
\tag{41}
$$

where minus signs appear to do reordering of basis elements. Equation 41 simplifies to

$$
\begin{aligned}
\boldsymbol{ab} = {} & a_0b_0 + a_1b_1 + a_2b_2 + a_3b_3 - a_{12}b_{12} - a_{13}b_{13} - a_{23}b_{23} - a_{123}b_{123}` \\
& + (a_0b_1 + a_1b_0 - a_2b_{12} - a_3b_{13} + a_{12}b_2 + a_{13}b_3 - a_{23}b_{123} - a_{123}b_{23})\, e_1 \\
& + (a_0b_2 + a_1b_{12} + a_2b_0 - a_3b_{23} - a_{12}b_1 + a_{13}b_{123} + a_{23}b_3 + a_{123}b_{13})\, e_2 \\
& + (a_0b_3 + a_1b_{13} + a_2b_{23} + a_3b_0 - a_{12}b_{123} - a_{13}b_1 - a_{23}b_2 - a_{123}b_{12})\, e_3 \\
& + (a_0b_{12} + a_1b_2 - a_2b_1 + a_3b_{123} + a_{12}b_0 - a_{13}b_{23} + a_{23}b_{13} + a_{123}b_3)\, e_1e_2 \\
& + (a_0b_{13} + a_1b_3 - a_2b_{123} - a_3b_1 + a_{12}b_{23} + a_{13}b_0 - a_{23}b_{12} - a_{123}b_2)\, e_1e_3 \\
& + (a_0b_{23} + a_1b_{123} + a_2b_3 - a_3b_2 - a_{12}b_{13} + a_{13}b_{12} + a_{23}b_0 + a_{123}b_1)\, e_2e_3 \\
& + (a_0b_{123} + a_1b_{23} - a_2b_{13} + a_3b_{12} + a_{12}b_3 - a_{13}b_2 + a_{23}b_1 + a_{123}b_0)\, e_1e_2e_3 \ . \quad (42)
\end{aligned}
$$

## A.3 THE ELECTROMAGNETIC FIELD IN 3 DIMENSIONS

Through the lens of $Cl(3,0)(\mathbb{R})$, an intriguing example of the duality of multivectors is found when writing the expression of the electromagnetic field $\boldsymbol{F}$ in terms of an electric vector field $E$ and a magnetic vector field $B$ (Hestenes & Sobczyk, 2012; Hestenes, 2003), such that

$$
\boldsymbol{F} = E + Bi_3 \ . \tag{43}
$$

Both the electric field $E$ and the magnetic field $B$ are described by Maxwell's equations (Griffiths, 2005). The two fields are strongly coupled, e.g. temporal changes of electric fields induce magnetic fields and vice versa. Probably the most illustrative co-occurence of electric and magnetic fields is when describing the propagation of light. In standard vector algebra, $E$ is a vector while $B$ is a pseudovector, i.e. the two kinds of fields are distinguished by a difference in sign under space inversion. Equation 43 naturally decomposes the electromagnetic field into vector and bivector parts via the pseudoscalar $i_3$. For example, for the base component $B_x e_1$ of $B$ it holds that $B_x e_1 i_3 = B_x e_1 e_1 e_2 e_3 = B_x e_2 e_3$, which is a bivector and the dual to the base component $e_1$ of $E$. Geometric algebra reveals that a pseudovector is nothing else than a bivector represented by its dual, so the magnetic field $B$ in Equation 43 is fully represented by the complete bivector $Bi_3$, rather than $B$ alone. Consequently, the multivector representing $\boldsymbol{F}$ consists of three vectors (the electric field components) and three bivectors $e_1 i_3 = e_2 e_3, e_2 i_3 = e_3 e_1, e_3 i_3 = e_1 e_2$ (the magnetic field components multiplied by $i_3$).

# B    CLIFFORD NEURAL LAYERS

This appendix supports Section 3 of the main paper.

Clifford convolutions are related to the work on complex networks by Trabelsi et al. (2017), and closely related to work on quaternion neural networks  (Zhu et al., 2018; Parcollet et al., 2018a; Gaudet & Maida, 2018; Parcollet et al., 2018b; 2019; 2020; Nguyen et al., 2021).  Probably the most related work are (i) by Zang et al. (2022) who build geometric algebra convolution networks to process spatial and temporal data, and (ii) Spellings (2021) who build rotation- and permutation-equivariant graph network architectures based on geometric algebra products of node features. Higher order information is built from available node inputs.

## B.1    CLIFFORD CONVOLUTION LAYERS

We derive the implementation of translation equivariant Clifford convolution layers for multivectors in $G^2$, i.e. multivectors of Clifford algebras generated by the 2-dimensional vector space $\mathbb{R}^2$. Finally, we make the extension to Clifford algebras generated by the 3-dimensional vector space $\mathbb{R}^3$.

**Regular CNN layers.**    Regular convolutional neural network (CNN) layers take as input feature maps $f : \mathbb{Z}^2 \to \mathbb{R}^{c_{\text{in}}}$ and convolve[12] them with a set of $c_{\text{out}}$ filters $\{w^i\}_{i=1}^{c_{\text{out}}} : \mathbb{Z}^2 \to \mathbb{R}^{c_{\text{in}}}$:

$$\left[ f \star w^i \right](x) = \sum_{y \in \mathbb{Z}^2} \left\langle f(y), w^i(y - x) \right\rangle \tag{44}$$

$$= \sum_{y \in \mathbb{Z}^2} \sum_{j=1}^{c_{\text{in}}} f^j(y) w^{i,j}(y - x) . \tag{45}$$

Equation 44 can be interpreted as inner product of the input feature maps with corresponding filters at every point $y \in \mathbb{Z}^2$. By applying $c_{\text{out}}$ filters, the output feature maps can be interpreted as $c_{\text{out}}-$ dimensional features vectors at every point $y \in \mathbb{Z}^2$. We now want to extend convolution layers such that the elementwise product of scalars $f^j(y) w^{i,j}(y - x)$ are replaced by the geometric product of multivector inputs and multivector filters $\boldsymbol{f}^j(y) \boldsymbol{w}^{i,j}(y - x)$.

**Clifford CNN layers.**    We replace the feature maps $f : \mathbb{Z}^2 \to \mathbb{R}^{c_{\text{in}}}$ by multivector feature maps $\boldsymbol{f} : \mathbb{Z}^2 \to (G^2)^{c_{\text{in}}}$ and convolve them with a set of $c_{\text{out}}$ multivector filters $\{\boldsymbol{w}^i\}_{i=1}^{c_{\text{out}}} : \mathbb{Z}^2 \to (G^2)^{c_{\text{in}}}$:

$$\left[ \boldsymbol{f} \star \boldsymbol{w}^i \right](x) = \sum_{y \in \mathbb{Z}^2} \sum_{j=1}^{c_{\text{in}}} \underbrace{\boldsymbol{f}^j(y) \boldsymbol{w}^{i,j}(y - x)}_{\boldsymbol{f}^j \boldsymbol{w}^{i,j} \, : \, G^2 \times G^2 \to G^2} . \tag{46}$$

### B.1.1    TRANSLATION EQUIVARIANCE OF CLIFFORD CONVOLUTIONS

**Theorem 1: Translation equivariance of Clifford convolutions**

Let $\boldsymbol{f} : \mathbb{Z}^2 \to (G^2)^{c_{\text{in}}}$ be a multivector feature map and let $\boldsymbol{w} : \mathbb{Z}^2 \to (G^2)^{c_{\text{in}}}$ be a multivector kernel, then for $Cl(2,0)(\mathbb{R})$ $[[L_t \boldsymbol{f}] \star \boldsymbol{w}](x) = [L_t [\boldsymbol{f} \star \boldsymbol{w}]](x)$.

---

[12]In deep learning, a convolution operation in the forward pass is implemented as cross-correlation.

*Proof.*

$$[[L_t\boldsymbol{f}]\star\boldsymbol{w}](x) = \sum_{y\in\mathbb{Z}^2}\sum_{j=1}^{c_{\text{in}}}\boldsymbol{f}(y-t)\boldsymbol{w}(y-x)$$

$$= \sum_{y\in\mathbb{Z}^2}\sum_{j=1}^{c_{\text{in}}} f_0(y-t)w_0(y-x) + f_1(y-t)w_1(y-x) + f_2(y-t)w_2(y-x) - f_{12}(y-t)w_{12}(y-x)$$

$$+ \Big(f_0(y-t)w_1(y-x) + f_1(y-t)w_0(y-x) - f_2(y-t)w_{12}(y-x) + f_{12}(y-t)w_2(y-x)\Big)e_1$$

$$+ \Big(f_0(y-t)w_2(y-x) + f_1(y-t)w_{12}(y-x) + f_2(y-t)w_0(y-x) - f_{12}(y-t)w_1(y-x)\Big)e_2$$

$$+ \Big(f_0(y-t)w_{12}(y-x) + f_1(y-t)w_2(y-x) - f_2(f-t)w_1(y-x) + f_{12}(y-t)w_0(y-x)\Big)e_1 e_2$$

(using $y\to y-t$)

$$= \sum_{y\in\mathbb{Z}^2}\sum_{j=1}^{c_{\text{in}}} f_0(y)w_0(y-(x-t)) + f_1(y)w_1(y-(x-t)) + f_2(y)w_2(y-(x-t)) - f_{12}(y)w_{12}(y-(x-t))$$

$$+ \Big(f_0(y)w_1(y-(x-t)) + f_1(y)w_0(y-(x-t)) - f_2(y)w_{12}(y-(x-t)) + f_{12}(y)w_2(y-(x-t))\Big)e_1$$

$$+ \Big(f_0(y)w_2(y-(x-t)) + f_1(y)w_{12}(y-(x-t)) + f_2(y)w_0(y-(x-t)) - f_{12}(y)w_1(y-(x-t))\Big)e_2$$

$$+ \Big(f_0(y)w_{12}(y-(x-t)) + f_1(y)w_2(y-(x-t)) - f_2(y)w_1(y-(x-t)) + f_{12}(y)w_0(y-(x-t))\Big)e_1 e_2$$

$$= [L_t[\boldsymbol{f}\star\boldsymbol{w}]](x)\,. \tag{47}$$

$\square$

**Implementation of $Cl_{2,0}(\mathbb{R})$ and $Cl_{0,2}(\mathbb{R})$ layers.** We can implement a $Cl(2,0)(\mathbb{R})$ Clifford CNN layer using Equation 25 where $\{b_0,\,b_1,\,b_2,\,b_{12}\}\to\{w_0^{i,j},\,w_1^{i,j},\,w_2^{i,j},\,w_{12}^{i,j}\}$ correspond to 4 different kernels representing one 2D multivector kernel, i.e. 4 different convolution layers, and $\{a_0,\,a_1,\,a_2,\,a_{12}\}\to\{f_0^j,\,f_1^j,\,f_2^j,\,f_{12}^j\}$ correspond to the scalar, vector and bivector parts of the input multivector field. The channels of the different layers represent different stacks of scalars, vectors, and bivectors. All kernels have the same number of input and output channels (number of input and output multivectors), and thus the channels mixing occurs for the different terms of Equations 25, 42 individually. Lastly, usually not all parts of the multivectors are present in the input vector fields. This can easily be accounted for by just omitting the respective parts of Equations 25, 42. A similar reasoning applies to the output vector fields. For $Cl(0,2)(\mathbb{R})$, the signs within the geometric product change slightly.

### B.1.2 ROTATIONAL CLIFFORD CNN LAYERS

Here we introduce an alternative parameterization to the Clifford CNN layer introduced in Equation 7 by using the isomorphism of the Clifford algebra $Cl_{0,2}(\mathbb{R})$ to quaternions [13]. We take advantage of the fact that a quaternion rotation can be realized by a matrix multiplication (Jia, 2008; Kuipers, 1999; Schwichtenberg, 2015). Using the isomorphism, we can represent the feature maps $\boldsymbol{f}^j$ and filters $\boldsymbol{w}^{i,j}$ as quaternions: $\boldsymbol{f}^j = f_0^j + f_1^j\hat{\imath} + f_2^j\hat{\jmath} + f_3^j\hat{k}$ and $\boldsymbol{w}^{i,j} = w_0^{i,j} + w_1^{i,j}\hat{\imath} + w_2^{i,j}\hat{\jmath} + w_3^{i,j}\hat{k}$[14]. Leveraging this quaternion representation, we can devise an alternative parameterization of the product between the feature map $\boldsymbol{f}^j$ and $\boldsymbol{w}^{i,j}$. To be more precise, we introduce a composite operation that results in a scalar quantity and a quaternion rotation, where the latter acts on the vector part of the quaternion $\boldsymbol{f}^j$ and only produces nonzero expansion coefficients for the vector part of the quaternion output. A quaternion rotation $\boldsymbol{w}^{i,j}\boldsymbol{f}^j(\boldsymbol{w}^{i,j})^{-1}$ acts on the vector part $(\hat{\imath},\hat{\jmath},\hat{k})$ of $\boldsymbol{f}^j$, and

---

[13]We could not find neural rotational quaternion convolutions in existing literature, we however used the codebase of https://github.com/Orkis-Research/Pytorch-Quaternion-Neural-Networks as inspiration.

[14]Note that the expansion coefficients for the feature map $\boldsymbol{f}^j$ and filters $\boldsymbol{w}^{i,j}$ in terms of the basis elements of $G^2$ and in terms of quaternion elements $\hat{\imath}$, $\hat{\jmath}$ and $\hat{k}$ are the same.

can be algebraically manipulated into a vector-matrix operation $\boldsymbol{R}^{i,j}\boldsymbol{f}^j$, where $\boldsymbol{R}^{i,j} : \mathbb{H} \to \mathbb{H}$ is built up from the elements of $\boldsymbol{w}^{i,j}$ (Kuipers, 1999). In other words, one can transform the vector part $(\hat{\imath}, \hat{\jmath}, \hat{k})$ of $\boldsymbol{f}^j \in \mathbb{H}$ via a rotation matrix $\boldsymbol{R}^{i,j}$ that is built from the scalar and vector part $(1, \hat{\imath}, \hat{\jmath}, \hat{k})$ of $\boldsymbol{w}^{i,j} \in \mathbb{H}$. Altogether, a rotational multivector filter $\{\boldsymbol{w}_{\mathrm{rot}}^i\}_{i=1}^{c_{\mathrm{out}}} : \mathbb{Z}^2 \to (G^2)^{c_{\mathrm{in}}}$ acts on the feature map $\boldsymbol{f}^j$ through a rotational transformation $\boldsymbol{R}^{i,j}(w_{\mathrm{rot},0}^{i,j}, w_{\mathrm{rot},1}^{i,j}, w_{\mathrm{rot},2}^{i,j}, w_{\mathrm{rot},12}^{i,j})$ acting on vector and bivector parts of the multivector feature map $\boldsymbol{f} : \mathbb{Z}^2 \to (G^2)^{c_{\mathrm{in}}}$, and an additional scalar response of the multivector filters:

$$\left[\boldsymbol{f} \star \boldsymbol{w}_{\mathrm{rot}}^i\right](x) = \sum_{y \in \mathbb{Z}^2} \sum_{j=1}^{c_{\mathrm{in}}} \boldsymbol{f}^j(y)\boldsymbol{w}_{\mathrm{rot}}^{i,j}(y-x)$$

$$= \sum_{y \in \mathbb{Z}^2} \sum_{j=1}^{c_{\mathrm{in}}} \underbrace{\left[\boldsymbol{f}^j(y)\boldsymbol{w}_{\mathrm{rot}}^{i,j}(y-x))\right]_0}_{\text{scalar output}} + \boldsymbol{R}^{i,j}(y-x) \cdot \begin{pmatrix} f_1^j(y) \\ f_2^j(y) \\ f_{12}^j(y) \end{pmatrix} , \qquad (48)$$

where $\left[\boldsymbol{f}^j(y)\boldsymbol{w}_{\mathrm{rot}}^{i,j}(y-x))\right]_0 = f_0^j w_{\mathrm{rot},0}^{i,j} - f_1^j w_{\mathrm{rot},1}^{i,j} - f_2^j w_{\mathrm{rot},2}^{i,j} - f_{12}^j w_{\mathrm{rot},12}^{i,j}$ , which is the scalar output of Equation 34. The rotational matrix $\boldsymbol{R}^{i,j}(y-x)$ in written out form reads:

$$\boldsymbol{R}^{i,j} = \begin{pmatrix} 1 - 2\left[(\hat{w}_{\mathrm{rot},2}^{i,j})^2 + (\hat{w}_{\mathrm{rot},12}^{i,j})^2\right] & 2\left[\hat{w}_{\mathrm{rot},1}^{i,j}\hat{w}_{\mathrm{rot},2}^{i,j} - \hat{w}_{\mathrm{rot},0}^{i,j}\hat{w}_{\mathrm{rot},12}^{i,j}\right] & 2\left[\hat{w}_{\mathrm{rot},1}^{i,j}\hat{w}_{\mathrm{rot},12}^{i,j} + \hat{w}_{\mathrm{rot},0}^{i,j}\hat{w}_{\mathrm{rot},2}^{i,j}\right] \\ 2\left[\hat{w}_{\mathrm{rot},1}^{i,j}\hat{w}_{\mathrm{rot},2}^{i,j} + \hat{w}_{\mathrm{rot},0}^{i,j}\hat{w}_{\mathrm{rot},12}^{i,j}\right] & 1 - 2\left[(\hat{w}_{\mathrm{rot},1}^{i,j})^2 + (\hat{w}_{\mathrm{rot},12}^{i,j})^2\right] & 2\left[\hat{w}_{\mathrm{rot},2}^{i,j}\hat{w}_{\mathrm{rot},12}^{i,j} - \hat{w}_{\mathrm{rot},0}^{i,j}\hat{w}_{\mathrm{rot},1}^{i,j}\right] \\ 2\left[\hat{w}_{\mathrm{rot},1}^{i,j}\hat{w}_{\mathrm{rot},12}^{i,j} - \hat{w}_{\mathrm{rot},0}^{i,j}\hat{w}_{\mathrm{rot},2}^{i,j}\right] & 2\left[\hat{w}_{\mathrm{rot},2}^{i,j}\hat{w}_{\mathrm{rot},12}^{i,j} + \hat{w}_{\mathrm{rot},0}^{i,j}\hat{w}_{\mathrm{rot},12}^{i,j}\right] & 1 - 2\left[(\hat{w}_{\mathrm{rot},1}^{i,j})^2 + (\hat{w}_{\mathrm{rot},2}^{i,j})^2\right] \end{pmatrix} ,$$
$$(49)$$

where $\hat{\boldsymbol{w}}_{\mathrm{rot}}^{i,j}(y-x) = \hat{w}_{\mathrm{rot},0}^{i,j}(y-x) + \hat{w}_{\mathrm{rot},1}^{i,j}(y-x)e_1 + \hat{w}_{\mathrm{rot},2}^{i,j}(y-x)e_2 + \hat{w}_{\mathrm{rot},12}^{i,j}(y-x)e_{12}$ is the normalized filter with $\|\hat{\boldsymbol{w}}_{\mathrm{rot}}^{i,j}\| = 1$. The dependency $(y-x)$ is omitted inside the rotation matrix $\boldsymbol{R}^{i,j}$ for clarity.

### B.1.3   3D CLIFFORD CONVOLUTION LAYERS

**Implementation of $Cl_{3,0}(\mathbb{R})$ layers.**   Analogously to the 2-dimensional case, we can implement a 3D Clifford CNN layer using Equation 42, where $\{b_0, b_1, b_2, b_{12}, b_{13}, b_{23}, b_{123}\}$ correspond to 8 different kernels representing one 3D multivector kernel, i.e. 8 different convolution layers, and $\{a_0, a_1, a_2, a_{12}, a_{13}, a_{23}, a_{123}\}$ correspond to the scalar, vector, bivector, and trivector parts of the input multivector field. Convolution layers for different 3-dimensional Clifford algebras change the signs in the geometric product.

### B.2   CLIFFORD NORMALIZATION

Different normalization schemes have been proposed to stabilize and accelerate training deep neural networks (Ioffe & Szegedy, 2015; Ba et al., 2016; Wu & He, 2018; Ulyanov et al., 2017). Their standard formulation applies only to real values. Simply translating and scaling multivectors such that their mean is $\boldsymbol{0}$ and their variance is $\boldsymbol{1}$ is insufficient because it does not ensure equal variance across all components.

**Batch normalization**   Trabelsi et al. (2017) extended the batch normalization formulation to apply to complex values. We build on the same principles to first propose an appropriate batch normalization scheme for multivectors, similar to the work of Gaudet & Maida (2018) for quaternions. For 2D multivectors of the form $\boldsymbol{a} = a_0 + a_1e_1 + a_2e_2 + a_{12}e_1e_2$, we can formulate the problem of batch normalization as that of whitening 4D vectors:

$$\tilde{\boldsymbol{a}} = (\mathbf{V})^{-\frac{1}{2}}(\boldsymbol{a} - \mathbb{E}[\boldsymbol{a}]) \qquad (50)$$

where the covariance matrix $\mathbf{V}$ is

$$\mathbf{V} = \begin{pmatrix} V_{a_0 a_0} & V_{a_0 a_1} & V_{a_0 a_2} & V_{a_0 a_{12}} \\ V_{a_1 a_0} & V_{a_1 a_1} & V_{a_1 a_2} & V_{a_1 a_{12}} \\ V_{a_2 a_0} & V_{a_2 a_1} & V_{a_2 a_2} & V_{a_2 a_{12}} \\ V_{a_{12} a_0} & V_{a_{12} a_1} & V_{a_{12} a_2} & V_{a_{12} a_{12}} \end{pmatrix} . \qquad (51)$$

The shift parameter $\beta$ is a multivector with 4 learnable components and the scaling parameter $\gamma$ is $4 \times 4$ positive matrix. The multivector batch normalization is defined as:

$$BN(\boldsymbol{a}) = \gamma\boldsymbol{a} + \beta \tag{52}$$

When the batch sizes are small, it can be more appropriate to use Group Normalization or Layer Normalization. These can be derived with appropriate application of Eq. 50 along appropriate tensor dimensions. As such, batch, layer, and group normalization can be easily extended to 3-dimensional Clifford algebras.

### B.3 CLIFFORD INITIALIZATION

Parcollet et al. (2018a); Gaudet & Maida (2018) introduced initialization schemes for quaternions which expands upon deep network initialization schemes proposed by Glorot & Bengio (2010); He et al. (2015). Similar to Clifford normalization, quaternion initialization schemes can be adapted to Clifford layers in a straight forward way. Effectively, tighter bounds are required for the uniform distribution form which Clifford weights are sampled. However, despite intensive studies we did not observe any performance gains over default PyTorch initialization schemes[15] for 2-dimensional experiments. Similar findings are reported in Hoffmann et al. (2020). However, 3-dimensional implementations necessitate much smaller initialization values (factor $1/8$).

### B.4 EQUIVARIANCE UNDER ROTATIONS AND REFLECTIONS

Clifford convolutions satisfy the property of equivariance under translation of the multivector inputs, as shown in this Appendix B. However, the current definition of Clifford convolutions is not equivariant under multivector rotations or reflections. Here, we derive a general kernel constraint which allows us to build generalized Clifford convolutions which are equivariant w.r.t rotations or reflections of the multivectors. That is, we like to prove equivariance of a Clifford layer under rotations and reflections (i.e. orthogonal transformations) if the multivector kernel multivector filters $\{\boldsymbol{w}^i\}_{i=1}^{c_{out}} : \mathbb{Z}^2 \to (G)^{c_{in}}$ satisfies the constraint:

$$\boldsymbol{w}^{i,j}(Tx) = \mathbf{T}\boldsymbol{w}^{i,j}(x) \,,$$

for $0 \le j < c_{in}$. We first define an orthogonal transformation on a multivector by,

$$\mathbf{T}\boldsymbol{f} = \pm\mathbf{u}\boldsymbol{f}\mathbf{u}^\dagger, \quad \mathbf{u}^\dagger\mathbf{u} = 1 \tag{53}$$

where $\mathbf{u}$ and $\boldsymbol{f}$ are multivectors which are multiplied using the geometric product. The minus sign is picked by reflections but not by rotations, i.e. it depends on the parity of the transformation. This construction is called a "versor" product. The construction can be found in e.g. Suter (2003) for vectors and its extension to arbitrary multivectors. The above construction makes it immediately clear that $\mathbf{T}(\boldsymbol{f}\boldsymbol{g}) = (\mathbf{T}\boldsymbol{f})(\mathbf{T}\boldsymbol{g})$. When we write $Tx$, we mean an orthogonal transformation of an Euclidean vector (which can in principle also be defined using versors). To show equivariance, we wish to prove for multivectors $\boldsymbol{f} : \mathbb{Z}^2 \to (G)^{c_{in}}$ and a set of $c_{out}$ multivector filters $\{\boldsymbol{w}^i\}_{i=1}^{c_{out}} : \mathbb{Z}^2 \to (G)^{c_{in}}$ that:

$$\boldsymbol{f}'(Tx) = \mathbf{T}\boldsymbol{f}(x) \,, \tag{54}$$

and

$$\boldsymbol{w}^i(Tx) = \mathbf{T}\boldsymbol{w}^i(x) \,, \tag{55}$$

Equations 54, 55 yield:

$$\Rightarrow \left[\boldsymbol{f} \star \boldsymbol{w}^i\right]'(Tx) = \mathbf{T}\left[\boldsymbol{f} \star \boldsymbol{w}^i\right](x) \,. \tag{56}$$

That is: if the input multivector field transforms as a multivector, and the kernel satisfies the stated equivariance constraint, then the output multivector field also transforms properly as a multivector. Note that $\mathbf{T}$ might act differently on the various components (scalars, vectors, pseudoscalars, pseudovectors) under rotations and/or reflections.

---

[15]The default PyTorch initialization of linear and convolution layers is He Uniform initialization (He et al., 2015) for 2-dimensional problems. The gain is calculated using LeakyRelu activation functions with negative part of 5, which effectively results in Glorot Uniform initialization.

Now,

$$
\begin{aligned}
&\left[\boldsymbol{f} \star \boldsymbol{w}^i\right]'(Tx) \\
&= \sum_{y \in \mathbb{Z}^2} \sum_{j=1}^{c_{\text{in}}} \boldsymbol{f}'^j(y)\boldsymbol{w}^{i,j}(y - Tx)) \\
&= \sum_{y \in \mathbb{Z}^2} \sum_{j=1}^{c_{\text{in}}} \boldsymbol{f}'^j(y)\boldsymbol{w}^{i,j}(T(T^{-1}y - x))) \\
&= \sum_{Ty' \in \mathbb{Z}^2} \sum_{j=1}^{c_{\text{in}}} \boldsymbol{f}'^j(Ty')\boldsymbol{w}^{i,j}(T(y' - x))), \;\; y' = T^{-1}y \\
&= \sum_{y' \in \mathbb{Z}^2} \sum_{j=1}^{c_{\text{in}}} \boldsymbol{f}'^j(Ty')\boldsymbol{w}^{i,j}(T(y' - x))) \\
&= \sum_{y' \in \mathbb{Z}^2} \sum_{j=1}^{c_{\text{in}}} \mathbf{T}\boldsymbol{f}^j(y')\mathbf{T}\boldsymbol{w}^{i,j}(y' - x)) \\
&= \sum_{y' \in \mathbb{Z}^2} \sum_{j=1}^{c_{\text{in}}} \mathbf{T}(\boldsymbol{f}^j(y')\boldsymbol{w}^{i,j}(y' - x))) \\
&= \mathbf{T}\sum_{y' \in \mathbb{Z}^2} \sum_{j=1}^{c_{\text{in}}} (\boldsymbol{f}^j(y')\boldsymbol{w}^{i,j}(y' - x))) \\
&= \mathbf{T}\left[\boldsymbol{f} \star \boldsymbol{w}_i\right](x)
\end{aligned}
\tag{57}
$$

where in the fourth line we transform variables $y \to y'$, in the fifth line we use the invariance of the summation "measure" under $T$, in the sixth line we use the transformation property of $\boldsymbol{f}$ and equivariance for $\boldsymbol{w}^i$, in the seventh line we use the property of multivectors, and in the eighth line we use linearity of $\mathbf{T}$.

### B.5  CLIFFORD FOURIER LAYERS

We derive the implementation of Clifford Fourier layers for multivectors in $G^2$ and $G^3$, i.e. multivectors of Clifford algebras generated by the 2-dimensional vector space $\mathbb{R}^2$ and the 3-dimensional vector space $\mathbb{R}^3$.

**Classical Fourier transform.**  In arbitrary dimension $n$, the Fourier transform $\hat{f}(\xi) = \mathcal{F}\{f\}(\xi)$ for a continuous $n$-dimensional complex-valued signal $f(x) = f(x_1, \ldots, x_n) : \mathbb{R}^n \to \mathbb{C}$ is defined as:

$$
\hat{f}(\xi) = \mathcal{F}\{f\}(\xi) = \frac{1}{(2\pi)^{n/2}} \int_{\mathbb{R}^n} f(x)e^{-2\pi i \langle x, \xi \rangle}\, dx \,, \; \forall \xi \in \mathbb{R}^n \,,
\tag{58}
$$

provided that the integral exists, where $x$ and $\xi$ are $n$-dimensional vectors and $\langle x, \xi \rangle$ is the contraction of $x$ and $\xi$. Usually, $\langle x, \xi \rangle$ is the inner product, and $\xi$ is an element of the dual vector space $\mathbb{R}^{n\star}$. The inversion theorem states the back-transform from the frequency domain into the spatial domain:

$$
f(x) = \mathcal{F}^{-1}\{\mathcal{F}\{f\}\}(x) = \frac{1}{(2\pi)^{n/2}} \int_{\mathbb{R}^n} \hat{f}(\xi)e^{2\pi i \langle x, \xi \rangle}\, d\xi \,, \; \forall x \in \mathbb{R}^n \,.
\tag{59}
$$

We can rewrite the Fourier transform of Equation 58 in coordinates:

$$
\hat{f}(\xi_1, \ldots, \xi_n) = \mathcal{F}\{f\}(\xi_1, \ldots, \xi_n) = \frac{1}{(2\pi)^{n/2}} \int_{\mathbb{R}^n} f(x_1, \ldots, x_n)e^{-2\pi i(x_1\xi_1 + \ldots + x_n\xi_n)}dx_1 \ldots dx_n \,.
\tag{60}
$$

**Discrete/Fast Fourier transform.** The discrete counterpart of Equation 58 transforms an n-dimensional complex signal $f(x) = f(x_1, \ldots, x_n) : \mathbb{R}^n \to \mathbb{C}$ at $M_1 \times \ldots \times M_n$ grid points into its complex Fourier modes via:

$$\hat{f}(\xi_1, \ldots, \xi_n) = \mathcal{F}\{f\}(\xi_1, \ldots, \xi_n) = \sum_{m_1=0}^{M_1} \ldots \sum_{m_n=0}^{M_n} f(m_1, \ldots, m_n) \cdot e^{-2\pi i \cdot \left( \frac{m_1 \xi_1}{M_1} + \ldots + \frac{m_n \xi_n}{M_n} \right)} ,$$
(61)

where $(\xi_1, \ldots, \xi_n) \in \mathbb{Z}_{M_1} \ldots \times \ldots \mathbb{Z}_{M_n}$. Fast Fourier transforms (FFTs) (Cooley & Tukey, 1965; Van Loan, 1992) immensely accelerate the computation of the transformations of Equation 61 by factorizing the discrete Fourier transform matrix into a product of sparse (mostly zero) factors.

### B.5.1  2D CLIFFORD FOURIER TRANSFORM

Analogous to Equation 58, for $Cl(2,0)(\mathbb{R})$ the Clifford Fourier transform (Ebling & Scheuermann, 2005; Hitzer, 2012) and the respective inverse transform for multivector valued functions $\boldsymbol{f}(x) : \mathbb{R}^2 \to G^2$ and vectors $x, \xi \in \mathbb{R}^2$ are defined as:

$$\hat{\boldsymbol{f}}(\xi) = \mathcal{F}\{\boldsymbol{f}\}(\xi) = \frac{1}{2\pi} \int_{\mathbb{R}_2} \boldsymbol{f}(x) e^{-2\pi i_2 \langle x, \xi \rangle} \, dx , \quad \forall \xi \in \mathbb{R}^2 ,$$
(62)

$$\boldsymbol{f}(x) = \mathcal{F}^{-1}\{\mathcal{F}\{\boldsymbol{f}\}\}(x) = \frac{1}{2\pi} \int_{\mathbb{R}_2} \hat{\boldsymbol{f}}(\xi) e^{2\pi i_2 \langle x, \xi \rangle} \, d\xi , \quad \forall x \in \mathbb{R}^2 ,$$
(63)

provided that the integrals exist. The differences to Equations 58 and 59 are that $\boldsymbol{f}(x)$ and $\hat{\boldsymbol{f}}(\xi)$ represent multivector fields in the spatial and the frequency domain, respectively, and that the pseudoscalar $i_2 = e_1 e_2$ is used in the exponent. Inserting the definition of multivector fields, we can rewrite Equation 62 as:

$$\begin{aligned}
\mathcal{F}\{\boldsymbol{f}\}(\xi) &= \frac{1}{2\pi} \int_{\mathbb{R}_2} \boldsymbol{f}(x) e^{-2\pi i_2 \langle x, \xi \rangle} \, dx , \\
&= \frac{1}{2\pi} \int_{\mathbb{R}_2} \left[ 1 \Big( \underbrace{f_0(x) + f_{12}(x) i_2}_{\text{spinor part}} \Big) + e_1 \Big( \underbrace{f_1(x) + f_2(x) i_2}_{\text{vector part}} \Big) \right] e^{-2\pi i_2 \langle x, \xi \rangle} \, dx \\
&= \frac{1}{2\pi} \int_{\mathbb{R}_2} 1 \Big( f_0(x) + f_{12}(x) i_2 \Big) e^{-2\pi i_2 \langle x, \xi \rangle} \, dx \\
&\quad + \frac{1}{2\pi} \int_{\mathbb{R}_2} e_1 \Big( f_1(x) + f_2(x) i_2 \Big) e^{-2\pi i_2 \langle x, \xi \rangle} \, dx \\
&= 1 \left[ \mathcal{F}\Big( f_0(x) + f_{12}(x) i_2 \Big)(\xi) \right] + e_1 \left[ \mathcal{F}\Big( f_1(x) + f_2(x) i_2 \Big)(\xi) \right] .
\end{aligned}$$
(64)

We obtain a Clifford Fourier transform by applying two standard Fourier transforms for the dual pairs $\boldsymbol{f}_0 = f_0(x) + f_{12}(x) i_2$ and $\boldsymbol{f}_1 = f_1(x) + f_2(x) i_2$, which both can be treated as a complex-valued signal $\boldsymbol{f}_0, \boldsymbol{f}_1 : \mathbb{R}^2 \to \mathbb{C}$. Consequently, $\boldsymbol{f}(x)$ can be understood as an element of $\mathbb{C}^2$. The 2D Clifford Fourier transform is the linear combination of two classical Fourier transforms. The discretized versions of the spinor/vector part ($\hat{f}_{s/v}$) reads analogously to Equation 61:

$$\hat{f}_{s/v}(\xi_1, \xi_2) = \mathcal{F}\{f_{s/v}\}(\xi_1, \xi_2) = \sum_{m_1=0}^{M_1} \sum_{m_2=0}^{M_2} f_{s/v}(m_1, m_2) \cdot e^{-2\pi i_2 \left( \frac{m_1 \xi_1}{M_1} + \frac{m_2 \xi_2}{M_2} \right)} ,$$
(65)

where again $(\xi_1, \xi_2) \in \mathbb{Z}_{M_1} \times \mathbb{Z}_{M_n}$. Similar to Fourier Neural Operators (FNOs) where weight tensors are applied pointwise in the Fourier space, we apply multivector weight tensors $\boldsymbol{W} \in (G^2)^{c_{\text{in}} \times c_{\text{out}} \times (\xi_1^{\max} \times \xi_2^{\max})}$ point-wise. Fourier modes above cut-off frequencies $(\xi_1^{\max}, \xi_2^{\max})$ are set to zero. In doing so, we modify the Clifford Fourier modes

$$\hat{\boldsymbol{f}}(\xi) = \mathcal{F}\{\boldsymbol{f}\}(\xi) = \hat{f}_0(\xi) + \hat{f}_1(\xi) e_1 + \hat{f}_2(\xi) e_2 + \hat{f}_{12}(\xi) e_{12}$$
(66)

via the geometric product. The Clifford Fourier modes follow naturally when combining spinor and vector parts of Equation 64. Analogously to FNOs, higher order modes are cut off. Finally, the residual connections used in FNO layers is replaced by a multivector weight matrix realized as Clifford convolution, ideally a $Cl_{2,0}(\mathbb{R})$ convolution layer. A schematic sketch of a Clifford Fourier layer is shown in Figure 5b in the main paper. For $Cl(0,2)(\mathbb{R})$, the the vector part changes to

$$e_1\left(f_1(x) - f_2(x)i_2\right).$$

### B.5.2 2D CLIFFORD CONVOLUTION THEOREM

In contrast to Ebling & Scheuermann (2005), we proof the 2D Clifford convolution theorem for multivector valued filters applied from the right, such that filter operations are consistent with Clifford convolution layers. We first need to show that the Clifford kernel commutes with the spinor and anti-commutes with the vector part of multivectors. We can write the product $ae^{i_2s}$ for every scalar $s \in \mathbb{R}$ and multivector $a \in G^2$ as

$$\mathbf{a}e^{i_2s} = \mathbf{a}\big(\cos(s) + i_2\sin(s)\big). \tag{67}$$

For the basis of the spinor part, we obtain $1i_2 = i_21$, and for the basis of the vector part $e_1i_2 = e_1e_1e_2 = -e_1e_2e_1 = -i_2e_1$. Thus, the Fourier kernel $e^{-2\pi i_2\langle x,\xi\rangle}$ commutes with the spinor part, and anti-commutes with the vector part of $a$, both for $Cl(2,0)(\mathbb{R})$ and $Cl(0,2)(\mathbb{R})$. We therefore proof the convolution theorem for the commuting spinor and the anti-commuting vector part of $a$.

---

**Theorem 2: 2D Clifford convolution theorem.**

Let the field $\boldsymbol{f} : \mathbb{R}^2 \to G^2$ be multivector valued, the filter $\boldsymbol{k_s} : \mathbb{R}^2 \to G^2$ be spinor valued, and the filter $\boldsymbol{k_v} : \mathbb{R}^2 \to G^2$ be vector valued, and let $\mathcal{F}\{\boldsymbol{f}\}, \mathcal{F}\{\boldsymbol{k_s}\}, \mathcal{F}\{\boldsymbol{k_v}\}$ exist, then

$$\mathcal{F}\{\boldsymbol{f} \star \boldsymbol{k_s}\}(\xi) = \mathcal{F}\{\boldsymbol{f}\}(\xi) \cdot \mathcal{F}\dagger\{\boldsymbol{k_s}\}(\xi),$$
$$\mathcal{F}\{\boldsymbol{f} \star \boldsymbol{k_v}\}(\xi) = \mathcal{F}\{\boldsymbol{f}\}(\xi) \cdot \mathcal{F}\{\boldsymbol{k_v}\}(\xi),$$

where $\mathcal{F}^\dagger\{\boldsymbol{k_s}\}(\xi) = \mathcal{F}\{\boldsymbol{k_s}\}(-\xi)$ and $\mathcal{F}^\dagger\{\boldsymbol{k_v}\}(\xi) = \mathcal{F}\{\boldsymbol{k_v}\}(-\xi)$.

---

*Proof.*

$$
\begin{aligned}
\mathcal{F}\{\boldsymbol{f} \star \boldsymbol{k_s}\}(\xi) &= \frac{1}{(2\pi)^2} \int_{\mathbb{R}^2} \left[\int_{\mathbb{R}^2} \boldsymbol{f}(y)\boldsymbol{k_s}(y-x)dy\right] e^{-2\pi i_2\langle x,\xi\rangle}\ dx \\
&= \frac{1}{(2\pi)^2} \int_{\mathbb{R}^2} \boldsymbol{f}(y)\left[\int_{\mathbb{R}^2} \boldsymbol{k_s}(y-x)e^{-2\pi i_2\langle x,\xi\rangle}\ dx\right] dy \\
&= \frac{1}{(2\pi)^2} \int_{\mathbb{R}^2} \boldsymbol{f}(y)\left[\underbrace{\int_{\mathbb{R}^2} \boldsymbol{k_s}(x)e^{-2\pi i_2\langle y-x,\xi\rangle}\ dx}_{\mathcal{F}\dagger\{\boldsymbol{k_s}\}(\xi)e^{-2\pi i_2\langle y,\xi\rangle}=e^{-2\pi i_2\langle y,\xi\rangle}\mathcal{F}\dagger\{\boldsymbol{k_s}\}(\xi)}\right] dy \\
&= \frac{1}{2\pi}\left[\int_{\mathbb{R}^2} \boldsymbol{f}(y)e^{-2\pi i_2\langle y,\xi\rangle}dy\right] \mathcal{F}^\dagger\{\boldsymbol{k_s}\}(\xi) \\
&= \mathcal{F}\{\boldsymbol{f}\}(\xi) \cdot \mathcal{F}^\dagger\{\boldsymbol{k_s}\}(\xi). \tag{68}
\end{aligned}
$$

$$\mathcal{F}\{\boldsymbol{f} \star \boldsymbol{k_v}\}(\xi) = \frac{1}{(2\pi)^2} \int_{\mathbb{R}^2} \left[ \int_{\mathbb{R}^2} \boldsymbol{f}(y)\boldsymbol{k_v}(y-x)dy \right] e^{-2\pi i_2 \langle x, \xi \rangle} \, dx$$

$$= \frac{1}{(2\pi)^2} \int_{\mathbb{R}^2} \boldsymbol{f}(y) \left[ \int_{\mathbb{R}^2} \boldsymbol{k_v}(y-x) e^{-2\pi i_2 \langle x, \xi \rangle} \, dx \right] dy$$

$$= \frac{1}{(2\pi)^2} \int_{\mathbb{R}^2} \boldsymbol{f}(y) \left[ \underbrace{\int_{\mathbb{R}^2} \boldsymbol{k_v}(x) e^{-2\pi i_2 \langle y-x, \xi \rangle} \, dx}_{\mathcal{F}^\dagger\{\boldsymbol{k_v}\}(\xi) e^{2\pi i_2 \langle y, \xi \rangle} = e^{-2\pi i_2 \langle y, \xi \rangle} \mathcal{F}\{\boldsymbol{k_v}\}(\xi) \, , \text{ where } -\xi \to \xi} \right] dy$$

$$= \frac{1}{2\pi} \left[ \int_{\mathbb{R}^2} \boldsymbol{f}(y) e^{-2\pi i_2 \langle y, \xi \rangle} dy \right] \mathcal{F}\{\boldsymbol{k_v}\}(\xi)$$

$$= \mathcal{F}\{\boldsymbol{f}\}(\xi) \cdot \mathcal{F}\{\boldsymbol{k_v}\}(\xi) \, . \tag{69}$$

$\square$

### B.5.3 3D Clifford Fourier transform

For $Cl(3,0)(\mathbb{R})$, analogous to Equation 58, the Clifford Fourier transform (Ebling & Scheuermann, 2005) and the respective inverse transform for multivector valued functions $\boldsymbol{f} : \mathbb{R}^3 \to G^3$ and vectors $x, \xi \in \mathbb{R}^3$ are defined as:

$$\hat{\boldsymbol{f}}(\xi) = \mathcal{F}\{\boldsymbol{f}\}(\xi) = \frac{1}{(2\pi)^{3/2}} \int_{\mathbb{R}_3} \boldsymbol{f}(x) e^{-2\pi i_3 \langle x, \xi \rangle} \, dx \, , \, \forall \xi \in \mathbb{R}^3 \, , \tag{70}$$

$$\boldsymbol{f}(x) = \mathcal{F}^{-1}\{\mathcal{F}\{\boldsymbol{f}\}\}(x) = \frac{1}{(2\pi)^{3/2}} \int_{\mathbb{R}_3} \hat{\boldsymbol{f}}(\xi) e^{2\pi i_3 \langle x, \xi \rangle} \, d\xi \, , \, \forall x \in \mathbb{R}^3 \, , \tag{71}$$

provided that the integrals exist. A multivector valued function $\boldsymbol{f} : \mathbb{R}^3 \to G^3$,

$$\boldsymbol{f} = f_0 + f_1 e_1 + f_2 e_2 + f_3 e_3 + f_{12} e_{12} + f_{13} e_{13} + f_{23} e_{23} + f_{123} e_{123} \tag{72}$$

can be expressed via the pseudoscalar $i_3 = e_1 e_2 e_3$ as:

$$\begin{aligned} \boldsymbol{f} = {}&(f_0 + f_{123} i_3)1 \\ &+ (f_1 + f_{23} i_3)e_1 \\ &+ (f_2 + f_{31} i_3)e_2 \\ &+ (f_3 + f_{12} i_3)e_3 \, , \end{aligned} \tag{73}$$

We obtain a 3-dimensional Clifford Fourier transform by applying four standard Fourier transforms for the four dual pairs $\boldsymbol{f}_0 = f_0(x) + f_{123}(x)i_3$, $\boldsymbol{f}_1 = f_1(x) + f_{23}(x)i_3$, $\boldsymbol{f}_2 = f_2(x) + f_{31}(x)i_3$, and $\boldsymbol{f}_3 = f_3(x) + f_{12}(x)i_3$, which all can be treated as a complex-valued signal $\boldsymbol{f}_0, \boldsymbol{f}_1, \boldsymbol{f}_2, \boldsymbol{f}_3 : \mathbb{R}^3 \to \mathbb{C}$. Consequently, $\boldsymbol{f}(x)$ can be understood as an element of $\mathbb{C}^4$. The 3D Clifford Fourier transform is the linear combination of four classical Fourier transforms:

$$\mathcal{F}\{\boldsymbol{f}\}(\xi) = \frac{1}{(2\pi)^{3/2}} \int_{\mathbb{R}_3} \boldsymbol{f}(x) e^{-2\pi i_3 \langle x, \xi \rangle} \, dx \,,$$

$$= \frac{1}{(2\pi)^{3/2}} \int_{\mathbb{R}_3} \left[ 1\Big( f_0(x) + f_{123}(x) i_3 \Big) + \boxed{e_1} \left( f_1(x) + f_{23}(x) i_3 \right) \right.$$

$$\left. + \boxed{e_2} \left( f_2(x) + f_{31}(x) i_3 \right) + \boxed{e_3} \left( f_3(x) + f_{12}(x) i_3 \right) \right] e^{-2\pi i_3 \langle x, \xi \rangle} \, dx$$

$$= \frac{1}{(2\pi)^{3/2}} \int_{\mathbb{R}_3} 1\Big( f_0(x) + f_{123}(x) i_3 \Big) e^{-2\pi i_3 \langle x, \xi \rangle} \, dx$$

$$+ \frac{1}{(2\pi)^{3/2}} \int_{\mathbb{R}_3} \boxed{e_1} \Big( f_1(x) + f_{23}(x) i_3 \Big) e^{-2\pi i_3 \langle x, \xi \rangle} \, dx$$

$$+ \frac{1}{(2\pi)^{3/2}} \int_{\mathbb{R}_3} \boxed{e_2} \Big( f_2(x) + f_{31}(x) i_3 \Big) e^{-2\pi i_3 \langle x, \xi \rangle} \, dx$$

$$+ \frac{1}{(2\pi)^{3/2}} \int_{\mathbb{R}_3} \boxed{e_3} \Big( f_3(x) + f_{12}(x) i_3 \Big) e^{-2\pi i_3 \langle x, \xi \rangle} \, dx$$

$$= 1\left[ \mathcal{F}\Big( f_0(x) + f_{12}(x) i_3 \Big)(\xi) \right] + \boxed{e_1} \left[ \mathcal{F}\Big( f_1(x) + f_{23}(x) i_3 \Big)(\xi) \right]$$

$$+ \boxed{e_2} \left[ \mathcal{F}\Big( f_2(x) + f_{31}(x) i_3 \Big)(\xi) \right] + \boxed{e_3} \left[ \mathcal{F}\Big( f_3(x) + f_{12}(x) i_3 \Big)(\xi) \right]. \tag{74}$$

Analogous to the 2-dimensional Clifford Fourier transform, we apply multivector weight tensors $\boldsymbol{W} \in (G^3)^{c_{\text{in}} \times c_{\text{out}} \times (\xi_1^{\max} \times \xi_2^{\max} \times \xi_3^{\max})}$ point-wise. Fourier modes above cut-off frequencies $(\xi_1^{\max}, \xi_2^{\max}, \xi_3^{\max})$ are set to zero. In doing so, we modify the Clifford Fourier modes

$$\hat{\boldsymbol{f}}(\xi) = \mathcal{F}\{\boldsymbol{f}\}(\xi)$$
$$= \hat{f}_0(\xi) + \hat{f}_1(\xi) e_1 + \hat{f}_2(\xi) e_2 + \hat{f}_3(\xi) e_3 + \hat{f}_{12}(\xi) e_{12} + \hat{f}_{31}(\xi) e_{31} + \hat{f}_{23}(\xi) e_{23} + \hat{f}_{123}(\xi) e_{123} \tag{75}$$

via the geometric product. The Clifford Fourier modes follow naturally when combining the four parts of Equation 74 . Finally, the residual connections used in FNO layers is replaced by a multivector weight matrix realized as Clifford convolution, ideally a $Cl_{3,0}(\mathbb{R})$ convolution layer. For other 3-dimensional Clifford algebras, the signs of the dual pairs in Equation 73 change accordingly.

### B.5.4 3D CLIFFORD CONVOLUTION THEOREM

This theorem adapted from Ebling & Scheuermann (2005)). First, again let's check if the Clifford kernel commutes with the different parts of multivectors. We can write the product $\boldsymbol{a} e^{i_3 s}$ for every scalar $s \in \mathbb{R}$ and multivector $\boldsymbol{a} \in G^3$ as

$$\mathbf{a} e^{i_3 s} = \mathbf{a}\big( \cos(s) + i_3 \sin(s) \big) \,. \tag{76}$$

First, we check again if the different basis vectors of the Fourier transforms of Equation 74 commute with the pseudoscalar $i_3$:

$$1 i_3 = i_3 1 \checkmark$$
$$e_1 i_3 = e_1 e_1 e_2 e_3 = -e_1 e_2 e_1 e_3 = e_1 e_2 e_3 e_1 = i_3 e_1 \checkmark$$
$$e_2 i_3 = e_2 e_1 e_2 e_3 = -e_1 e_2 e_2 e_3 = e_1 e_2 e_3 e_2 = i_3 e_2 \checkmark$$
$$e_3 i_3 = e_3 e_1 e_2 e_3 = -e_1 e_3 e_2 e_3 = e_1 e_2 e_3 e_3 = i_3 e_3 \checkmark \tag{77}$$

In contrast to the 2-dimensional Clifford Fourier transform, now all four parts of the multivector of Equation 73 commute with $i_3$. This holds for all 3-dimensional Clifford algebras.

> **Theorem 3: 3D Clifford convolution theorem.**
>
> Let the field $\boldsymbol{f} : \mathbb{R}^3 \to G^3$ be multivector valued, the filter $\boldsymbol{k_a} : \mathbb{R}^3 \to G^3$ be multivector valued, and let $\mathcal{F}\{\boldsymbol{f}\}, \mathcal{F}\{\boldsymbol{k_a}\}$ exist, then
>
> $$\mathcal{F}\{\boldsymbol{f} \star \boldsymbol{k_a}\}(\xi) = \mathcal{F}\{\boldsymbol{f}\}(\xi) \cdot \mathcal{F}^\dagger\{\boldsymbol{k_a}\}(\xi) \,,$$
>
> where $\mathcal{F}^\dagger\{\boldsymbol{k_a}\}(\xi) = \mathcal{F}\{\boldsymbol{k_a}\}(-\xi)$.

*Proof.*

$$
\begin{aligned}
\mathcal{F}\{\boldsymbol{f} \star \boldsymbol{k_a}\}(\xi) &= \frac{1}{(2\pi)^3} \int_{\mathbb{R}^3} \left[ \int_{\mathbb{R}^3} \boldsymbol{f}(y)\boldsymbol{k_a}(y-x)dy \right] e^{-2\pi i_3 \langle x, \xi \rangle} \, dx \\
&= \frac{1}{(2\pi)^3} \int_{\mathbb{R}^3} \boldsymbol{f}(y) \left[ \int_{\mathbb{R}^3} \boldsymbol{k_a}(y-x)e^{-2\pi i_3 \langle x, \xi \rangle} \, dx \right] dy \\
&= \frac{1}{(2\pi)^3} \int_{\mathbb{R}^3} \boldsymbol{f}(y) \left[ \underbrace{\int_{\mathbb{R}^3} \boldsymbol{k_a}(x)e^{-2\pi i_3 \langle y-x, \xi \rangle} \, dx}_{\mathcal{F}^\dagger\{\boldsymbol{k_a}\}(\xi)e^{-2\pi i_3 \langle y, \xi \rangle} = e^{-2\pi i_3 \langle y, \xi \rangle} \mathcal{F}^\dagger\{\boldsymbol{k_a}\}(\xi)} \right] dy \\
&= \frac{1}{(2\pi)^{3/2}} \left[ \int_{\mathbb{R}^3} \boldsymbol{f}(y)e^{-2\pi i_3 \langle y, \xi \rangle} dy \right] \mathcal{F}^\dagger\{\boldsymbol{k_a}\}(\xi) \\
&= \mathcal{F}\{\boldsymbol{f}\}(\xi) \cdot \mathcal{F}^\dagger\{\boldsymbol{k_a}\}(\xi) \,.
\end{aligned}
\tag{78}
$$

$\square$

### B.5.5  IMPLEMENTATION OF CLIFFORD FOURIER LAYERS

We implement a 2D Clifford Fourier layer by applying two standard Fourier transforms on the dual pairs of Equation 11. These dual pairs can be treated as complex valued inputs. Similarly, we implement a 3D Clifford Fourier layer by applying four standard Fourier transforms on the dual pairs of e.g. $Cl_{3,0}$ (Equation 37 - Equation 40). Since Clifford convolution theorems hold both for the vector and the spinor parts and for the four dual pairs for $Cl_{2,0}$ and $Cl_{3,0}$, respectively, we multiply the modes in the Fourier space using the geometric product. Finally, we apply an inverse Fourier transformation and resemble the multivectors in the spatial domain.

### B.6  PSEUDOCODE

Algorithm 1 sketches the implementation of a Clifford convolution, Algorithm 2 of a rotational Clifford convolution, and Algorithm 3 of a Clifford Fourier layer.

```
1: function CLIFFORDKERNEL2D(W)
2:     kernel ← ⎡ W[0]  W[1]   W[2]  −W[3] ⎤
                ⎢ W[1]  W[0]  −W[3]   W[2] ⎥
                ⎢ W[2]  W[3]   W[0]  −W[1] ⎥
                ⎣ W[3]  W[2]  −W[1]   W[0] ⎦
3:     return kernel
4: function CLIFFORDCONV2D(W, x)
5:     kernel ← CLIFFORDKERNEL2D(W)
6:     input ← VIEW_AS_REALVECTOR(x)
7:     output ← CONV2D(kernel, input)
8:     return VIEW_AS_MULTIVECTOR(output)
```

Algorithm 1: Pseudocode for 2D Clifford convolution using $Cl_{2,0}$.

```
1: function CLIFFORDKERNEL2D_ROT(W)
2:     sq_12 ← W[1]² + W[2]²
3:     sq_13 ← W[1]² + W[3]²
4:     sq_23 ← W[2]² + W[3]²
5:     sumsq ← W[0]² + W[1]² + W[2]² + W[3]² + ε
6:     rot_12 ← W[0]W[1]/sumsq
7:     rot_13 ← W[0]W[2]/sumsq
8:     rot_14 ← W[0]W[3]/sumsq
9:     rot_23 ← W[1]W[2]/sumsq
10:    rot_24 ← W[1]W[3]/sumsq
11:    rot_34 ← W[2]W[3]/sumsq
```

$$
12: \quad \text{kernel} \leftarrow \begin{bmatrix} W[0] & -W[1] & -W[2] & -W[3] \\ W[5] & W[4](1.0 - sq_{23}) & W[4](rot_{23} - rot_{14}) & W[4](rot_{24} + rot_{13}) \\ W[5] & W[4](rot_{23} + rot_{14}) & W[4](1.0 - sq_{13}) & W[4](rot_{34} - rot_{12}) \\ W[5] & W[4](rot_{24} - rot_{13}) & W[4](rot_{34} + rot_{12}) & W[4](1.0 - sq_{12}) \end{bmatrix}
$$

```
13:    return kernel
14: function CLIFFORDCONV2D_ROT(W, x)
15:    kernel ← CLIFFORDKERNEL2D_ROT(W)
16:    input ← VIEW_AS_REALVECTOR(x)
17:    output ← CONV2D(kernel, input)
18:    return VIEW_AS_MULTIVECTOR(output)
```

Algorithm 2: Pseudocode for 2D rotational Clifford convolution using $Cl_{0,2}$.

```
1: function CLIFFORDSPECTRALCONV2D(W, x, m_1, m_2)
2:     x_v, x_v ← VIEW_AS_DUAL_PARTS(x)
3:     f(x_v) ← FFT2(x_v)                               ▷ Complex 2D FFT of vector part
4:     f(x_s) ← FFT2(x_s)                               ▷ Complex 2D FFT of scalar part
```

$$
5: \quad f^*(x_v) \leftarrow \begin{bmatrix} f(x_v)[\ldots, : m_1, : m_2] & f(x_v)[\ldots, : m_1, -m_2 :] \\ f(x_v)[\ldots, -m_1 :, : m_2] & f(x_v)[\ldots, -m_1 :, -m_2 :] \end{bmatrix} \quad \triangleright \text{Vector modes}
$$

$$
6: \quad f^*(x_s) \leftarrow \begin{bmatrix} f(x_s)[\ldots, : m_1, : m_2] & f(x_s)[\ldots, : m_1, -m_2 :] \\ f(x_s)[\ldots, -m_1 :, : m_2] & f(x_s)[\ldots, -m_1 :, -m_2 :] \end{bmatrix} \quad \triangleright \text{Scalar modes}
$$

```
7:     f*(x) ← f*(x_s).r + f*(x_v).r + f*(x_v).i + f*(x_s).i   ▷ Multivector Fourier modes
8:     f̂*(x) ← f*(x)W                                    ▷ Geometric product in the Fourier space
9:     x̂_v ← IFFT2(f̂*(x)[1] + f̂*(x)[2])                  ▷ Inverse 2D FFT of vector part
10:    x̂_2 ← IFFT2(f̂*(x)[0] + f̂*(x)[3])                  ▷ Inverse 2D FFT of scalar part
11:    x̂ ← VIEW_AS_MULTIVECTOR(x̂_v, x̂_s)
12:    return x̂
13: function CLIFFORDFOURIERLAYER2D(W_f, W_c, x)
14:    y_1 ← CLIFFORDSPECTRALCONV(W_f, x, m_1, m_2)
15:    x_2 ← VIEW_AS_REALVECTOR(x)
16:    y_2 ← CLIFFORDCONV(W_c, x_2)
17:    y_2 ← VIEW_AS_MULTIVECTOR(y_2)
18:    out ← ACTIVATION(y_1 + y_2)
19:    return out
20:
```

Algorithm 3: Pseudocode for 2D Clifford Fourier layer using $Cl_{2,0}$.

# C EXPERIMENTS

This appendix supports Section 4 of the main paper.

## C.1 LOSS FUNCTION AND METRICS

We report the summed MSE (SMSE) loss defined as:

$$\mathcal{L}_{\text{SMSE}} = \frac{1}{N_y} \sum_{y \in \mathbb{Z}^2 (\text{or} \mathbb{Z}^3)} \sum_{j=1}^{N_t} \sum_{i=1}^{N_{\text{fields}}} \|u_i(y, t_j) - \hat{u}_i(y, t_j)\|_2^2 , \tag{79}$$

where $u$ is the target, $\hat{u}$ the model output, $N_{\text{fields}}$ comprises scalar fields as well as individual vector field components, and $N_y$ is the total number of spatial points. Equation 79 is used for training with $N_t = 1$, and further allows us to define four metrics:

- *One-step* loss where $N_t = 1$ and $N_{\text{fields}}$ comprises all scalar and vector components.
- *Vector* loss where $N_t = 1$ and $N_{\text{fields}}$ comprises only vector components.
- *Scalar* loss where $N_t = 1$ and $N_{\text{fields}}$ comprises only the scalar field.
- *Rollout* loss where $N_t = 5$ and $N_{\text{fields}}$ comprises all scalar and vector components.

For Maxwell's equation, *electric* and *magnetic* loss are defined analogously to the vector and the scalar loss for Navier-Stokes and shallow water experiments.

## C.2 MODELS

We experiment with two architecture families: ResNet models (He et al., 2016) and Fourier Neural Operators (FNOs) (Li et al., 2020). All baseline models are fine-tuned for all individual experiments with respect to number of blocks, number of channels, number of modes (FNO), learning rates, normalization and initialization procedures, and activation functions. The best models are reported, and for reported Clifford results each convolution layer is substituted with a Clifford convolution, each Fourier layer with a Clifford Fourier layer, each normalization with a Clifford normalization and each non-linearity with a Clifford non-linearity. A Clifford non-linearity in this context is a the application of the corresponding default linearity to the different multivector components.

**ResNet architectures.** For Navier-Stokes and shallow water experiments, we use ResNet architectures with 8 residual blocks, each consisting of two convolution layers with $3 \times 3$ kernels, shortcut connections, group normalization (Wu & He, 2018), and GeLU activation functions (Hendrycks & Gimpel, 2016). We further use two embedding and two output layers, i.e. the overall architectures could be classified as Res-20 networks. In contrast to standard residual networks for image classification, we don't use any down-projection techniques, e.g. convolution layers with strides larger than 1 or via pooling layers. In contrast, the spatial resolution stays constant throughout the network. We therefore also use the same number of hidden channels throughout the network, that is 128 channels per layer. Overall this results in roughly 2.4 million parameters. Increasing the number of residual blocks or the number of channels did not increase the performance significantly.

**Clifford ResNet architectures.** For every ResNet-based experiment, we replaced the fine-tuned ResNet architectures with two Clifford counterparts: each CNN layer is replaced with a (i) Clifford CNN layer, and (ii) with a rotational Clifford CNN layer. To keep the number of weights similar, instead of 128 channels the resulting architectures have 64 multivector channels, resulting again in roughly 1.6 million floating point parameters. Additionally for both architectures, GeLU activation functions are replaced with Clifford GeLU activation functions, group normalization is replaced with Clifford group normalization. Using Clifford initialization techniques did not improve results.

**Fourier Neural Operator architectures.** For Navier-Stokes and shallow water experiments, we used 2-dimensional Fourier Neural Operators (FNOs) consisting of 8 FNO blocks, two embedding and two output layers. Each FNO block comprised a convolution path with a $1 \times 1$ kernel and an FFT path. We used 16 Fourier modes (for $x$ and $y$ components) for point-wise weight multiplication,

and overall use 128 hidden channels. We used GeLU activation functions (Hendrycks & Gimpel, 2016). Additional shortcut connections or normalization techniques, such as batchnorm or group, norm did not improve performance, neither did larger numbers of hidden channels, nor more FNO blocks. Overall this resulted in roughly 140 million parameters for FNO based architectures.

For 3-dimensional Maxwell experiments, we used 3-dimensional Fourier Neural Operators (FNOs) consisting of 4 FNO blocks, two embedding and two output layers. Each FNO block comprised a 3D convolution path with a $1 \times 1$ kernel and an FFT path. We used 6 Fourier modes (for $x$, $y$, and $z$ components) for point-wise weight multiplication, and overall used 96 hidden channels. Interestingly, using more layers or more Fourier modes degraded performances. Similar to the 2D experiments, we applied GeLU activation functions, and neither apply shortcut connections nor normalization techniques, such as batchnorm or groupnorms. Overall this resulted in roughly 65 million floating point parameters for FNO based architectures.

**Clifford Fourier Neural Operator architectures.** For every FNO-based experiment, we replaced the fine-tuned FNO architectures with respective Clifford counterparts: each FNO layer is replaced by its Clifford counterpart. To keep the number of weights similar, instead of 128 channels the resulting architectures have 48 multivector channels, resulting in roughly the same number of parameters. Additionally, GeLU activation functions are replaced with Clifford GeLU activation functions. Using Clifford initialization techniques did not improve results.

For 3-dimensional Maxwell experiments, we replaced each 3D Fourier transform layer with a 3D Clifford Fourier layer and each 3D convolution with a respective Clifford convolution. We also use 6 Fourier modes (for $x$, $y$, and $z$ components) for point-wise weight multiplication, and overall used 32 hidden multivector channels, which results in roughly the same number of parameters (55 millions). In contrast to 2-dimensional implementations, Clifford initialization techniques proved important for 3-dimensional architectures. Most notably, too large initial values of the weights of Clifford convolution layers hindered gradient flows through the Clifford Fourier operations.

### C.3 Training and model selection.

We optimized models using the Adam optimizer (Kingma & Ba, 2014) with learning rates $[10^{-4}, 2 \cdot 10^{-4}, 5 \cdot 10^{-4}]$ for 50 epochs and minimized the summed mean squared error (SMSE) which is outlined in Equation 79. We used cosine annealing as learning rate scheduler (Loshchilov & Hutter, 2016) with a linear warmup. For baseline ResNet models, we optimized number of layers, number of channels, and normalization procedures. We further tested different activation functions. For baseline FNO models, we optimized number of layers, number of channels, and number of Fourier modes. Larger numbers of layers or channels did not improve the performances for both ResNet and FNO models. For the respective Clifford counterparts, we exchanged convolution and Fourier layers by Clifford convolution and Clifford Fourier layers. We further used Clifford normalization schemes. We decreased the number of layers to obtain similar numbers of parameters. We could have optimized Clifford architectures slightly more by e.g. using different numbers of hidden layers than the baseline models did. However, this would (i) slightly be against the argument of having "plug- and play" replace layers, and (ii) would have added quite some computational overhead. Finally, we are quite confident that the used architectures are very close to the optimum for the current tasks.

**Computational resources.** All FNO and CFNO experiments used $4 \times 16$ GB NVIDIA V100 machines for training. All ResNet and Clifford ResNet experiments used $8 \times 32$ GB NVIDIA V100 machines. Average training times varied between $3$ h and $48$ h, depending on task and number of trajectories. Clifford runs on average took twice as long to train for equivalent architectures and epochs.

## C.4 NAVIER-STOKES IN 2D

The incompressible Navier-Stokes equations are built upon momentum and mass conservation of fluids. Momentum conservation yields for the velocity flow field $v$

$$\frac{\partial v}{\partial t} = -v \cdot \nabla v + \mu \nabla^2 v - \nabla p + f \, , \tag{80}$$

where $v \cdot \nabla v$ is the convection, $\mu \nabla^2 v$ the viscosity, $\nabla p$ the internal pressure and $f$ an external force. Convection is the rate of change of a vector field along a vector field (in this case along itself), viscosity is the diffusion of a vector field, i.e. the net movement form higher valued regions to lower concentration regions, $\mu$ is the viscosity coefficient. The incompressibility constrained yields mass conservation via

$$\nabla \cdot v = 0 \, . \tag{81}$$

Additional to the velocity field $v(x)$, we introduce a scalar field $s(x)$ representing a scalar quantity that is being transported through the velocity field. For example, $v$ might represent velocity of air inside a room, and $s$ might represent concentration of smoke. As the vector field changes, the scalar field is transported along it, i.e. the scalar field is *advected* by the vector field. Similar to convection, advection is the transport of a scalar field along a vector field:

$$\frac{ds}{dt} = -v \cdot \nabla s \, . \tag{82}$$

We implement the 2D Navier-Stokes equation using $\Phi\texttt{Flow}$[16] (Holl et al., 2020). Solutions are propagated where we solve for the pressure field and subtract its spatial gradients afterwards. Semi-Lagrangian advection (convection) is used for $v$, and MacCormack advection for $s$. Additionally, we express the external buoyancy force $f$ in Equation 80 as force acting on the scalar field. Solutions are obtained using Boussinesq approximation (Kleinstreuer, 1997), which ignores density differences except where they appear in terms multiplied by the acceleration due to gravity. The essence of the Boussinesq approximation is that the difference in inertia is negligible but gravity is sufficiently strong to make the specific weight appreciably different between the two fluids.

**Equation details.** We obtain data for the 2D Navier-Stokes equations on a grid with spatial resolution of $128 \times 128$ ($\Delta x = 0.25$, $\Delta y = 0.25$), and temporal resolution of $\Delta t = 1.5\,\text{s}$. The equation is solved on a closed domain with Dirichlet boundary conditions ($v = 0$) for the velocity, and Neumann boundaries $\frac{\partial s}{\partial x} = 0$ for the scalar smoke field. The viscosity parameter is set to $\nu = 0.01$, and a buoyancy factor of $(0, 0.5)^T$ is used. The scalar field is initialized with random Gaussian noise fluctuations, and the velocity field is initialized to $0$. We run the simulation for $21\,\text{s}$ and sample every $1.5\,\text{s}$. Trajectories contain scalar and vector fields at $14$ different time points.

**Results.** Results are summarized in Figures 10, 9, and detailed in Table 1. Figure 11 displays examples of Navier-Stokes rollouts of scalar and vector fields obtained by Clifford Fourier surrogates, and contrasts them with ground truth trajectories. For ResNet-like architectures, we observe that both CResNet and CResNet$_{\text{rot}}$ improve upon the ResNet baseline. Additionally, we observe that rollout losses are also lower for the two Clifford based architectures, which we attribute to better and more stable models that do not overfit to one-step predictions so easily. Lastly, while in principle CResNet and CResNet$_{\text{rot}}$ based architectures are equally flexible, CResNet$_{\text{rot}}$ ones in general perform better than CResNet ones. For FNO and respective Clifford Fourier based (CFNO) architectures, the loss is in general much lower than for ResNet based architectures. CFNO architectures improve upon FNO architectures for all dataset sizes, and for one-step as well as rollout losses.

---

[16] https://github.com/tum-pbs/PhiFlow

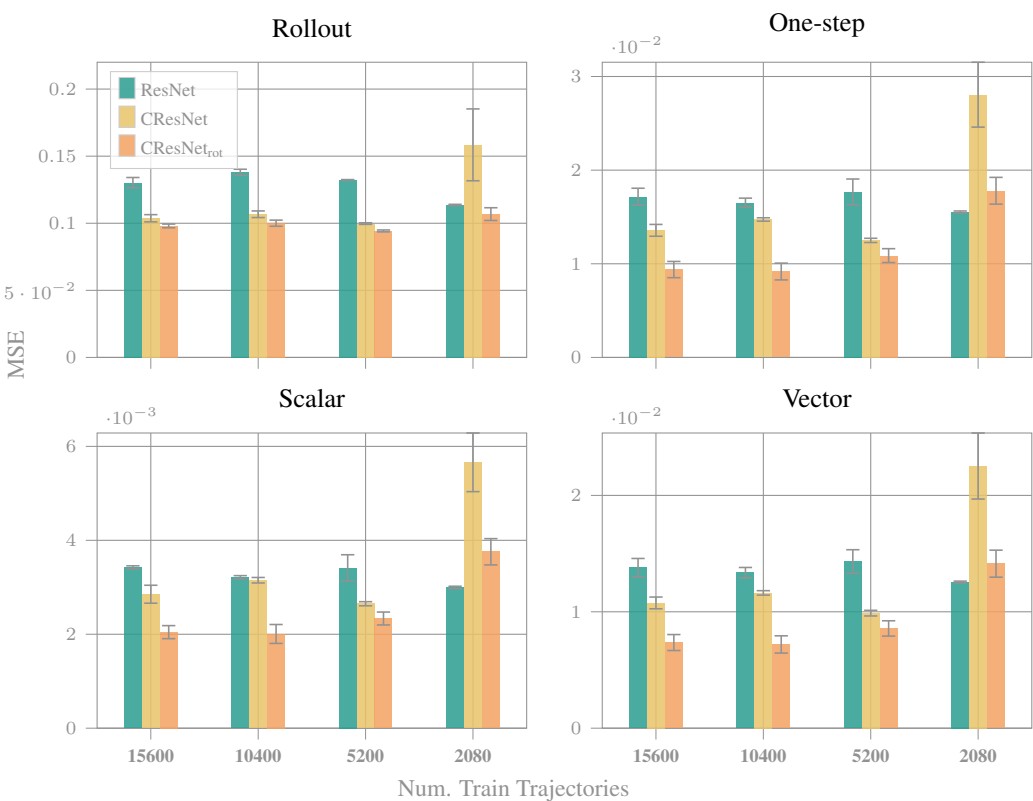

Figure 9: Results on Navier-Stokes equations obtained by ResNet based architectures. Unrolled loss, one-step loss, scalar loss and vector loss are reported for ResNet, CResNet, and CResNet$_{rot}$ architectures. Models are trained on training sets with increasing number of trajectories. ResNet based architectures have a much higher loss than FNO based architectures in the low data regime, where possibly smearing and averaging operations are learned first.

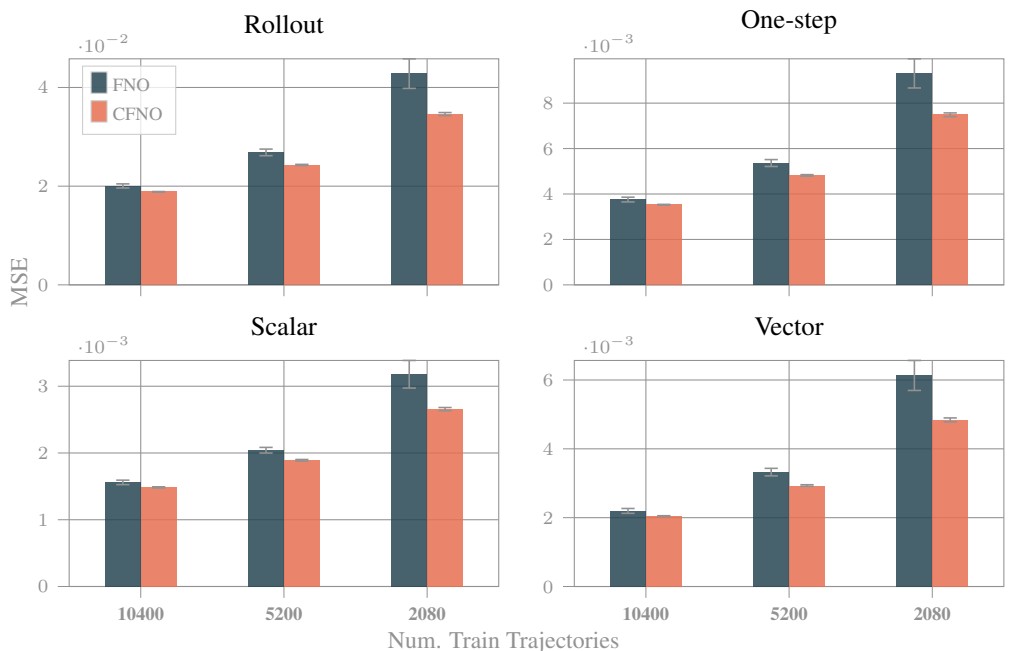

Figure 10: Results on Navier-Stokes equations obtained by Fourier based architectures. Rollout loss, one-step loss, scalar loss and vector loss are reported for FNO and CFNO architectures. Models are trained on three training sets with increasing number of trajectories.

Table 1: Model comparison on four different metrics for neural PDE surrogates which are trained on Navier-Stokes training datasets of varying size. Error bars are obtained by running experiments with three different initial seeds.

| | | SMSE | | | |
|---|---|---|---|---|---|
| METHOD | Trajs. | scalar | vector | onestep | rollout |
| ResNet | | $0.003\,00 \pm 0.000\,03$ | $0.012\,55 \pm 0.000\,08$ | $0.015\,53 \pm 0.000\,11$ | $0.113\,62 \pm 0.000\,48$ |
| CResNet | 2080 | $0.005\,66 \pm 0.000\,62$ | $0.022\,52 \pm 0.002\,84$ | $0.028\,06 \pm 0.003\,46$ | $0.158\,44 \pm 0.026\,77$ |
| CResNet$_{rot}$ | | $0.003\,76 \pm 0.000\,28$ | $0.014\,13 \pm 0.001\,16$ | $0.017\,80 \pm 0.001\,43$ | $0.106\,81 \pm 0.004\,76$ |
| ResNet | | $0.003\,41 \pm 0.000\,28$ | $0.014\,31 \pm 0.001\,02$ | $0.017\,67 \pm 0.001\,38$ | $0.132\,34 \pm 0.000\,20$ |
| CResNet | 5200 | $0.002\,65 \pm 0.000\,04$ | $0.009\,88 \pm 0.000\,24$ | $0.012\,50 \pm 0.000\,22$ | $0.099\,75 \pm 0.000\,60$ |
| CResNet$_{rot}$ | | $0.002\,34 \pm 0.000\,14$ | $0.008\,57 \pm 0.000\,66$ | $0.010\,87 \pm 0.000\,74$ | $0.094\,27 \pm 0.000\,71$ |
| ResNet | | $0.003\,21 \pm 0.000\,04$ | $0.013\,37 \pm 0.000\,44$ | $0.016\,53 \pm 0.000\,48$ | $0.138\,02 \pm 0.002\,23$ |
| CResNet | 10400 | $0.003\,15 \pm 0.000\,06$ | $0.011\,62 \pm 0.000\,19$ | $0.014\,73 \pm 0.000\,18$ | $0.106\,71 \pm 0.002\,46$ |
| CResNet$_{rot}$ | | $0.002\,01 \pm 0.000\,20$ | $0.007\,19 \pm 0.000\,74$ | $0.009\,17 \pm 0.000\,90$ | $0.100\,05 \pm 0.002\,29$ |
| ResNet | | $0.003\,42 \pm 0.000\,03$ | $0.013\,79 \pm 0.000\,79$ | $0.017\,16 \pm 0.000\,91$ | $0.130\,30 \pm 0.003\,79$ |
| CResNet | 15600 | $0.002\,85 \pm 0.000\,19$ | $0.010\,76 \pm 0.000\,51$ | $0.013\,57 \pm 0.000\,63$ | $0.103\,72 \pm 0.002\,69$ |
| CResNet$_{rot}$ | | $0.002\,04 \pm 0.000\,14$ | $0.007\,36 \pm 0.000\,69$ | $0.009\,38 \pm 0.000\,87$ | $0.097\,99 \pm 0.001\,39$ |
| FNO | 2080 | $0.003\,18 \pm 0.000\,21$ | $0.006\,13 \pm 0.000\,44$ | $0.009\,31 \pm 0.000\,64$ | $0.042\,81 \pm 0.003\,00$ |
| CFNO | | $0.002\,66 \pm 0.000\,02$ | $0.004\,84 \pm 0.000\,06$ | $0.007\,49 \pm 0.000\,08$ | $0.034\,61 \pm 0.000\,31$ |
| FNO | 5200 | $0.002\,04 \pm 0.000\,04$ | $0.003\,32 \pm 0.000\,11$ | $0.005\,36 \pm 0.000\,15$ | $0.026\,84 \pm 0.000\,67$ |
| CFNO | | $0.001\,89 \pm 0.000\,01$ | $0.002\,93 \pm 0.000\,02$ | $0.004\,82 \pm 0.000\,03$ | $0.024\,30 \pm 0.000\,12$ |
| FNO | 10400 | $0.001\,56 \pm 0.000\,03$ | $0.002\,20 \pm 0.000\,07$ | $0.003\,75 \pm 0.000\,10$ | $0.020\,05 \pm 0.000\,42$ |
| CFNO | | $0.001\,48 \pm 0.000\,01$ | $0.002\,05 \pm 0.000\,01$ | $0.003\,53 \pm 0.000\,02$ | $0.018\,86 \pm 0.000\,06$ |

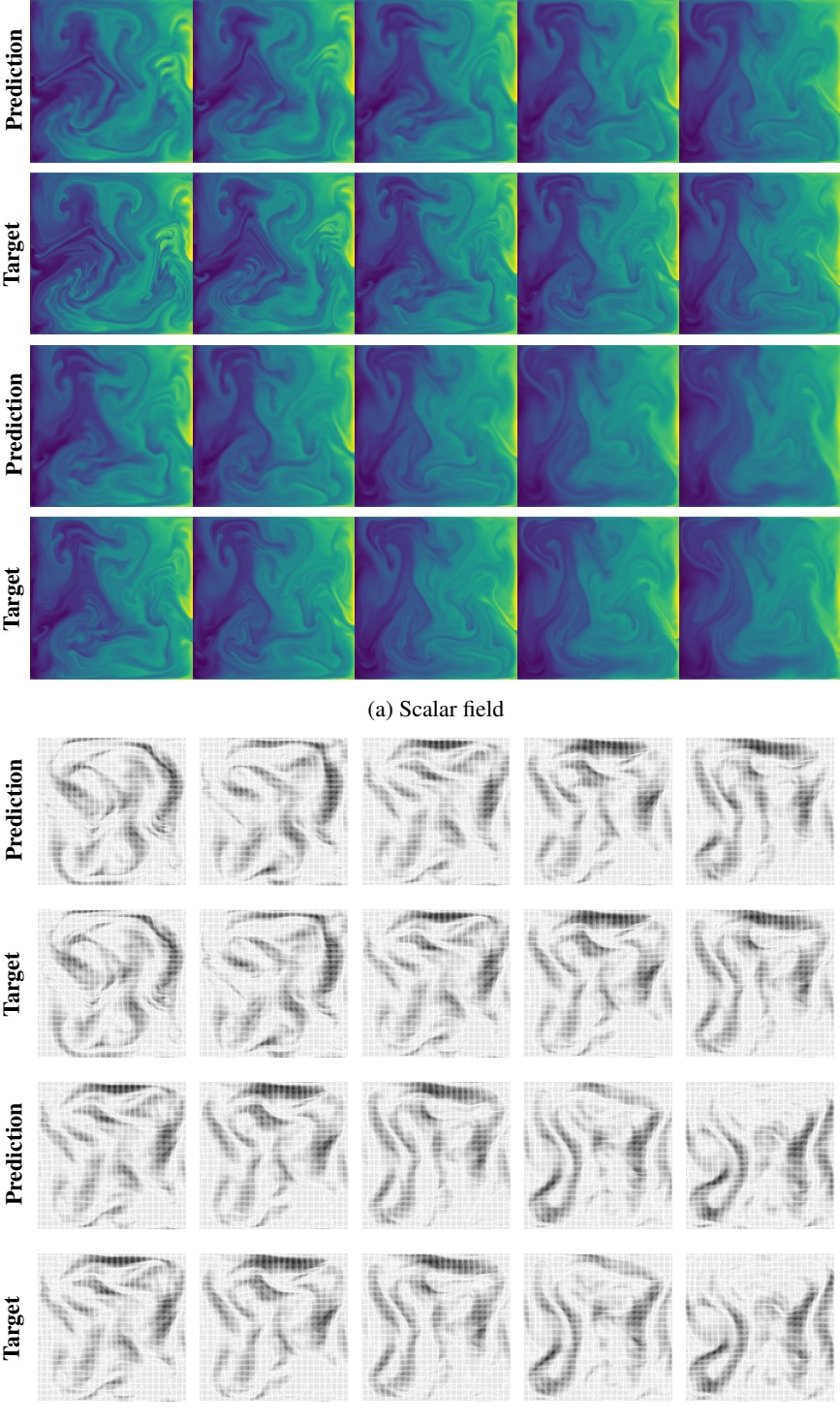

(a) Scalar field

(b) Vector field

Figure 11: Example rollouts of the scalar and vector field of the Navier-Stokes experiments, obtained by a Clifford Fourier PDE surrogate and the ground truth.

## C.5    SHALLOW WATER EQUATIONS.

The shallow water equations (Vreugdenhil, 1994) describe a thin layer of fluid of constant density in hydrostatic balance, bounded from below by the bottom topography and from above by a free surface. For example, the deep water propagation of a tsunami can be described by the shallow water equations, and so can a simple weather model. The shallow water equations read:

$$
\frac{\partial v_x}{\partial t} + v_x \frac{\partial v_x}{\partial x} + v_y \frac{\partial v_x}{\partial y} + g \frac{\partial \eta}{\partial x} = 0 \,,
$$
$$
\frac{\partial v_y}{\partial t} + v_x \frac{\partial v_y}{\partial x} + v_y \frac{\partial v_y}{\partial y} + g \frac{\partial \eta}{\partial y} = 0 \,,
$$
$$
\frac{\partial \eta}{\partial t} + \frac{\partial}{\partial x} \Big[ (\eta + h) v_x \Big] + \frac{\partial}{\partial y} \Big[ (\eta + h) v_y \Big] = 0 \,, \tag{83}
$$

where $v_x$ is the velocity in the $x$-direction, or zonal velocity, $v_y$ is the velocity in the $y$-direction, or meridional velocity, $g$ is the acceleration due to gravity, $\eta(x, y)$ is the vertical displacement of free surface, which subsequently is used to derive pressure fields; $h(x, y)$ is the topography of the earth's surface. We modify the implementation in `SpeedyWeather.jl`[17](Klöwer et al., 2022) to further randomize initial conditions to generate our dataset. `SpeedyWeather.jl` combines the shallow water equations with spherical harmonics for the linear terms and Gaussian grid for the non-linear terms with the appropriate spectral transforms. It internally uses a leapfrog time scheme with a Robert and William's filter to dampen the computational modes and achieve 3rd oder accuracy. `SpeedyWeather.jl` is based on the atmospheric general circulation model `SPEEDY` in Fortran (Molteni, 2003; Kucharski et al., 2013).

**Equation details.**    We obtain data for the 2D shallow water equations on a grid with spatial resolution of $192 \times 96$ ($\Delta x = 1.875°$, $\Delta y = 3.75°$), and temporal resolution of $\Delta t = 6\,\text{h}$. The equation is solved on a closed domain with periodic boundary conditions. We rollout the simulation for 20 days and sample every 6 h. Here 20 days is of course not the actual simulation time but rather the simulated time. Trajectories contain scalar pressure and wind vector fields at 84 different time points.

**Results.**    Results are summarized in Figures 12, 13, 14, and detailed in Tables 2, 3. Figure 15 displays examples of shallow water equations rollouts of scalar pressure and vector wind fields obtained by Clifford Fourier surrogate models, and contrasts them with ground truth trajectories. The predictions are fairly indistinguishable from ground truth trajectories. We observe similar results than for the Navier-Stokes experiments. However, performance differences between baseline and Clifford architectures are even more pronounced, which we attribute to the stronger coupling of the scalar and the vector fields. For ResNet-like architectures, CResNet and CResNet$_{\text{rot}}$ improve upon the ResNet baseline, rollout losses are much lower for the two Clifford based architectures, and CResNet$_{\text{rot}}$ based architectures in general perform better than CResNet based ones. For Fourier based architectures, the loss is in general much lower than for ResNet based architectures (a training set size of 56 trajectories yields similar (C)FNO test set performance than a training set size of 896 trajectories for ResNet based architectures). CFNO architectures improve upon FNO architectures for all dataset sizes, and for one-step as well as rollout losses, which is especially pronounced for low number of training trajectories.

---

[17]https://github.com/milankl/SpeedyWeather.jl

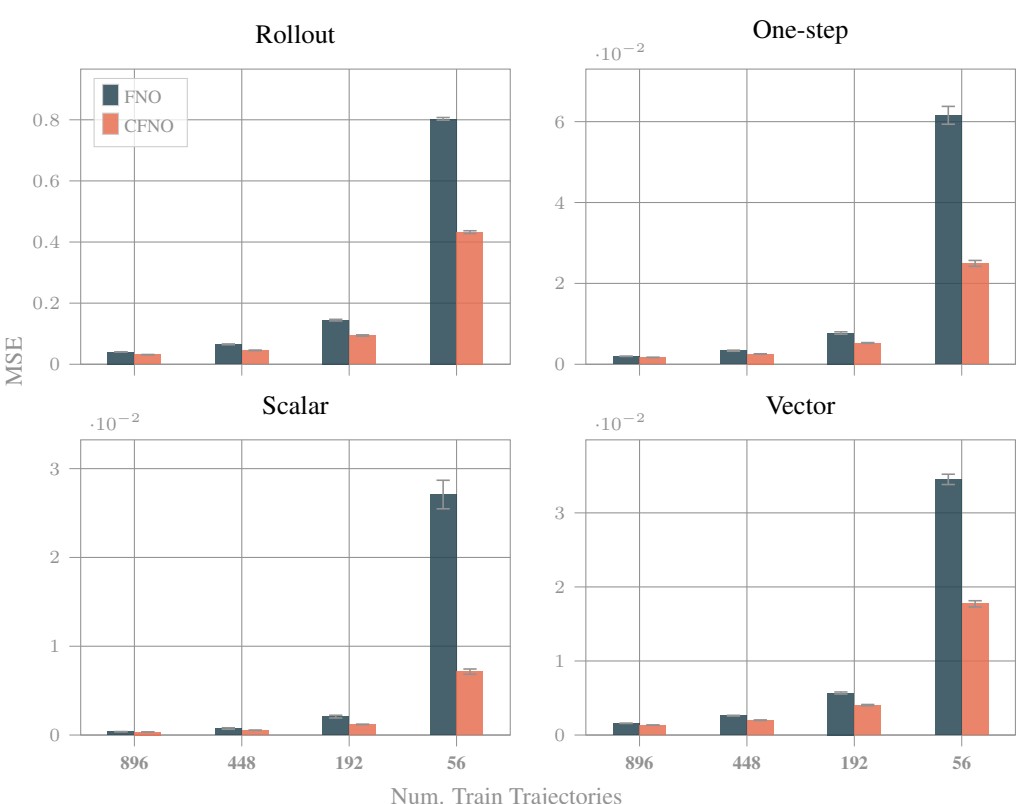

Figure 12: Results on the shallow water equations obtained by Fourier based architectures using a **two timestep history input**. Unrolled loss, one-step loss, scalar loss and vector loss are reported for FNO and CFNO architectures. Models are trained on three training sets with increasing number of trajectories.

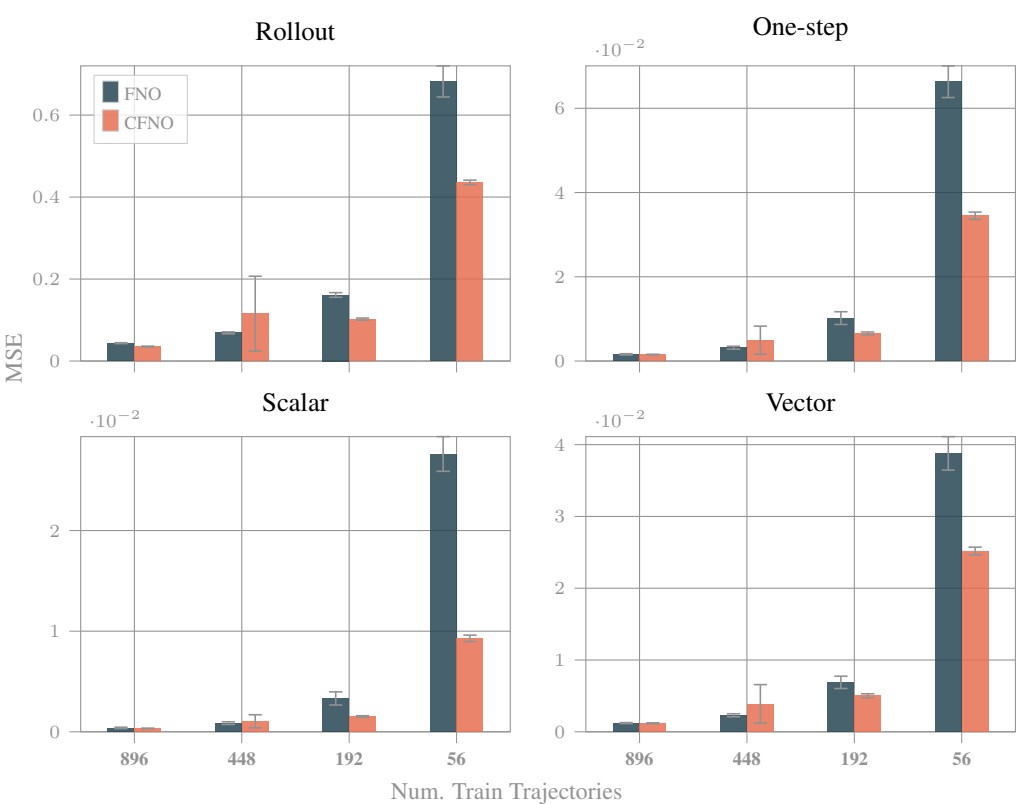

Figure 13: Results on the shallow water equations obtained by Fourier based architectures using a **four timestep history input**. Rollout loss, one-step loss, scalar loss and vector loss are reported for FNO and CFNO architectures. Models are trained on three training sets with increasing number of trajectories.

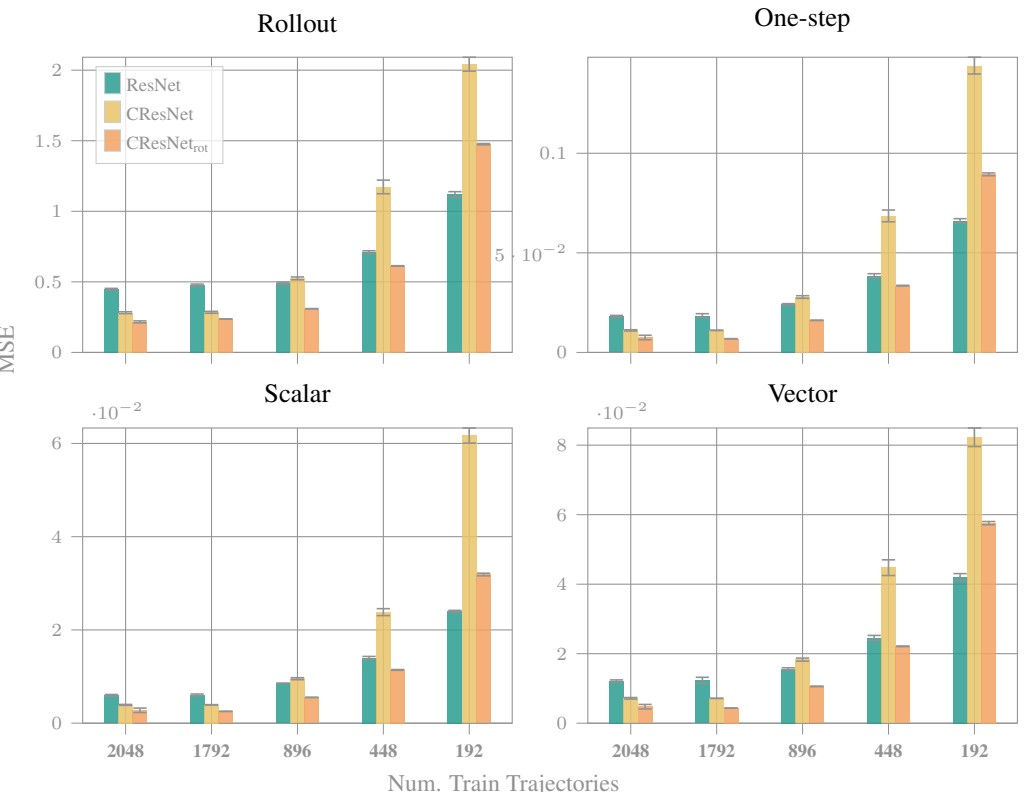

Figure 14: Results on the shallow water equations obtained by ResNet based architectures using a **two timestep history input**. Rollout loss, one-step loss, scalar loss and vector loss are reported for ResNet, CResNet, and CResNet$_{rot}$ architectures. Models are trained on training sets with increasing number of trajectories. ResNet based architectures have a much higher loss than FNO based architectures in the low data regime, where possibly smearing and averaging operations are learned first.

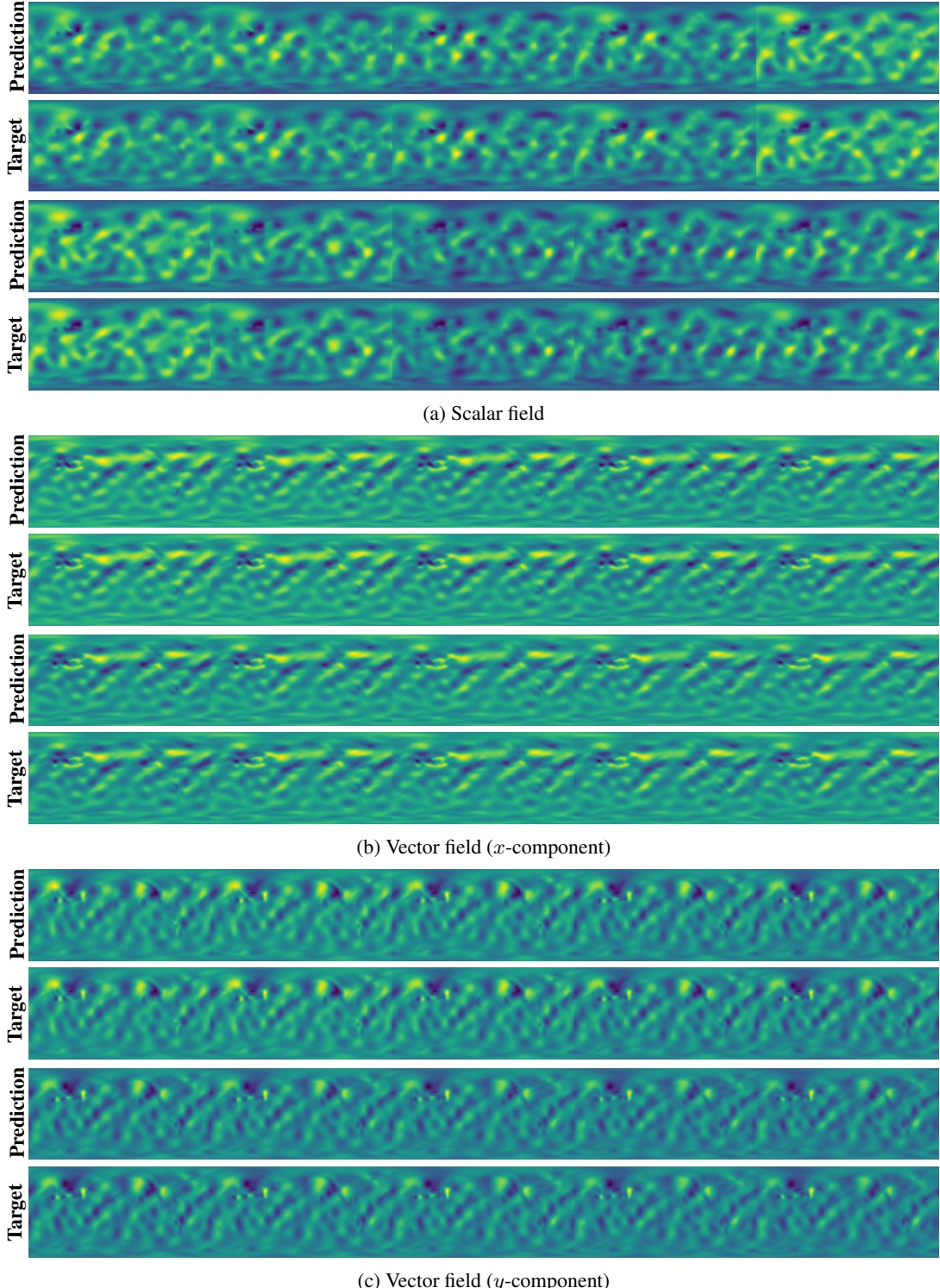

(a) Scalar field

(b) Vector field ($x$-component)

(c) Vector field ($y$-component)

Figure 15: Example rollouts of the scalar and vector field of the shallow water experiments, obtained by a Clifford Fourier PDE surrogate (top) and the ground truth (bottom).

Table 2: Model comparison on four different metrics for neural PDE surrogates which are trained on the shallow water equations training datasets of varying size. Results are obtained by using a **two timestep history input**. Error bars are obtained by running experiments with three different initial seeds.

| METHOD | Trajs. | SMSE | | | |
|---|---|---|---|---|---|
| | | scalar | vector | onestep | rollout |
| ResNet | | $0.0240 \pm 0.0002$ | $0.0421 \pm 0.0010$ | $0.0661 \pm 0.0011$ | $1.1195 \pm 0.0197$ |
| CResNet | 192 | $0.0617 \pm 0.0016$ | $0.0823 \pm 0.0027$ | $0.1440 \pm 0.0042$ | $2.0423 \pm 0.0494$ |
| CResNet$_{rot}$ | | $0.0319 \pm 0.0003$ | $0.0576 \pm 0.0005$ | $0.0894 \pm 0.0007$ | $1.4756 \pm 0.0044$ |
| ResNet | | $0.0140 \pm 0.0003$ | $0.0245 \pm 0.0007$ | $0.0385 \pm 0.0010$ | $0.7083 \pm 0.0119$ |
| CResNet | 448 | $0.0238 \pm 0.0007$ | $0.0448 \pm 0.0023$ | $0.0685 \pm 0.0030$ | $1.1727 \pm 0.0483$ |
| CResNet$_{rot}$ | | $0.0114 \pm 0.0001$ | $0.0221 \pm 0.0001$ | $0.0335 \pm 0.0002$ | $0.6127 \pm 0.0018$ |
| ResNet | | $0.0086$ | $0.0156 \pm 0.0003$ | $0.0242 \pm 0.0003$ | $0.4904 \pm 0.0080$ |
| CResNet | 896 | $0.0095 \pm 0.0002$ | $0.0183 \pm 0.0004$ | $0.0278 \pm 0.0006$ | $0.5247 \pm 0.0101$ |
| CResNet$_{rot}$ | | $0.0055$ | $0.0106 \pm 0.0001$ | $0.0161 \pm 0.0001$ | $0.3096 \pm 0.0010$ |
| ResNet | | $0.0061 \pm 0.0002$ | $0.0123 \pm 0.0009$ | $0.0184 \pm 0.0010$ | $0.4780 \pm 0.0062$ |
| CResNet | 1792 | $0.0039$ | $0.0071$ | $0.0111 \pm 0.0001$ | $0.2842 \pm 0.0067$ |
| CResNet$_{rot}$ | | $0.0025$ | $0.0044$ | $0.0069$ | $0.2370$ |
| ResNet | | $0.0060 \pm 0.0002$ | $0.0121 \pm 0.0003$ | $0.0181 \pm 0.0005$ | $0.4480 \pm 0.0058$ |
| CResNet | 2048 | $0.0039 \pm 0.0001$ | $0.0072 \pm 0.0002$ | $0.0111 \pm 0.0003$ | $0.2816 \pm 0.0065$ |
| CResNet$_{rot}$ | | $0.0028 \pm 0.0005$ | $0.0480 \pm 0.0006$ | $0.0075 \pm 0.0011$ | $0.2164 \pm 0.0070$ |
| FNO | 56 | $0.0271 \pm 0.0016$ | $0.0345 \pm 0.0007$ | $0.0616 \pm 0.0022$ | $0.8032 \pm 0.0043$ |
| CFNO | | $0.0071 \pm 0.0003$ | $0.0177 \pm 0.0004$ | $0.0250 \pm 0.0007$ | $0.4323 \pm 0.0046$ |
| FNO | 192 | $0.0021 \pm 0.0002$ | $0.0057 \pm 0.0001$ | $0.0077 \pm 0.0003$ | $0.1444 \pm 0.0026$ |
| CFNO | | $0.0012$ | $0.0040 \pm 0.0001$ | $0.0053 \pm 0.0001$ | $0.0941 \pm 0.0021$ |
| FNO | 448 | $0.0007 \pm 0.0001$ | $0.0026$ | $0.0034 \pm 0.0001$ | $0.0651 \pm 0.0014$ |
| CFNO | | $0.0005$ | $0.0020$ | $0.0026 \pm 0.0001$ | $0.0455 \pm 0.0009$ |
| FNO | 896 | $0.0004$ | $0.0016$ | $0.0020 \pm 0.0001$ | $0.0404 \pm 0.0005$ |
| CFNO | | $0.0003$ | $0.0013$ | $0.0017 \pm 0.0001$ | $0.0315 \pm 0.0004$ |

Table 3: Model comparison on four different metrics for neural PDE surrogates which are trained on the shallow water equations training datasets of varying size. Results are obtained by using a **four timestep history input**. Error bars are obtained by running experiments with three different initial seeds.

| METHOD | Trajs. | SMSE | | | |
|---|---|---|---|---|---|
| | | scalar | vector | onestep | rollout |
| FNO | 56 | $0.0276 \pm 0.0017$ | $0.0388 \pm 0.0023$ | $0.0663 \pm 0.0038$ | $0.6821 \pm 0.0379$ |
| CFNO | | $0.0093 \pm 0.0003$ | $0.0252 \pm 0.0005$ | $0.0345 \pm 0.0009$ | $0.4357 \pm 0.0056$ |
| FNO | 192 | $0.0033 \pm 0.0007$ | $0.0069 \pm 0.0009$ | $0.0102 \pm 0.0015$ | $0.1612 \pm 0.0057$ |
| CFNO | | $0.0015 \pm 0.0001$ | $0.0050 \pm 0.0003$ | $0.0065 \pm 0.0003$ | $0.1023 \pm 0.0026$ |
| FNO | 448 | $0.0009 \pm 0.0001$ | $0.0023 \pm 0.0002$ | $0.0032 \pm 0.0003$ | $0.0687 \pm 0.0023$ |
| CFNO | | $0.0010 \pm 0.0006$ | $0.0039 \pm 0.0027$ | $0.0050 \pm 0.0033$ | $0.1156 \pm 0.0913$ |
| FNO | 896 | $0.0004 \pm 0.0001$ | $0.0012 \pm 0.0001$ | $0.0016 \pm 0.0001$ | $0.0436 \pm 0.0011$ |
| CFNO | | $0.0003$ | $0.0012 \pm 0.0001$ | $0.0015 \pm 0.0001$ | $0.0353 \pm 0.0010$ |

C.6 MAXWELL'S EQUATIONS IN MATTER IN 3D.

Electromagnetic simulations play a critical role in understanding light–matter interaction and designing optical elements. Neural networks have been already successful applied in inverse-designing photonic structures (Ma et al., 2021b; Lim & Psaltis, 2022).

Maxwell's equations in matter read:

$$\nabla \cdot D = \rho \qquad\qquad \text{Gauss's law} \qquad\qquad (84)$$
$$\nabla \cdot B = 0 \qquad\qquad \text{Gauss's law for magnetism} \qquad\qquad (85)$$
$$\nabla \times E = -\frac{\partial B}{\partial t} \qquad\qquad \text{Faraday's law of induction} \qquad\qquad (86)$$
$$\nabla \times H = \frac{\partial D}{\partial t} + \boldsymbol{j} \qquad\qquad \text{Ampère's circuital law} \qquad\qquad (87)$$

In isotropic media, the displacement field $D$ is related to the electrical field via $D = \epsilon_0 \epsilon_r E$, where $\epsilon_0$ is the permittivity of free space and $\epsilon_r$ is the permittivity of the media. Similarly, the magnetization field $H$ in isotropic media is related to the magnetic field $B$ via $H = \mu_0 \mu_r B$, where $\mu_0$ is the permeability of free space and $\mu_r$ is the permeability of the media. Lastly, $\boldsymbol{j}$ is the electric current density and $\rho$ the total electric charge density.

We propagate the solution of Maxwell's equation in matter using a finite-difference time-domain method[18], where the discretized Maxwell's equations are solved in a leapfrog manner. First, the electric field vector components in a volume of space are solved at a given instant in time. Second, the magnetic field vector components in the same spatial volume are solved at the next instant in time.

**Equation details.** We obtain data for the 3D Maxwell's equations on a grid with spatial resolution of $32 \times 32 \times 32$ ($\Delta x = \Delta y = \Delta z = 5 \cdot 10^{-7} m$), and temporal resolution of $\Delta t = 50$ s. We randomly place 18 (6 in the $x-y$ plane, 6 in the $x-z$ plane, 6 in the $y-z$ plane) different light sources outside a cube which emit light with different amplitude and different phase shifts, causing the resulting $D$ and $H$ fields to interfere with each other. The wavelength of the emitted light is $10^{-5} m$. The equation is solved on a closed domain with periodic boundary conditions. We run the simulation for $400$ s and sample data every $50$ s. Trajectories contain displacement $D$ and the magnetization field $H$ components. Exemplary trajectories are shown in Figure 16.

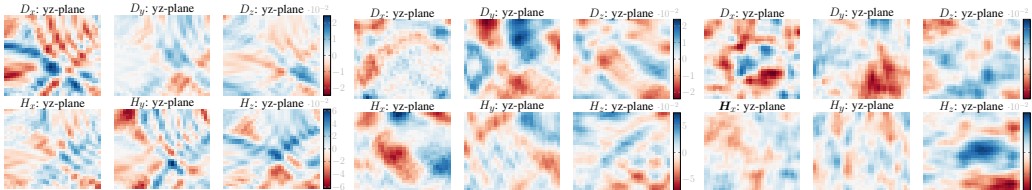

Figure 16: An example propagation of the displacement field $D$ and the magnetization field $H$. Shown are the field components for an arbitrary slice of the $x-y$ plane.

**Results.** Results are summarized in Figure 17 and detailed in Table 4.

---

[18]https://github.com/flaport/fdtd

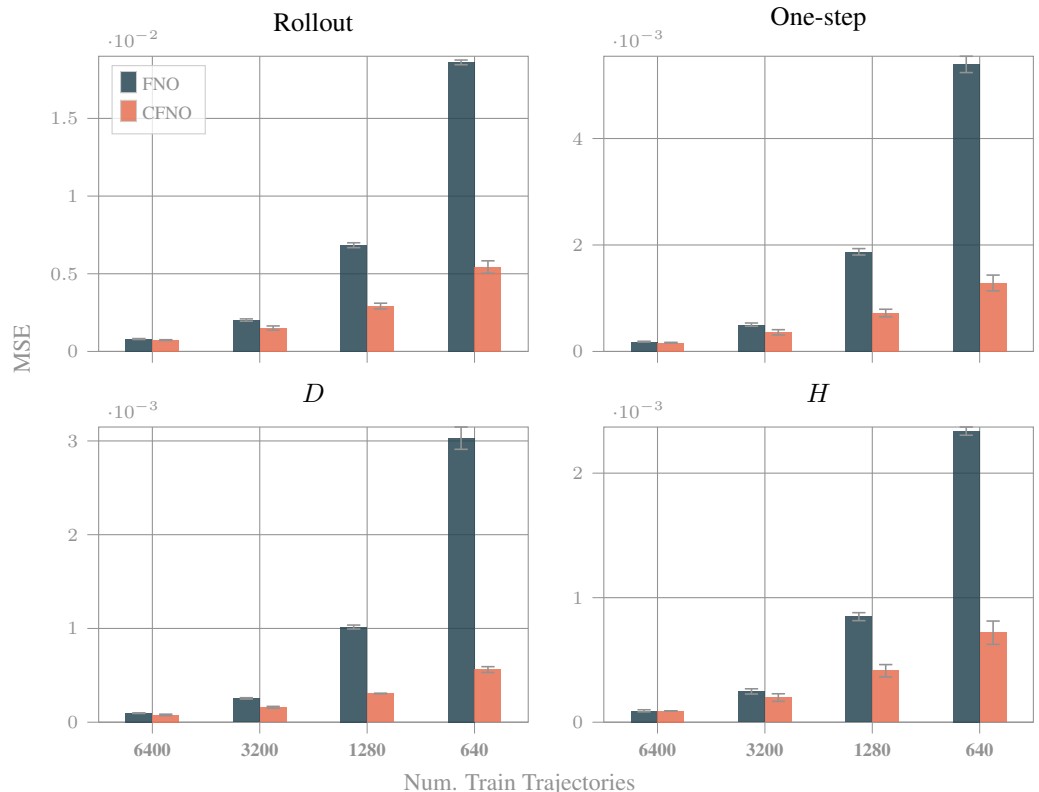

Figure 17: Results on the Maxwell equations obtained by Fourier based architectures using a **two timestep history input**. Rollout loss, one-step loss, displacement field $D$ loss, and magnetization field $H$ loss are reported for FNO and CFNO architectures. Models are trained on four training sets with increasing number of trajectories.

Table 4: Model comparison on four different metrics for neural PDE surrogates which are trained on the Maxwell equations training datasets of varying size. Results are obtained by using a **two timestep history input**. Error bars are obtained by running experiments with three different initial seeds.

| | | SMSE | | | |
|---|---|---|---|---|---|
| METHOD | Trajs. | $D$ | $H$ | onestep | rollout |
| FNO | 640 | $0.0030 \pm 0.0006$ | $0.002\,33 \pm 0.000\,50$ | $0.0054 \pm 0.0011$ | $0.0186 \pm 0.0083$ |
| CFNO | | $0.0006 \pm 0.0001$ | $0.000\,72 \pm 0.000\,10$ | $0.0013 \pm 0.0002$ | $0.0054 \pm 0.0023$ |
| FNO | 1280 | $0.0010 \pm 0.0002$ | $0.000\,85 \pm 0.000\,20$ | $0.0019 \pm 0.0004$ | $0.0068 \pm 0.0036$ |
| CFNO | | $0.0003 \pm 0.0001$ | $0.000\,41 \pm 0.000\,10$ | $0.0007 \pm 0.0002$ | $0.0029 \pm 0.0016$ |
| FNO | 3200 | $0.0003 \pm 0.0001$ | $0.000\,25 \pm 0.000\,10$ | $0.0005 \pm 0.0001$ | $0.0020 \pm 0.0011$ |
| CFNO | | $0.0002$ | $0.000\,20 \pm 0.000\,10$ | $0.0004 \pm 0.0001$ | $0.0015 \pm 0.0009$ |
| FNO | 6400 | $0.0001$ | $0.000\,09$ | $0.0002$ | $0.0008 \pm 0.0004$ |
| CFNO | | $0.0001$ | $0.000\,09$ | $0.0002$ | $0.0007 \pm 0.0004$ |

## D  RELATED WORK

This appendix supports detailed discussions of how our work relates to complex and quaternion neural networks, to work on Clifford algebras and Clifford Fourier transforms in computer vision, to Fourier Neural Operators, equivariant neural networks and geometric deep learning approaches, to neural operator learning and neural PDE surrogates.

**Clifford (geometric) algebra and Clifford Fourier transform.**  (Real) Clifford algebras (also known as geometric algebras), as an extension of elementary algebra to work with geometrical objects such as vectors are extensively discussed in Suter (2003); Hestenes (2003); Dorst et al. (2010); Hestenes (2012); Renaud (2020). Compared to other formalisms for manipulating geometric objects, Clifford algebras are tailored towards vector manipulation of objects of different dimensions. Hypercomplex and quaternion Fourier transforms are extensively discussed in Ell (1992; 1993); Ell & Sangwine (2006); Ell et al. (2014). This work heavily builds on the concepts introduced in Ebling & Scheuermann (2003; 2005); Ebling (2006). Comprehensive summaries of Clifford and quaternion Fourier transforms can be found in Hitzer & Sangwine (2013); Brackx et al. (2013); Hitzer (2021). Clifford algebras and Clifford Fourier transforms are already deployed to solve PDEs numerically in Alfarraj & Wei (2022). More precisely, the Clifford-Fourier transform is used to solve the mode decomposition process in PDE transforms.

**Clifford neural networks.**  Neural networks in the Clifford domain were proposed already in 1994 by Pearson & Bisset (1994), and later by Pearson (2003). These works put the emphasis on the geometric perceptron (Melnyk et al., 2021), i.e. how to recast vanilla multilayer perceptrons (MLPs) as Clifford MLPs. Similarly, (Hoffmann et al., 2020) generalized from complex numbers and quaternions to a set of alternative algebras. Besides Clifford MLPs, Clifford algebras have been used in recurrent neural networks (RNNs) (Kuroe, 2011), and have been used to formulate quantum neural networks (Trindade et al., 2022). Their applicability to neural computing has been studied in general (Buchholz & Sommer, 2001; Buchholz, 2005), exploring global exponential stabilities of Clifford MLPs with time-varying delays and impulsive effects. Probably the most related wors are: (i) Zang et al. (2022) who build geometric algebra convolution networks to process spatial and temporal data of 3D traffic data. Multidimensional traffic parameters are encoded as multivectors which allows to model correlation between traffic data in both spatial and temporal domains. (ii) Spellings (2021) who build rotation- and permutation-equivariant graph network architectures based on geometric algebra products of node features. Higher order information is built from available node inputs.

In contrast to previous works, we are the first to introduce the multivector viewpoint of field components which allows us to effectively connect Clifford neural layers with the geometric structure of the input data. We further connect neural Clifford convolutions on multivectors with various works on complex numbers and quaternions. We are further the first to introduce neural Clifford Fourier transforms.

**Complex and quaternion neural networks.**  Trabelsi et al. (2017) introduced the key components for complex-valued deep neural networks. More precisely, they introduced convolutional (Le-Cun et al., 1998) feed-forward and convolutional LSTM (Shi et al., 2015; Hochreiter & Schmidhuber, 1997) networks, together with complex batch-normalization, and complex weight initialization strategies. Quaternions are a natural extension of complex neural networks. Already in classical computer vision, quaternions as hypercomplex convolution (Sangwine & Ell, 2000) and hypercomplex correlation (Moxey et al., 2003) techniques were introduced for color image processing. Quaternion based deep learning architectures are a natural extension of complex neural networks. In quaternion neural networks (Zhu et al., 2018; Parcollet et al., 2018a; Gaudet & Maida, 2018; Parcollet et al., 2018b; 2019; 2020; Nguyen et al., 2021; Moya-Sánchez et al., 2021), concepts such as complex convolution, complex batchnorm, and complex initialization are transfered from the complex numbers $\mathbb{C}$, which are algebra-isomorph to $Cl(0,1)(\mathbb{R})$ to $Cl(0,2)(\mathbb{R})$, which is algebra-isomorph to the quaternions $\mathbb{H}$. Although Hoffmann et al. (2020) generalized these from complex numbers and quaternions to a set of alternative algebras, their tasks did not really leverage any multivector structure in data.

**Fourier Neural Operators.** Fourier Neural Operators (FNOs) (Li et al., 2020) have had tremendous impact towards improving neural PDE solver surrogates. Efficient implementations of FNO layers come as physics-informed neural networks (PINO) (Li et al., 2021b), as U-shaped network architectures (UNO) (Rahman et al., 2022b), as spectral surrogate for vision transformer architectures (Rao et al., 2021; Guibas et al., 2021), as Markov neural operators (MNO) for chaotic systems (MNO) (Li et al., 2021a), and as generative adversarial neural operators (GANOs) (Rahman et al., 2022a). Applications range from weather forecasting (Pathak et al., 2022), $CO_2$-water multiphase problems (Wen et al., 2022), multiscale method for crystal plasticity (Liu et al., 2022), seismic wave propagation (Yang et al., 2021), photoaccustic wave propagation (Guan et al., 2021), PDE-constrained control problems (Hwang et al., 2022), and for thermochemical curing of composites (Chen et al., 2021). Recently, FNOs have been successfully applied to PDEs on general geometries (Li et al., 2022b). Furthermore, universal approximation and error bounds have been studied for FNOs (Kovachki et al., 2021).

**Neural PDE solvers/surrogates.** The intersection of PDE solving, deep learning, fluid dynamics, and weather forecasting has developed into a very active hub of research lately (Thuerey et al., 2021). We roughly group recent approaches to learn neural PDE surrogates and neural PDE solvers into three categories:

(i) *hybrid approaches*, where neural networks augment numerical solvers or replace parts of numerical solvers;

(ii) *direct approaches*,

    (a) where the mapping from an initial state to a solution is learned, i.e. the solution function of the underlying PDE is approximated;

    (b) where the mapping from an initial state to a final state of an underlying PDE, i.e. the solution operator is learned, ideally as mapping between function spaces to be able to generalize across e.g. parameters.

Ad (i): Neural networks augment numerical solvers by learning data-driven discretizations for PDEs (Bar-Sinai et al., 2019) or by controlling learned approximations inside the calculation of standard numerical solver used for computational fluid dynamics (Kochkov et al., 2021). In Greenfeld et al. (2019), a prolongation is learned which maps from discretized PDE solutions to multigrid solutions, Hsieh et al. (2019) learn to modify the updates of an existing solver, Praditia et al. (2021) adopt the numerical structure of Finite Volume Methods (FVMs), and Um et al. (2020) learn a correction function of conventional PDE solvers to improve accuracy. All these approaches are hybrid approaches (Garcia Satorras et al., 2019), where the computational graph of the solver is preserved and heuristically-chosen parameters are predicted with a neural network. A different flavor of hybrid approaches can be assigned to the works of Sanchez-Gonzalez et al. (2020); Pfaff et al. (2020); Mayr et al. (2021) who predict accelerations of particles/meshes to numerical update the respective positions. Finally, PauliNet (Hermann et al., 2020) and FermiNet (Pfau et al., 2020) approximate wave-functions of many-electron systems, and thus replace the hand-crafted ansatz which is conventionally used in variational quantum Monte Carlo methods.

Ad (ii.a): Sirignano & Spiliopoulos (2018); Han et al. (2018) approximate the solution of high-dimensional Black-Scholes and Hamilton-Jacobi-Bellman equations, respectively. Physics-informed neural networks (PINNs) (Raissi et al., 2019) embed the underlying physics in the training process, and can be used to solve both forward (Jin et al., 2021) as well as backward (Raissi et al., 2020) dynamics. Zubov et al. (2021) allow automating many of these aspects under a single coherent interface.

Ad (ii.b): Guo et al. (2016) learned a surrogate CNN-based model to approximate steady-state flow field predictions, similarly Bhatnagar et al. (2019) trained a surrogate CNN-based model to predict solutions for unseen flow conditions and geometries, and Zhu & Zabaras (2018) used Baysian CNNs for surrogate PDE modeling and uncertainty quantification. Fourier Neural Operators (FNOs) (Li et al., 2020) proposed the mapping from parameter space to solution spaces, and had tremendous impact towards improving neural PDE solver surrogates. In parallel, Lu et al. (2021) introduced DeepONet, which learns mappings between function spaces, and was successfully applied to many parametric ODEs and PDEs. Both, FNOs and DeepONets have been combined with PINNs and trained in a physics-informed style (Li et al., 2022b; Wang et al., 2021). A comprehensive compar-

ison of these two neural operator approaches is done by Lu et al. (2022). Other directions include the modeling of PDE solution operators via latent space models, transformers, and graph neural networks (GNNs). Wu et al. (2022) present the modeling of the systems dynamics in a latent space with fixed dimension where the latent modeling is done via MLPs, and the encoding and decoding via CNNs, which can also be replaced by graph neural networks (GNNs). Cao (2021) propose the Galerkin transformer, a simple attention based operator learning method without softmax normalization, LOCA (Learning Operators with Coupled Attention) (Kissas et al., 2022) maps the input functions to a finite set of features and attends to them by output query locations, and Li et al. (2022a) propose a transformer which provides a flexible way to implicitly exploit the patterns within inputs. Brandstetter et al. (2022b) formulated a message passing neural network approach that representationally contains several conventional numerical PDE solving schemes. Further GNN based approaches are Lötzsch et al. (2022) who learn the operator for boundary value problems on finite element method (FEM) (Brenner et al., 2008) ground truth data, and Lienen & Günnemann (2022) who derive their GNN models from FEM in a principled way.

A practical use case for neural PDE surrogates is replacing expensive classical PDE solvers. There is however a major chicken-and-egg problem here (Brandstetter et al., 2022a; Shi et al., 2022): obtaining high quality ground truth training data for neural PDE surrogates often requires using these expensive solvers. Minimizing this data requirement is beginning to be approached in recent works. Geneva & Zabaras (2020); Wandel et al. (2020; 2022) achieve "data-free" training in various settings. "Data-free" refers to the self-supervised training steps, which are done without ground truth data. The current state-of-art generic approach is introduced in Shi et al. (2022) as the mean squared residual (MSR) loss constructed by the discretized PDE itself. However, for e.g. generating realistic initial conditions numerical solvers are still needed. Pestourie et al. (2021) identify how incorporating limited physical knowledge in the form of a low-fidelity "coarse" solver can allow training PDE surrogate models with an order of magnitude less data. Another direction to improve data efficiency is by exploiting the Lie point symmetries of the underlying PDEs, either via data augmentation (Brandstetter et al., 2022a) or by building equivariant PDE surrogates (Wang et al., 2020b). Our current work in a way also improves data efficiency by capturing the inductive bias appropriate for multivector fields. Overall we believe hybrids of such approaches are going to be necessary for making neural PDE surrogates of practical use in many domains.

Neural PDE surrogates for fluid flow and weather forecasting applications are gaining momentum. In weather forecasting, Pathak et al. (2022) introduced FourCastNet as high-resolution weather modeling built on Adaptive Fourier Neural Operators (Guibas et al., 2021), Keisler (2022) successfully applied a graph neural network based approach to weather forecasting, Rasp & Thuerey (2021) achieved data-driven medium-range weather prediction with a ResNet which was pretrained on climate simulations, Weyn et al. (2020) use CNNs on a cubed sphere for global weather prediction, Weyn et al. (2021) forecast weather sub-seasonally with a large ensemble of deep-learning weather prediction models, Arcomano et al. (2020) build a reservoir computing-based, low-resolution, global prediction model, and MetNet (Sønderby et al., 2020) takes as input radar and satellite data to forecast probabilistic precipitation maps. Finally, data assimilation is improved by deep learning techniques in Frerix et al. (2021) and Maulik et al. (2022). Similarly, in fluid dynamics, Ma et al. (2021a) applied U-Nets (Ronneberger et al., 2015) to achieve physics-driven learning of steady Navier-Stokes equations, Stachenfeld et al. (2021) learned coarse models for turbulence simulations, TF-Net (Wang et al., 2020a) introduced domain-specific variations of U-Nets along with trainable spectral filters in a coupled model of Reynolds-averaged Navier-Stokes and Large Eddy Simulation.

This exhaustive list of neural PDE solver surrogates shows that many of the architectures are based on convolutional or Fourier layers. For these two, Clifford layers are applicable as a drop-in replacement in almost all cases. For graph neural network and attention based architectures, we leave the implementation of respective Clifford counterparts to future work.

**Geometric deep learning.** The core idea of geometric deep learning (Bronstein et al., 2017; 2021) is to exploit underlying low-dimensionality and structure of the physical world, in order to design deep learning models which can better learn in high dimensional spaces. Incorporating underlying symmetries would be one way to achieve this. If done correctly, it can drastically shrink the search space, which has proven to be quite successful in multiple scenarios. The most obvious examples are CNNs (Fukushima & Miyake, 1982; LeCun et al., 1998), where the convolution operation commutes with the shift operator, and thus provides a way to equip layers and subsequently networks

with translation equivariant operations. Group convolution networks (Cohen & Welling, 2016a; Kondor & Trivedi, 2018; Cohen et al., 2019) generalize equivariant layers beyond translations, i.e. provide a concept of how to build general layers that are equivariant to a broader range of groups, such as rotation groups. An appealing way of how to build such group equivariant layers is via so-called steerable basis functions (Hel-Or & Teo, 1998), which allow to write transformation by specific groups as a linear combination of a fixed, finite set of basis functions. This concept leads to steerable group convolution approaches (Cohen & Welling, 2016b; Worrall et al., 2017). Two concrete examples are: (i) circular harmonics, which are respective basis functions for building layers that are equivariant to the group SO(2), the rotation group in 2 dimensions (Worrall et al., 2017; Weiler & Cesa, 2019); (ii) spherical harmonics, which are respective basis functions for building layers that are equivariant to the group SO(3), the rotation group in 3 dimensions (Weiler et al., 2018; Geiger & Smidt, 2022; Brandstetter et al., 2021). The similarity to multivector fields becomes more obvious if we have a closer look at spherical harmonics, which are defined as homogeneous polynomials of degree $l$, where the $l = 0$ case corresponds to scalars, the $l = 1$ case to vectors, and $l \geq 2$ to higher order objects. Finally, Jenner & Weiler (2021) built steerable PDE operators such as curl or divergence as equivariant neural network components.

**Grouped convolution.** In their seminal work, Krizhevsky et al. (2012) introduced filter grouping, which allowed them to reduce the parameters in CNNs. The respective grouped convolutions (not to be confused with *group* convolutions) divide the filter maps at channel dimension, as the channel dimension most of the time increases strongly for deeper layers, and thus dominates the parameter count. Subsequent work showed that it is beneficial to additionally shuffle the channels for each filter group (Zhang et al., 2018), and to adaptively recalibrate channel-wise feature responses (Hu et al., 2018). All these approaches can be seen in the wider spectrum of effective model scaling (Tan & Le, 2019; Sandler et al., 2018).

Clifford convolutions in contrast do not have groupings in the channel dimensions, but instead group together elements as multivectors. In Clifford convolution, the Clifford kernel is therefore a constrained object where weight blocks appear multiple times (due to the nature of the geometric product). Thus, Clifford convolutions are more parameter efficient than standard convolutions, and all tricks of effective model scaling could in principle be applied on top of Clifford convolutions. Findings from Hoffmann et al. (2020) with respect to higher compute density of alternative algebras are applicable to our work as well.

## E GLOSSARY

This short appendix summarizes notations used throughout the paper (Table 5), and contrasts the most fundamental concepts which arise when using Clifford algebras.

Table 5: Notations used throughout the paper.

| Notation | Meaning |
|---|---|
| $e_1, e_2, e_3$ | Basis vectors of the *generating* vector space of the Clifford algebra. |
| $e_i \wedge e_j$ | Wedge (outer) product of basis vectors $e_i$ and $e_j$. |
| $e_i \cdot e_j = \langle e_1, e_j \rangle$ | Inner product of basis vectors $e_i$ and $e_j$. |
| $e_1 e_2, e_3 e_1, e_2 e_3$ | Basis bivectors of the vector space of the Clifford algebra. |
| $e_1 e_2 e_3$ | Basis trivector of the vector space of the Clifford algebra. |
| $i_2 = e_1 e_2$ | Pseudoscalar for Clifford algebras of grade 2. |
| $i_3 = e_1 e_2 e_3$ | Pseudoscalar for Clifford algebras of grade 3. |
| $x$ | Euclidean vector $\in \mathbb{R}^n$. |
| $x \wedge y$ | wedge (outer) product of Euclidean vectors $x$ and $y$. |
| $x \cdot y = \langle x, y \rangle$ | Inner product of vectors $x$ and $y$. |
| $\boldsymbol{a}$ | Multivector. |
| $\boldsymbol{ab}$ | Geometric product of multivectors $\boldsymbol{a}$ and $\boldsymbol{b}$. |
| $\hat{i}, \hat{j}, \hat{k}$ | Base elements of quaternions. |

**Geometric, Exterior, and Clifford algebras.** A geometric algebra is a Clifford algebra of the real numbers. Since we are only using $Cl_{2,0}(\mathbb{R})$, $Cl_{0,2}(\mathbb{R})$, and $Cl_{3,0}(\mathbb{R})$, we are effectively working with geometric algebras. The exterior or Grassmann algebra is built up from the same concepts of scalars, vectors, bivectors, ..., $k$-vectors, but only exterior (wedge) products exist. Therefore, the exterior algebra has a zero quadratic form (all base vectors square to zero). Clifford algebras are a generalization thereof with nonzero quadratic forms.

**Complex numbers, quaternions, hypercomplex numbers.** Hypercomplex numbers are elements of finite-dimensional algebras over the real numbers that are unital, i.e. contain a multiplicative identity element, but not necessarily associative or commutative. Elements are generated for a basis $\{\hat{\imath}, \hat{\jmath}, \ldots\}$ such that $\hat{\imath}^2, \hat{\jmath}^2, \ldots \in \{-1, 0, 1\}$. Complex numbers, quaternions, octonions are all hypercomplex numbers which can be characterized by different Clifford algebras. The bivector, trivector (and higher objects) of the Clifford algebras directly translate into basis elements of the respective algebras. For example, quaternions (which are of the form $a + b\hat{\imath} + c\hat{\jmath} + d\hat{k}$, where $\hat{\imath}^2 = \hat{\jmath}^2 = \hat{k}^2 = -1$) are isomorphic to the Clifford algebra $Cl_{0,2}(\mathbb{R})$ where the basis element $e_1$, $e_2$, and $e_1 e_2$ directly translate to $\hat{\imath}, \hat{\jmath}, \hat{k}$.

**Spinor.** Spinors arise naturally in discussions of the Lorentz group, the group to describe transformations in special relativity. One could say that a spinor is the most basic sort of mathematical object that can be Lorentz-transformed. In its essence, a spinor is a complex two-component vector-like quantity in which rotations and Lorentz boosts (relativistic translations) are built into the overall formalism. More generally, spinors are elements of complex vector spaces that can be associated with Euclidean vector spaces. However, unlike vectors, spinors transform to their negative when the space is rotated by $360°$. In this work, the subalgebra $Cl^0(2,0)(\mathbb{R})$, spanned by even-graded basis elements of $Cl_{2,0}(\mathbb{R})$, i.e. 1 and $e_1 e_2$, determines the space of spinors via linear combinations of 1 and $e_1 e_2$. It is thus isomorphic to the field of complex numbers $\mathbb{C}$. Most notably, spinors of $Cl_{2,0}(\mathbb{R})$ commute with the Fourier kernel, whereas vectors do not. For a detailed introduction to spinors we recommend Steane (2013), and the comprehensive physics book of Schwichtenberg (2015).

**Pseudoscalar.** A pseudoscalar – unlike a scalar – changes sign when you invert the coordinate axis. The easiest example of a pseudoscalar is the scalar triplet product of three arbitrary vectors of an Euclidean vector space $x, y, z \in \mathbb{R}^n$ with inner product $\langle ., . \rangle$. The scalar triplet product becomes negative for any parity inversion, i.e. swapping any two of the three operands: $x \cdot (y \times z) = -x \cdot (z \times y) = -y \cdot (x \times z) = -z \cdot (y \times x)$.

**Scalar field, vector field.** A *field* is any (physical) quantity which takes on different values at different points in space (space-time). A scalar field is map $\mathbb{D} \to \mathbb{R}$, where $\mathbb{D} \subseteq \mathbb{R}^n$. A vector field is map $\mathbb{D} \to \mathbb{R}^n$, where $\mathbb{D} \subseteq \mathbb{R}^n$. For example, $n = 2$ results in a vector field in plane, and $n = 3$ in a vector field in space. For an interesting history of the evolution of the concept of fields in physics we recommend Mirowski (1991); McMullin (2002). In Table 6, we list various important vector and scalar fields for comparison.

Table 6: Examples of various vector and scalar fields. Vector fields ascribe a vector to each point in space, e.g. force, electric current (stream of charged particles), or velocity. Scalar fields on the other hand collate each field point with a scalar value such as temperature.

| Example | Field quantity | Type | Coordinates |
|---|---|---|---|
| Gravitational field (strength) | Force per unit mass ($N/kg$) | Vector | $\mathbb{R}^3 \to \mathbb{R}^3$ |
| Electric field (strength) | Force per unit electric charge ($N/C$) | Vector | $\mathbb{R}^3 \to \mathbb{R}^3$ |
| Magnetic field (strength) | Electric current per meter ($A/m$) | Vector | $\mathbb{R}^3 \to \mathbb{R}^3$ |
| Pressure field | Force per unit square ($N/m^2$) | | |
| | = energy per unit volume ($J/m^3$) | Scalar | $\mathbb{R}^3 \to \mathbb{R}$ |
| Mean sea level pressure | Pressure field at mean sea level | Scalar | $\mathbb{R}^2 \to \mathbb{R}$ |
| Flow velocity field | Change of point along its streamline[1] ($v$) | Vector | $\mathbb{R}^2 \to \mathbb{R}^2, \mathbb{R}^3 \to \mathbb{R}^3$ |
| Flow speed field | Length of flow velocity vector ($|v|$) | Scalar | $\mathbb{R}^2 \to \mathbb{R}, \mathbb{R}^3 \to \mathbb{R}$ |
| Wind velocity field | Air flow velocity field ($v$) | Vector | $\mathbb{R}^2 \to \mathbb{R}^2, \mathbb{R}^3 \to \mathbb{R}^3$ |
| Temperature field | Temperature at space point ($K$) | Scalar | $\mathbb{R}^3 \to \mathbb{R}$ |
| Signed distance field (SDF) | Signed distance | Scalar | $\mathbb{R}^3 \to \mathbb{R}$ |
| Occupancy field | Occupancy | Scalar | $\mathbb{R}^3 \to \mathbb{R}$ |

[1] Streamlines are a family of curves whose tangent vectors constitute the velocity vector field of the flow. Streamlines differ over time when the flow of a fluid changes. The flow velocity vector field itself shows the direction in which a massless fluid element will travel at any spatial coordinate in time, and therefore describes and characterizes a fluid.

