# OpenReview forum: "Clifford Neural Layers for PDE Modeling"
_ICLR.cc/2023/Conference — ICLR 2023 poster_

### Official Review · Reviewer_S1uq · 2022-10-21

**Confidence:** 5
**Correctness:** 4
**Technical Novelty And Significance:** 3
**Empirical Novelty And Significance:** 2
**Recommendation:** 6

**Clarity, Quality, Novelty And Reproducibility:**

The paper is clearly written. As some point it goes a bit too much into details that might a) not be too interesting to the ICLR audience, b) not very new for people who are aware about the concepts of Clifford algebras.

To my knowledge, using the multiplicative part of Clifford algebras is rather new to the community. Nonetheless, it is important to note that the complex numbers can be seen as a special case of a Clifford algebra ($\mathbb{C}=Cl_{1,0}$). And complex neural networks have been explored, especially for the physical motivated application of radar perception. This fact is missing in the manuscript.

I have some concerns w.r.t. the reproducibility about the experiments, since the used datasets are not mentioned.

**Strength And Weaknesses:**

Strength:
+ Every details of the Clifford algebra as well as the two new operations (C-convolutions and C-FT) is explained extensively
+ The conducted experiments are based on well-known architecture where only the basic operations are replaced
+ The experiments focus on physical problems, which is the motivation for these new operations

Weakness:
- For the experiments it is not clear how the data is generated. Thus, results are difficult to reproduce.
- Section 2 takes 2 pages to explain well-known facts about Clifford algebras. This could be explained in a more compact manner.
- The paper including appendices is too excessive with more than 55 pages. It appears a bit too long for a conference paper

**Summary Of The Paper:**

The paper addresses the problem of modelling a PDE in the means of neural networks. The main motivation stems from the fact that Clifford algebras is a very popular tool in modelling physical relationships. As a consequence, the paper proposes to use the product of the algebra (instead of the products of a scalar field) to define both, convolutions and FT (Fourier transforms).

With these basic operations at hand, solutions to both, Navier-Stokes and Maxwell equations are explored. To this end, standard neural networks like ResNet and FNOs (Fourier Neural Operator Networks) are used and convolutions resp. FT are replaced with the Clifford counterpart.

**Summary Of The Review:**

Overall, I consider the idea behind the paper very nice and I think the concept of Clifford neural networks should be presented to the machine learning community. ~Due to the missing datasets, I consider the paper below the acceptance threshold. If the authors can provide the datasets for the final version, I would raise my vote to an acceptance vote.~

Edit: Since the data set will be provided, I am happy to raise my vote to an acceptance vote.

---

### Official Review · Reviewer_ugoG · 2022-10-22

**Confidence:** 3
**Correctness:** 3
**Technical Novelty And Significance:** 4
**Empirical Novelty And Significance:** 3
**Recommendation:** 8

**Clarity, Quality, Novelty And Reproducibility:**

The presentation is clear and the provided Clifford algebras background is very helpful. The proposed approach is novel. Experiments are well conducted and the results look very promising. Overall the quality is very good.

**Strength And Weaknesses:**

Strengths:

- The proposed Clifford neural layers are novel and technically solid. Based on the observation that different fields in PDE are often related, the authors propose to leverage Clifford algebras to naturally incorporate this inductive bias into neural PDE surrogate models. As far as I know, this is the first time that people introduces Clifford algebras to neural PDE modeling.
- The experiment results look promising.  Through experiments in three different settings, Cliffod neural layers bring consistent improvement compared to their standard counterparts. This is promising for an first attempt of this type.

Weaknesses:

- Some experimental results need further explanations or investigations. For example, I noticed that in Figure 7 ResNet based results, CResNet gives larger error than ResNet for low numbers of trajectories. This is different from 2D Navier-Stokes (Figure 6). It also contradicts with the claim in Sec. 5 ("the performance gap increased with less data availability across all settings"). Any explanation for this particular result.
- For 3D Maxwell's equations, the paper only shows the results for FNO based architectures. There is no explanation why ResNet based architectures are note tested here. If possible, I think it's better to include them for completeness.


**Summary Of The Paper:**

This paper introduces Clifford neural layers for both convolution and Fourier operations in the context of deep learning. The main motivation is that Clifford algebras can describe the algebraic property (e.g. multiplication, addition) of multivector fields consisting scalar, vector, as well as higher-order components. The resulting Clifford neural layers are universally applicable, and the authors explore their usage in neural PDE surrogate models (for solving 2D Navier-Stokes and 3D Maxwell's equations). By replacing convolution and Fourier operations in common neural PDE surrogates by their Cliffold couterparts, the authors show that Clifford neural layers consistently improves their performance.

**Summary Of The Review:**

The proposed Clifford neural layers is a novel and solid technical contribution. The results look promising for the first attempt of this kind. Though there are some obvious limitations (e.g., slow runtime), they are potentially solvable. Therefore, I recommend for acceptance.

---

### Official Review · Reviewer_TyjS · 2022-10-23

**Confidence:** 2
**Clarity, Quality, Novelty And Reproducibility:** See above.
**Correctness:** 4
**Technical Novelty And Significance:** 4
**Empirical Novelty And Significance:** 3
**Recommendation:** 8

**Strength And Weaknesses:**

Strengths:
1. The theory is comprehensive and self-contained. The results are promising and supportive.

Weakness:
1. In Fig. 7, it is not discussed why CResNet works worse than ResNet in some of the experiments. Please add some analysis here.
2. Clifford algebra seems to be general and can potentially be applied to more than Neural PDE, as long as there are inputs of multiple data types. It would be great if the author can provide more insight into other applicable scenarios and future directions to achieve this.

**Summary Of The Paper:**

This paper introduces a new architecture for Neural PDEs based on Clifford algebra. By incorporating fields with different dimensions together into one algebra system, the method yields better results when used in neural networks due to better coupling among the data.

**Summary Of The Review:**

It is a very comprehensive and self-supportive paper. I would recommend acceptance.

---

### Official Review · Reviewer_EJCG · 2022-10-24

**Confidence:** 2
**Correctness:** 3
**Technical Novelty And Significance:** 3
**Empirical Novelty And Significance:** 3
**Recommendation:** 6

**Clarity, Quality, Novelty And Reproducibility:**

The paper is clearly written and the main method is effectivley presented
Some information about the experiement section is missing
The codes are not provided with the paper.

**Strength And Weaknesses:**

The main strenghts of the paper are:

1) The paper is well-written and the main ideas are effectively presented.
2) The background section covers the pre-requisite knowledge required to understand the clifford neural layers
3) A novel method is proposed in this paper which can be useful in the Physical science domain


The main weaknesses of the paper in my opinion are:
1) As someone who does not work in the Physical sciences I could not get much information about the datasets used from the paper. I am not sure what the training sets are for these dataset and hence can't make an informed judgement on the quality of the method. [I did not go through the supplementary material in any detail ]
2) The baseline information is not very clear form the paper.


**Summary Of The Paper:**

In this paper the authors present Clifford neural layers that can be useful in the areas of fluid dynamics, weather forecasting, and the modeling of physical systems in general.  The author emipirically validate the use of Cifford layer by incorporating them in the common neural PDE surrogates for 2D Navier-Stokes, weather modeling, and 3D Maxwell equations.

**Summary Of The Review:**

The paper present clifford layer which can be used for application in Physical sciences. The method is tested on three datasets and the results look promising. The experiment section is missing some details which makes it a bit difficult for non-experts to read.

---

### Author Response · Authors · 2022-11-11
**General response**

We thank all the reviewers for their constructive feedback. We were pleased to see that reviewers found the idea novel, the paper well-written, and the theory comprehensive and self-contained.

Introducing the multivector viewpoint which allows us to model various field components together, along with respective Clifford convolution and Clifford Fourier operations on such multivectors is a new direction in Deep Learning. We strongly believe that this direction can be extended and exploited in many different fields of Deep Learning, e.g. computer vision. Therefore, we are keen to present this work at a large Deep Learning venue.

However, this comes with the hardest challenge of this paper, namely to write it in a manner that is comprehensive and yet as self-contained as possible, where the long appendix solely serves the purpose of additional information. All key details needed should already be found in the main paper.

One common feedback amongst the reviewers was that the experiments lack detail. We re-worked the paper such that the experimental section gets more space, which allows us to discuss the experiments, the architectures, the datasets, and finally the results in much greater detail. This helped to strongly improve the paper.

For better visibility, and as requested by two reviewers, we have uploaded the codebase for data production, as well as links to the datasets itself. We will release the full codebase upon publications, pseudocode of the respective layers can already be found in appendix B.6.

Once again, we greatly appreciate the comments and suggestions by the reviewers as we believe their incorporation led to valuable improvements to the paper in terms of clarity, especially for improving the thoroughness of the experiments.

---

### Public Comment · ~Shuhao_Cao1 · 2023-02-06
**Some questions and comments**

I was browsing PDE-related papers and found this nice work on Maxwell's equations, and I thought that using channels to represent the stacking of multivector is the nicest DL-ish port of traditional methods among all ICLR 2023 papers.

Q: is the Clifford layer gauge invariant? For example, when there is no source in Maxwell's equation (in free space), if there is a scalar field smooth enough that does not vary in the time dimension, then adding its gradient to the solution magnetic field still makes it a solution.

Another two questions about the data generation:
- Is Phi-flow for NS using the pressure-velocity formulation? or vorticity+pseudospectral like the FNO paper uses?
- In Maxwell's data gen, are $E$ and $B$ later post-processed to live on the same time step (since technically they live a staggered temporal grid due to FDTD+leapfrog)?

Also I found a minor typo at the beginning of Appendix E: "summmarizes".

Last are some comments about references:
- Maybe Hestenes's 1984 book should be referenced on the unified representation of Maxwell's equation without Helmholtz decomposition aside from his 2002 lecture notes.
- Several other papers on Transformers and PDE-operator learning earlier than Li et al, Transformer for partial differential equations’ operator learning. 2022a:
    - Cao, Choose a Transformer: Fourier or Galerkin, NeurIPS 2021.
    - Kissas et al, Learning operators with coupled attention, JMLR 2022.

---

> ### Author Response · Authors · 2023-02-07
> **Thank you very much for the clever questions / comments**
>
> Let us thank you for the clever questions and comments, which we all happily include in the final version of the paper.
>
> -  Regarding **gauge invariance**: At the moment we have designed the architectures such that the hidden representation is always a full multivector and the primary transformation is done via the geometric product. Thus, "property" preserving transformation, which are often used as fundamental building blocks in geometric algebra, are not necessarily ensured. A typical example is that in geometric algebra only isometry preserving transformations are possible via the so-called sandwich operation, whereas the geometric product allows for all sorts of mixing behavior. This also relates to gauge invariance of Maxwell's equations, so far this is not guaranteed. But we are working on an update which comes with more formal guarantees, achieved via precisely such geometric algebra operations which ensure that transformations within neural network layers obey geometric criteria.
>
> - Regarding **pressure-velocity formulation**: This paper uses the pressure-velocity formulation - an thorough comparison between those two formulations and performance of operator learning methods can be found for example in Gupta et al, Towards Multi-spatiotemporal-scale Generalized PDE Modeling. The vorticity formulation makes it difficult to "group" fields into multivector fields.
>
> - Regarding **leap-frog formulation**: This is an excellent catch! We have taken the data as it is and tested if our inductive bias improves the architectures. We fully agree that correcting for the staggered temporal grid is a very interesting further test.
>
> - Regarding **references**: Thanks a lot, highly appreciated. We include them in the discussion.

---

### Decision · Program_Chairs · 2023-01-20

**Decision:**

Accept: poster

**Justification For Why Not Higher Score:**

The paper will probably be interesting for a limited audience. The core of their contribution is based on the Clifford algebra formalism and, as mentioned in the review, it is difficult to present in a limited time.

**Justification For Why Not Lower Score:**

The evaluators agree to an acceptance, with well-argued reasons.

**Metareview: Summary, Strengths And Weaknesses:**

The paper addresses the problem of building ML surrogate models for solving PDEs. The authors introduce Clifford neural layers, that can model the interactions between different observation fields and among the components of these fields. Clifford layers can be used in place of e.g. convolutions and Fourier operators in existing NN architectures. This development relies on Clifford algebras that can be used to model physical relationships between vector fields. The authors evaluate this idea on a variety of 2D and 3D PDEs with two NN architectures: convolutions in ResNets and Fourier operations in Fourier Neural Operators are respectively replaced by their Clifford counterparts. They show that the new components improve the performance.

All reviewers consider this to be an original and solid contribution, and think that the presentation to the ML community of the concepts of Clifford algebra and their use for NN components is worthwhile. A challenge in this paper is to introduce these Clifford algebra concepts. The authors provide an introduction in the main text and a lengthy appendix. There were several exchanges with the reviewers in order to simplify the presentation: the final result is considered acceptable. Several reviewers mentioned insufficient details for the experiments and required access to data production codebase and datasets. The authors added a more detailed description of the results and provided access to the data, this was acknowledged as a satisfying addition.

**Note From Pc:**

if the above contains the word "oral" or "spotlight" please see: "oral" presentation means -> notable-top-5% and "spotlight" means -> notable-top-25%. As stated in our emails, we are disassociating presentation type from AC recommendations